

# Northern vs. southern hemisphere differences in the stratospheric influence on variability in tropospheric nitrous oxide

Cynthia D. Nevison[1], Qing Liang[2], Paul A. Newman[2], Britton B. Stephens[3], Geoff Dutton[4,5], Xin Lan[4,5], Roisin Commane[6], Yenny Gonzalez[7,8], Eric Kort[9]

[1]Institute for Arctic and Alpine Research, University of Colorado, Boulder, CO, USA
[2]NASA Goddard Space Flight Center, Greenbelt, MD, USA
[3]NSF National Center for Atmospheric Research, Boulder, CO, USA
[4]Global Monitoring Laboratory, NOAA Earth System Research Laboratory, Boulder, CO, USA
[5]Cooperative Institute for Research in Environmental Sciences (CIRES), University of Colorado, Boulder, CO, USA
[6]Department of Earth & Environmental Sciences, Lamont-Doherty Earth Observatory, Columbia University, Palisades, NY, USA
[7]CIMEL Electronique, Paris, 75011, France
[8]Izana Atmospheric Research Center, AEMET, Santa Cruz de Tenerife, 38001, Spain
[9]Department of Climate & Space Sciences & Engineering, University of Michigan, Ann Arbor, MI, USA

*Correspondence to*: Cynthia D. Nevison (cynthia.nevison@colorado.edu)

**Abstract.** We present a chemistry-climate model with a tagged stratospheric nitrous oxide ($N_2O$) tracer that predicts distinct seasonal cycles in tropospheric $N_2O$ caused by descent of $N_2O$-depleted stratospheric air in polar regions. We identify similar phenomena in recently available aircraft profiles from global campaigns and routine monitoring. Long-term atmospheric measurements from the National Oceanic and Atmospheric Administration (NOAA) global surface monitoring network provide additional support for a significant impact on surface $N_2O$ originating from the stratosphere. In the northern hemisphere, the

NOAA surface $N_2O$ atmospheric growth rate anomaly is negatively correlated with the previous winter's polar lower stratospheric temperature. This negative correlation is consistent with increased (decreased) transport in years with a strong (weak) Brewer Dobson circulation of warm, $N_2O$-depleted air from the middle and upper stratosphere into the lower stratosphere, with subsequent cross-tropopause transport of the $N_2O$-depleted air into the troposphere. In the southern hemisphere, polar lower stratospheric temperature is correlated to monthly summertime anomalies in tropospheric $N_2O$ as it

descends into its seasonal minimum, a result that is supported by aircraft data as well as the chemistry-climate model. However, the $N_2O$ atmospheric growth rate anomaly in the southern hemisphere is better correlated to the stratospheric quasi-biennial oscillation (QBO) index, as well as the El Niño Southern Oscillation index, than to polar lower stratospheric temperature. These hemispheric differences in the factors influencing the $N_2O$ atmospheric growth rate are consistent with known atmospheric dynamics and the complex interaction of the QBO with the Brewer Dobson circulation.





## 1 Introduction

Nitrous oxide (N₂O) is an important ozone-depleting substance and long-lived greenhouse gas, with a global warming potential (GWP) of 265 relative to CO₂ over a 100 year time horizon (*WMO*, 2018). N₂O has an atmospheric lifetime of about 120 years and is destroyed slowly in the stratosphere by both photolysis and oxidation, with a fraction of the oxidation product yielding NOₓ, a catalyst of stratospheric ozone destruction (*Crutzen*, 1970; *Prather et al.*, 2015). N₂O has abundant natural microbial sources in soil, freshwater and oceans, which account for the majority of global emissions, although anthropogenic sources are becoming increasingly important (*Canadell et al.*, 2021).

The atmospheric N₂O concentration has risen from about ~270 ppb preindustrially to 336 ppb by 2022 (*MacFarling-Meure et al.*, 2006; *Lan et al.*, 2022). This rise has been attributed largely to the Haber-Bosch process of industrial N fixation for the production of agricultural fertilizer, which has increased the N substrate available to nitrogen cycling microbes (*Park et al.*, 2012). Recent evidence suggests that N₂O is increasing at an accelerating rate in the atmosphere, possibly due to a nonlinear response of microbes to increasing N inputs in intensively fertilized agricultural systems (*Thompson et al.*, 2019; *Liang et al.*, 2022).

Interannual variability in the atmospheric growth rate (AGR) and small-amplitude seasonal cycles (in the range of 0.1 – 0.3% of the background mixing ratio) are detectable in the high precision N₂O measurements made in recent decades (*Nevison et al.*, 2004; 2007; 2011; *Jiang et al.*, 2007; *Thompson et al.*, 2013). A few studies have inferred information about surface biogeochemical sources based on the observed seasonal cycle in atmospheric N₂O. However, these studies have cautioned that the transport of N₂O-depleted air from the stratosphere is a major cause of both seasonal and interannual variability in surface N₂O, which complicates the interpretation of surface emission signals (*Nevison et al.*, 2005; 2011; 2012; *Thompson et al.*, 2014; *Ray et al.*, 2020; *Ruiz et al.*, 2021). Other studies have argued that El Niño Southern Oscillation (ENSO) cycles are likely the major driver of interannual variability in tropospheric N₂O (*Ishijima et al.*, 2009; *Thompson et al.*, 2013) or that ENSO-driven variability can obscure the influence of the stratosphere in some years (*Ruiz et al.*, 2021).

Studies of the stratospheric influence on surface N₂O variability have differed with respect to the relative impact on the Northern Hemisphere (NH) vs. Southern Hemisphere (SH). *Ray et al.* (2020) found direct correlations between the stratospheric QBO at 50 hPa, lagged 8-12 months, and the observed NOAA surface station N₂O AGR, but in the SH only. They hypothesized that the correlation of QBO and N₂O AGR was less evident in the NH due to the increased influence of surface emissions. *Ruiz et al.* (2021) found that, despite a clear QBO correlation to N₂O loss rates in the tropical middle stratosphere, variability in N₂O at the surface appeared to be governed by stratosphere-troposphere exchange (STE) dynamics in the lowermost stratosphere, rather than directly by the QBO. They showed evidence for a coherent influence of STE on the surface N₂O seasonal cycle in the NH but not the SH. *Nevison et al.* (2011) argued for a STE-driven N₂O seasonal minimum



in both hemispheres, based on significant correlations between surface $N_2O$ seasonal anomalies and stratospheric temperature as well as polar vortex breakup indices.


A better grasp of the controls on tropospheric $N_2O$ variability has potentially important implications for the interpretation of biogeochemical signals in $N_2O$ data. If abiotic factors associated with the downward transport of $N_2O$-depleted air from the stratosphere contribute significantly to variability, they must be disentangled from the data before inferring information about surface biogeochemistry and emissions. Understanding the influence of stratospheric variability on surface $N_2O$ also may

provide insight into anomalous changes in the AGR of CFC-11, which has a stratospheric sink similar to that of $N_2O$ (*Ray et al.*, 2020; *Ruiz et al.*, 2021; *Lickley et al.*, 2021).

This paper analyzes the causes of interannual variability in both the seasonal cycle and the AGR of tropospheric $N_2O$. The methodology includes examination of vertical profiles of atmospheric $N_2O$, collected by aircraft campaigns and routine monitoring, and analysis of output from the Goddard Earth Observing System Chemistry-Climate Model (GEOSCCM). This

model couples the GEOS general circulation model (GCM) to a full atmospheric chemistry module (Nielsen et al., 2017) and has been modified to distinguish a stratospheric $N_2O$ tracer from tropospheric tracers of fresh surface emissions (*Liang et al.*, 2022). GEOSCCM is shown to indicate a profound and dominant influence of the stratosphere on the tropospheric $N_2O$ seasonal cycle, with similar patterns to those observed in aircraft data. A correlation analysis is performed on the surface $N_2O$ AGR observed by NOAA and two indices of stratospheric variability as well as the El Niño Southern Oscillation (ENSO)

index. A similar correlation analysis is performed with output from GEOSCCM, which simulates its own QBO. The correlation analyses support the importance of the stratospheric influence on interannual variability in surface $N_2O$ but also suggest a role for ENSO-driven variability in the SH.

## 2 Methods

### 2.1 GEOSCCM with tagged stratospheric tracers

The GEOS-5 chemistry climate model (GEOSCCM) was used to simulate atmospheric $N_2O$ with geographically resolved surface emissions from soil, ocean and anthropogenic sources, and full stratospheric chemistry with stratospheric $N_2O$ destruction due to photolysis and $O(^1D)$ oxidation (*Nielsen et al.,* 2017). The model was run at 1°x1° resolution with 72 vertical layers from the surface to 0.01 hPa. In addition to the standard chemistry mechanism, four $N_2O$ tracers were included to track: 1) aged air from the stratosphere ($N2O_{ST}$), and 2) soil, 3) ocean, and 4) anthropogenic sources freshly emitted in the

troposphere following the same approach as in *Liang et al.* (2008) for chlorofluorocarbons (CFCs). $N2O_{ST}$ was used to provide a model estimate of the stratospheric influence on tropospheric $N_2O$ mean seasonal cycles, while a total $N_2O$ tracer was defined to include the influences of both stratospheric destruction and surface sources. The full GEOSCCM simulation spanned 2000-2019 (*Liang et al.*, 2022). The climatological seasonal cycle was analyzed based on the last 5 years of the simulation. The full 20-year simulation was used in the correlation analysis between model surface $N_2O$ anomalies and QBO and polar lower



stratospheric temperature (PLST). GEOSCCM temperature and QBO are both internally generated by the GEOS GCM and do not necessarily correspond to observations. However, they were computed in the same way as the observed indices, as described below in Section 2.4.1 and 2.4.2, respectively.

## 2.2 N$_2$O Data

### 2.2.1 Surface N$_2$O from NOAA long-term monitoring sites

Surface atmospheric N$_2$O data from the late 1990s onward were obtained from the NOAA Global Monitoring Laboratory (GML) for comparison to GEOSCCM output. NOAA has two programs that measure N$_2$O, the Halocarbons and other Atmospheric Trace Species (NOAA/HATS) (*Thompson et al.*, 2004) and the Carbon Cycle Greenhouse Gases group (NOAA/CCGG). NOAA/HATS provides *in situ* data measured every ~ 60 minutes using the Chromatograph for Atmospheric Trace Species (CATS) instruments at 5 baseline sites. NOAA/CCGG maintains a flask-air sampling network at ~55 widely distributed surface sampling sites, in which duplicate samples are collected about weekly and shipped to Boulder, Colorado for analysis by gas chromatography (GC) with electron capture detection and by a Tunable Infrared Laser Direct Absorption Spectroscopy (TILDAS) after August, 2019. The instruments are calibrated with a suite of standards on the WMO X2006A scale maintained by NOAA GML (*Hall et al.*, 2007). Uncertainties of the measurements (68% confidence interval) range from 0.26 to 0.43 ppb with GC-ECD and 0.16 ppb with TILDAS. The mean uncertainties in the CATS GC data are 0.2 to 1.2 ppb (68% confidence interval) over most of the 2000s, with an increase in recent years as the instruments near their lifetime.

This study used the NOAA combined HATS/CCGG N$_2$O product from 1998-2020 (https://doi.org/10.15138/GMZ7-2Q16), which is based on monthly medians from the CATS *in situ* program (at 5 sites) and monthly means from the CCGG flask program (at 13 background sites). The combined monthly data are first aggregated at the measurement program level for each sampling location. If both HATS and CCGG measure at a location, a weighted mean is calculated based on the programs' monthly uncertainties. In addition to the individual sites, global and hemispheric means are estimated from the latitude-binned and mass-weighted means of the combined monthly means for 12 background sites (*Hall et al.*, 2011).

### 2.2.2 NOAA Empirical Background for atmospheric N$_2$O

The NOAA empirical background (EBG) is a 4-dimensional (4-D) field, constructed from NOAA surface and aircraft N$_2$O data, which is used in North American regional inversions to represent the background concentration of atmospheric N$_2$O prior to the influence of continental surface fluxes (*Nevison et al.*, 2018). The EBG is defined daily over North America from 500-7500 m every 1000 m, from 170°-50°W every 10° longitude and from 20-70°N every 5° latitude (or, prior to 2017, from 20-80°N every 10° latitude). To construct the 4-D field, NOAA data are categorized as marine boundary layer, free troposphere



or continental boundary layer, depending on the location of each sample. These three categories are treated individually as

follows: For the marine boundary layer, time- and latitude-dependent reference surfaces are computed separately for the Pacific

and Atlantic (*Masarie and Tans*, 1995, updated as described in *Lan et al.*, 2023). For the free-troposphere, reference surfaces

are created using a similar approach, with an additional "domain-filling" step informed by backward and forward trajectories

for each aircraft sample collected above 3000 magl. For the continental boundary layer, $N_2O$ data are detrended by subtracting

the latitude and time dependent marine boundary layer reference values, where the transition from Pacific to Atlantic is

represented by linear interpolation as a function of longitude across the continent. Then, a multi-year mean seasonal cycle is

computed as a function of latitude, longitude and day of year using local Kriging following (*Hammerling et al.,* 2012).

**2.2.3 QCLS atmospheric $N_2O$ data from vertical profiling campaigns**

Atmospheric $N_2O$ measurements have been made *in situ* with the Harvard/Aerodyne Quantum Cascade Laser Spectrometer

(QCLS) on a variety of aircraft campaigns designed to study the atmospheric profiles of greenhouse and related gases (*Wofsy*

*et al.*, 2011; *Stephens et al.*, 2018). QCLS $N_2O$ data are retrieved at 1-Hz with 1s precision of 0.09 ppb and reproducibility

with respect to the WMO $N_2O$ scale of 0.2 ppb (*Kort et al., 2011; Santoni et al., 2014*) on the NOAA-2006 scale (*Hall et al.*,

2007). The first of the vertical profiling campaigns used here, the HIAPER Pole to Pole Observations (HIPPO) project,

consisted of 5 roughly month-long sets of flights centered over the central Pacific Ocean extending from the surface to the

upper troposphere/lower stratosphere and nearly pole to pole. These flights were timed between January 2009 and November

2011 to create a climatological seasonal cycle (*Wofsy et al.*, 2011). The second campaign (ORCAS), took place in January-

February 2016 and focused specifically on the Southern Ocean south of ~35°S (*Stephens et al.*, 2018). Most recently,

the Atmospheric Tomography Mission (ATom) campaign extended nearly pole to pole over both the Pacific and Atlantic

Oceans. ATom consisted of four ~month-long sets of flights over 3 years, timed to create a climatological seasonal cycle

150    (*Thompson et al.*, 2022). QCLS $N_2O$ was measured during ATom deployments 2-4 in January/February 2017,

September/October, 2017 and April/May 2018, respectively (*Gonzalez et al.*, 2021). (Note: technical issues interfered with

the $N_2O$ measurements on the ATom-1 deployment in July/August 2016).

**2.3 Correlation analysis for surface $N_2O$**

**2.3.1 Interannual variability in the atmospheric growth rate**

155    Interannual variability in the atmospheric growth rate of surface $N_2O$ at NOAA surface monitoring sites was calculated by first

removing the seasonal cycle from the monthly mean time series by computing a 12-month running average,

$$X_i = (C_{i-6} + 2\sum_{k=i-5}^{i+5} C_k + C_{i+6})/24, \qquad (1)$$



where *C* is the original monthly mean time series and *X* is the deseasonalized time series. The slope of the deseasonalized time series then was computed as a central difference,

$$S_i = 12 \, \frac{x_{i+1} - x_{i-1}}{2}, \tag{2}$$

where S is the centrally differenced slope and the scalar 12 converts *S* from units of ppb/month to ppb/yr. To account for the increasing growth rate of atmospheric $N_2O$ observed over the 21[st] Century (*Liang et al.*, 2022), the absolute slopes *S* were converted to atmospheric growth rate anomalies by removing an optimal (increasing) linear fit determined by recursive least squares regression. The atmospheric growth rate (AGR) anomalies constituted a monthly-resolved time series, which was plotted against various proxies and indices for both stratospheric influences and ENSO, as described below. Least squares linear regression correlation coefficients and p-values were computed with the assumption that a p-value < 0.05 was statistically significant at the 95% confidence level.

**2.3.2 Interannual variability in the magnitude of the seasonal $N_2O$ minimum**

To calculate interannual anomalies in the magnitude of the seasonal minimum, the raw monthly mean $N_2O$ data were detrended with a 3[rd]-order polynomial and a climatological seasonal cycle was constructed by taking the average of the detrended data for all Januaries, Februaries, etc. This climatological annual cycle was subtracted from the original raw data to produce a deseasonalized (but not detrended) time series. A running 12-month annual mean of this curve was then computed as in Equation 1, but where *C* is now the deseasonalized time series rather than the original monthly mean time series. At stations with gaps in the monthly data, the original 3[rd] order polynomial fit was used as a placeholder in the running mean. The running mean was subtracted from the deseasonalized curve to remove the secular trend and other low frequency variability, thus isolating the residual high frequency anomalies.

The high frequency residuals were sorted by month and selected months were plotted against the PLST BDC proxy described in Section 2.4.1. The months selected were those surrounding the seasonal $N_2O$ minimum, which is the most distinct feature of the seasonal cycle at remote baseline NOAA sites and which were hypothesized, based on previous work, as most likely to be influenced by the descent of $N_2O$-depleted air from the stratosphere (*Nevison et al.*, 2011). (Note: strong local sources can create large seasonal signals in atmospheric $N_2O$ that dominate the stratospheric influence at some sites, e.g., those influenced by agriculture or coastal upwelling (*Lueker et al.*, 2003; *Nevison et al.*, 2018; *Ganesan et al.*, 2020)).

The monthly $N_2O$ anomaly analysis was applied only to PLST BDC proxy and not to the QBO or ENSO indices, because the latter are monthly indices for which it is not straightforward to choose a representative month to correlate to the $N_2O$ anomaly,



given that the anomaly might result from the cumulative effect over multiple months. PLST in contrast has one unique value
each year that can be plotted against that year's N$_2$O anomaly for any given month.

## 2.4 Proxies and indices for the correlation analysis

### 2.4.1 Polar lower stratospheric temperature as proxy for the Brewer Dobson circulation

Mean polar (60°-90°) lower stratospheric temperature at 100 hPa in winter/spring (January-March in the NH and spring
(September-November) in the SH was computed from MERRA-2 reanalyses (*Gelaro et al.,* 2017). PLST reflects the
cumulative effect of fall/winter stratospheric downwelling. The mean PLST in each hemisphere was treated as a proxy for the
integrated strength of the BDC, which brings warm N$_2$O-depleted air from the middle to upper tropical stratosphere into the
polar winter lower stratosphere, with warmer PLST corresponding to stronger downwelling (*Nevison et al.*, 2007; 2011).
Winter months were averaged in the NH and spring months in the SH to account for the later seasonal breakup of the Antarctic
polar vortex compared to the Arctic polar vortex (*Nevison et al.*, 2011). For the monthly analysis, the PLST proxy was
regressed against the monthly N$_2$O anomaly in each of the subsequent months leading up to and encompassing the seasonal
minimum in atmospheric N$_2$O, which occurs in summer in the NH and autumn in the SH. For the AGR analysis, the mean
N$_2$O AGR anomaly was averaged over 12 months (considering a range of start/end months) for regression against PLST.

### 2.4.2 Quasi-Biennial Oscillation (QBO)

The QBO is a tropical, lower stratospheric, downward-propagating zonal wind variation with an average period of ~28 months
that dominates the variability of tropical lower stratospheric meteorology (*Baldwin et al.*, 2001; *Butchart*, 2014). The QBO
was quantified using monthly mean stratospheric zonal wind values in m/s derived from twice daily balloon radiosondes
conducted by the Meteorological Service Singapore Upper Air Observatory at a station located at 1.34°N, 103.89°E
([https://acd-ext.gsfc.nasa.gov/Data_services/met/qbo/QBO_Singapore_Uvals_GSFC.txt](https://acd-ext.gsfc.nasa.gov/Data_services/met/qbo/QBO_Singapore_Uvals_GSFC.txt)). A positive QBO indicates westerly
winds and a negative QBO indicates easterly winds. A range of altitudes from 10 mb to 100 mb was considered. Since the
QBO index is a monthly mean time series, it can be compared directly to the monthly mean N$_2$O AGR time series. However,
delays are expected between the QBO and its influence on tropospheric N$_2$O (*Strahan et al.*, 2015; *Ray et al.*, 2020). Therefore,
a range of lag times was considered spanning 6-24 months when correlating with the N$_2$O AGR anomalies to identify the
optimal QBO altitude and lag in each hemisphere.

### 2.4.3 ENSO

The El Niño Southern Oscillation (ENSO) index refers to the oscillation between warm (El Niño) and cold (La Niña) phases
in the eastern tropical Pacific Ocean. El Niño is a periodic warming and deepening of the thermocline in the eastern tropical
Pacific associated with westerly wind anomalies that excite eastward propagating downwelling equatorial Kelvin waves





(*McPhadden et al.*, 1998). The Niño 3.4 index, which is based on sea surface temperature anomalies from 5°S to 5°N and 170° to 120°W, defines an El Niño event as a temperature anomaly of > 0.4 degrees C.  Conversely, the index defines a

temperature anomaly of < -0.4°C defines as a La Niña event.  Monthly Niño 3.4 indices were obtained from https://www.cpc.ncep.noaa.gov/data/indices/sstoi.indices. Like the QBO index, Niño 3.4 is a monthly time series that can be compared directly to the monthly mean $N_2O$ AGR time series.  In the comparison presented here, a range of lag times in the Niño 3.4 index was considered spanning 0-12 months to identify the optimal lag in each hemisphere.

### 2.5 GEOSCCM correlation analysis

Equations 1 and 2 were applied to GEOSCCM $N_2O$ output sampled at the coordinates of NOAA monitoring sites to create modeled $N_2O$ AGR time series and monthly anomalies, using both total $N_2O$ and $N_2O_{ST}$.  Similarly, mean winter and spring PLST at 100 hPa was calculated for GEOSCCSM output in the NH and SH, as described in Section 2.4.1, for each model year from 2000-2019.  Finally, a GEOSCCM monthly QBO index was calculated at a range of altitudes from 10 mb to 100 mb by averaging the model zonal wind component in m/s between 5°S and 5°N over each of the 240 months from 2000-

2019.  A correlation analysis was performed using the GEOSCCM $N_2O$ AGR and monthly anomaly time series regressed against GEOSCCM PLST and QBO, similar to that described for the observed quantities in Sections 2.3-2.4.  (Note: the ENSO correlation analysis was not applied to GEOSCCM output because the model did not attempt to reproduce the impact of ENSO on surface flux variability (*Liang et al.*, 2022).)

### 3. Results

**3.1 Stratospheric influence on tropospheric $N_2O$ in model and aircraft data**

GEOSCCM simulates a strong stratospheric influence on the surface $N_2O$ seasonal cycle in both hemispheres, in which air depleted in $N_2O$ accumulates during winter in the polar lower stratosphere and crosses the tropopause in springtime (early summer) in the NH (SH) (Figure 1). The $N_2O$ depleted air moves downward and mixes equatorward from January to May at SH mid-to-high latitudes and from March to August at NH mid-to-high latitudes.  Due to these lags in downward propagation

and mixing, the surface minimum in the lower troposphere is not felt until autumn in both hemispheres (Figure 2).



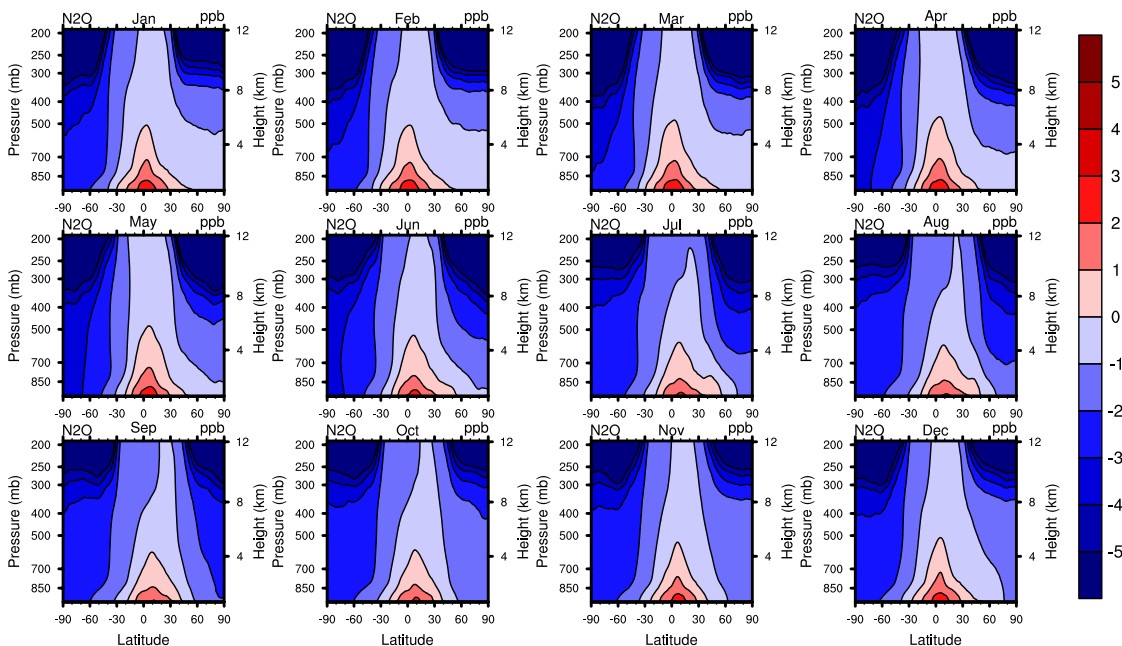

**Figure 1: GEOSCCM N$_2$O anomalies plotted in a monthly sequence of latitude vs. altitude plots extending from the surface up to 30 hPa (about 24 km). The GEOSCCM N$_2$O fields are detrended based on a deseasonalized fit to the model time series sampled at Mauna Loa and the mean value at Mauna Loa is removed to create the anomalies.**

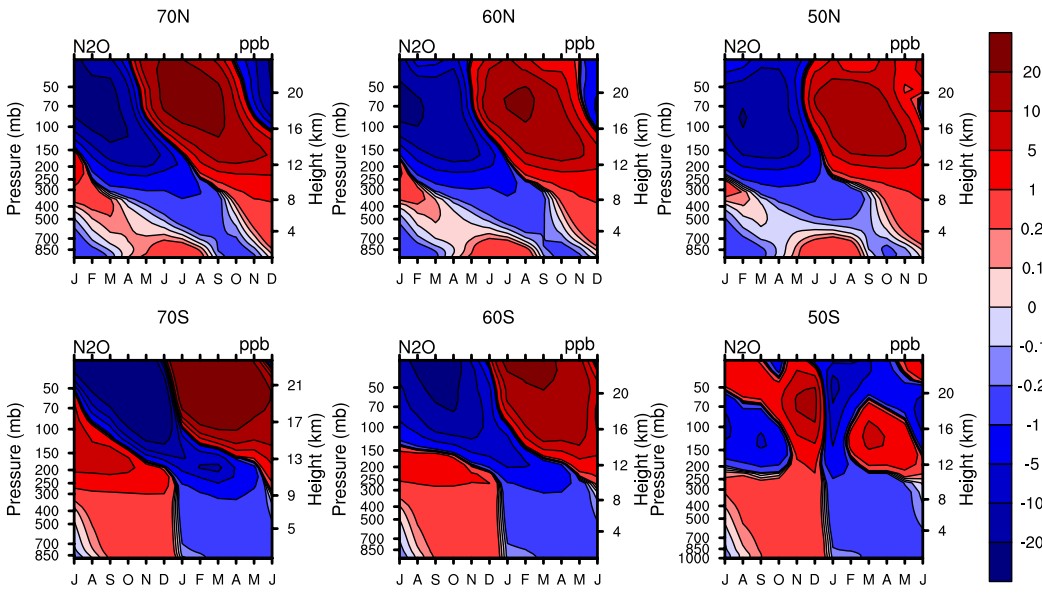


**Figure 2: GEOSCCM N$_2$O anomalies vs. month over a mean seasonal cycle, plotted from the surface to 30 hPa in the northern (top row) and southern (bottom row) hemispheres. The GEOSCCM N$_2$O fields are detrended based on the model time series sampled at the latitude, longitude and altitude of Mauna Loa. Zonal averages are computed at 70°, 60° and 50° latitude bins in each**



hemisphere and pressure level.  Monthly anomalies are then computed by subtracting the annual mean value at each pressure level.
**The SH panels are plotted with a 6 month shift to facilitate comparison of the seasonal phasing relative to the NH.**

The GEOSCCM seasonal cycle in tropospheric $N_2O$ is dominated by stratospheric loss that is transported to the surface, rather than by the influence of emissions from soil, ocean and anthropogenic sources, although the surface emissions tend to pull the total $N_2O$ seasonal minimum about 1 month earlier than the $N2O_{ST}$ minimum (Figure 3).  Comparison to data at long-term
monitoring sites suggests that GEOSCCM captures the mean seasonal cycle in $N_2O$ relatively well in the Southern Hemisphere but overestimates the amplitude of the cycle at high northern latitudes, with a ~1-2 month delay in phasing relative to observations (Figure 3).

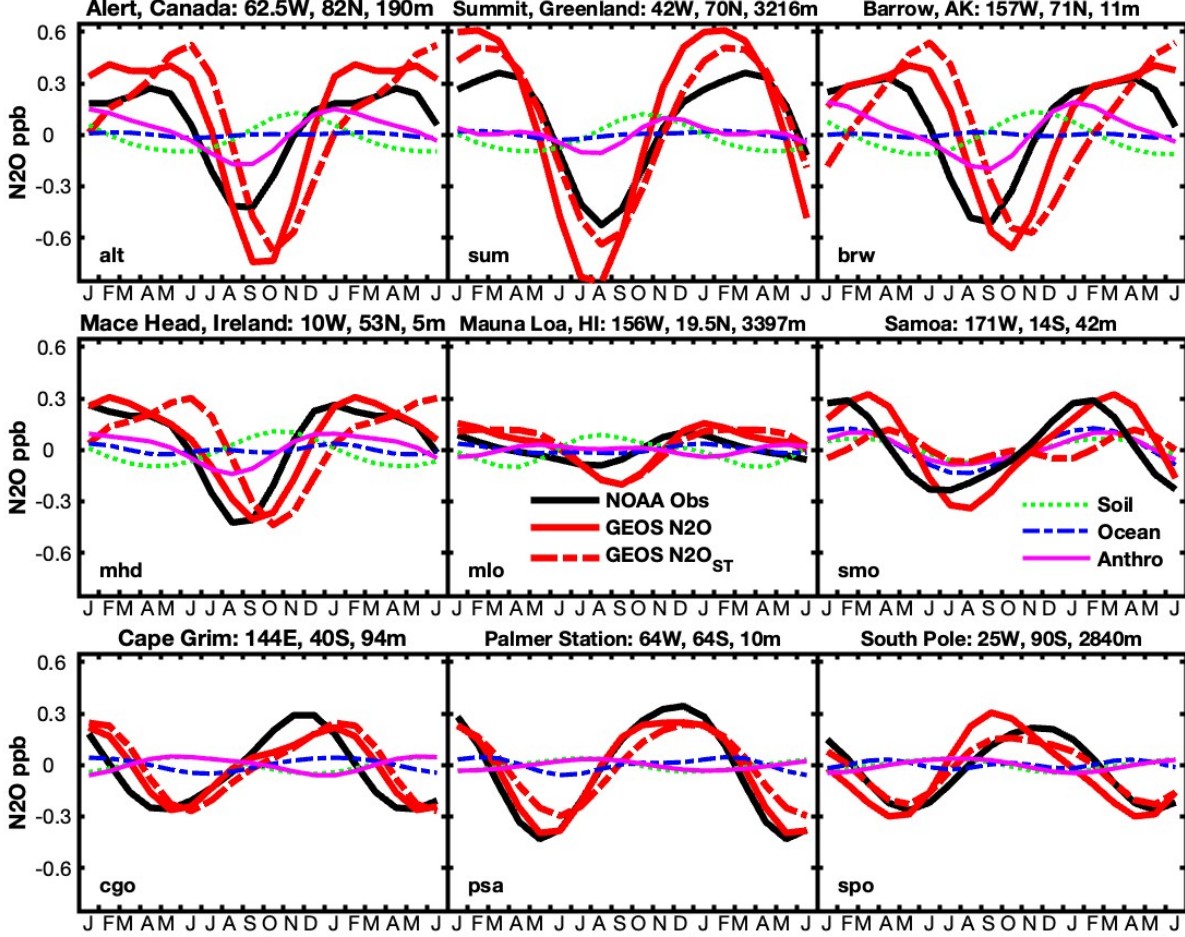

**Figure 3: Detrended seasonal cycles in atmospheric $N_2O$ modeled by GEOSCCM and compared to NOAA surface station data at 9 surface sites.  The red line is total $N_2O$ from all forcings, while the dashed red line is the tagged stratospheric tracer $N2O_{ST}$.  The black heavy line is observed $N_2O$.  Surface anomalies are shown for natural soil (green), ocean (blue), and anthropogenic, including agriculture, industry and biomass burning (magenta) surface emission sources.**



The NOAA $N_2O$ empirical background has similar features to those simulated by GEOSCCM. When viewed as a 12-month
sequence of NH altitude vs. latitude contours, extending up to 8 km, the NOAA data indicate that the North American
background signal of stratospheric depletion originates at polar latitudes in the upper troposphere in spring and is felt in the
midlatitude lower troposphere by July, with a peak influence around August (Figure 4). The effect on the troposphere is
strongest near the pole and weakens substantially moving equatorward. A comparison of altitude vs. month contours for
GEOSCCM and NOAA suggests a faster, more direct propagation of the stratospheric signal down to the surface in the NOAA
data compared to the model (Figure 5). As a result, the phasing of the GEOSCCM surface minimum is delayed ~1-2 months
relative to the NOAA empirical background, consistent with the comparison to NOAA surface monitoring data in Figure 3.

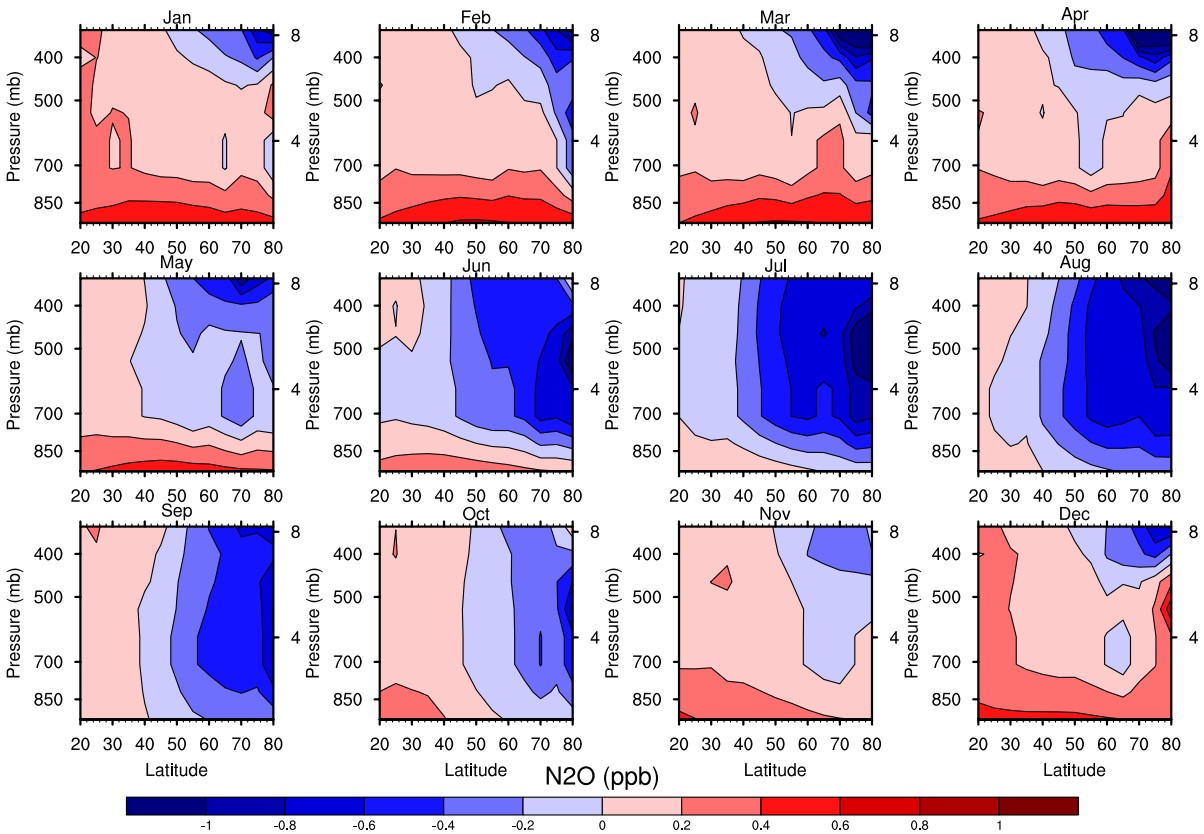

**Figure 4: Northern hemisphere $N_2O$ anomalies from the NOAA empirical background based on NOAA regularly sampled aircraft
flights, plotted in a monthly sequence of altitude vs. latitude plots extending up to ~8 km (330 hPa) and from 20° to 80°N, zonally
averaged over 160 to 60°W. The NOAA $N_2O$ fields are detrended based on a deseasonalized fit to the observed time series sampled
at the latitude, longitude and altitude of Mauna Loa.**



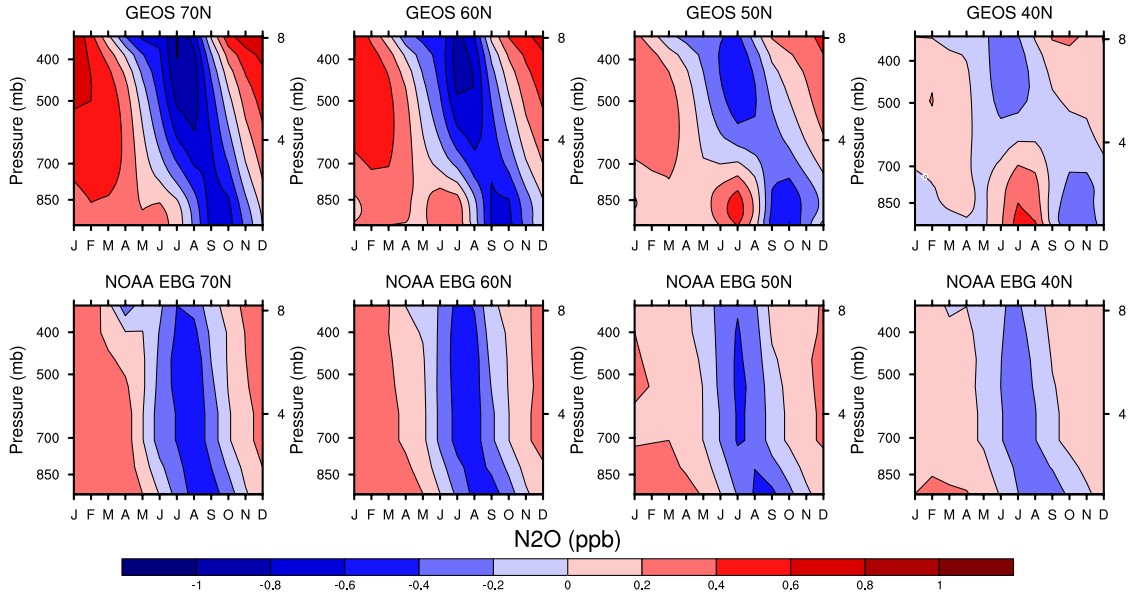

**Figure 5: Northern hemisphere N₂O anomalies vs. month over a mean seasonal cycle, plotted over the height of the troposphere up to ~8 km (330 hPa), comparing GEOSCCM (top row) and the NOAA empirical background (bottom row) at 70°, 60°, 50° and 40°N zonally averaged 10° latitude bins. The GEOSCCM and NOAA N₂O fields are detrended based on their respective time series sampled at Mauna Loa for model and observations, respectively. Monthly anomalies are computed by subtracting the annual mean value at each pressure level.**

HIPPO aircraft data extend up to 14 km and thus provide a broader perspective with respect to altitude of the stratospheric influence on tropospheric N₂O, but with sparser temporal coverage than the NOAA empirical background. The southbound transects from the five HIPPO deployments, when detrended and arranged chronologically over an annual mean cycle, form a sequence that is most readily seen in the NH (Figure 6). N₂O-depleted air accumulates in the polar lower stratosphere in January and crosses the tropopause by March/April. By June it has descended into the troposphere and moved equatorward, reaching its maximum influence at the surface in August. By October/November, the stratospheric signal is no longer visible at the surface following tropospheric mixing and dilution. Notably, this seasonal progression is less apparent although still discernible in a fuller dataset that also includes the ATom and northbound HIPPO transects (Supplementary Figure 1).





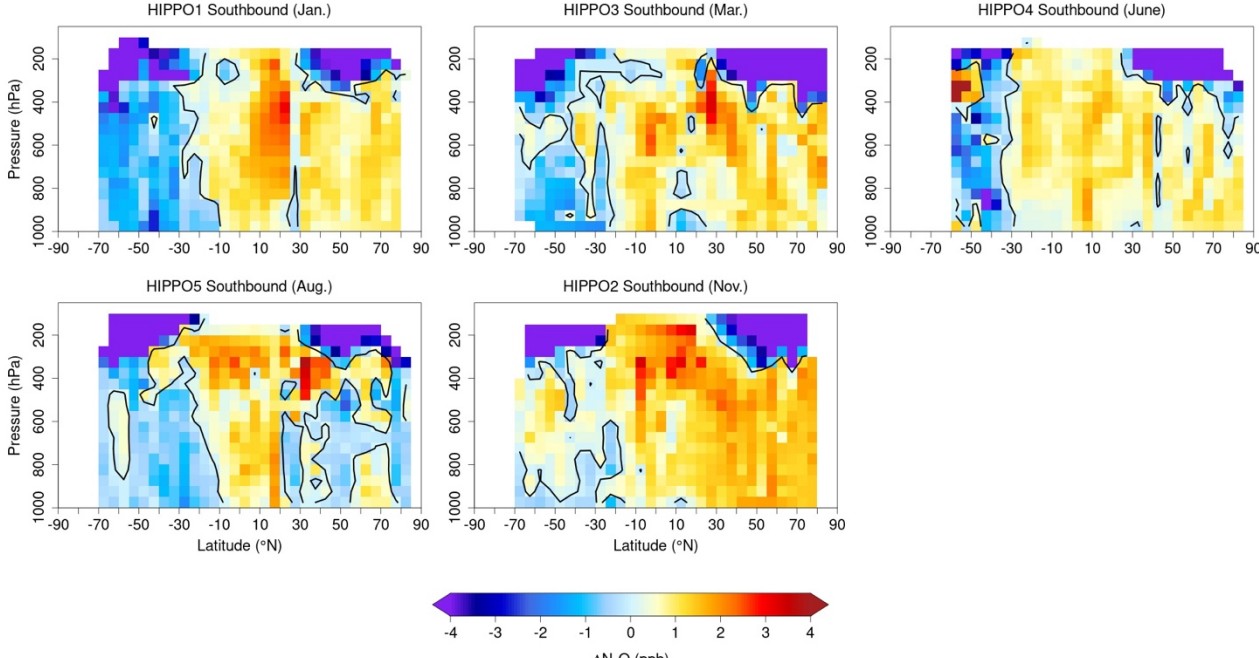

**Figure 6: Sequence of five southbound HIPPO transects arranged to form an annual sequence. Flight track data were interpolated onto a 5 degree latitude by 50 hPa grid using the akima package in R (Akima, 1978). A deseasonalized fit to the NOAA time series at Mauna Loa has been subtracted from all data, since the HIPPO deployments spanned several years over a period when atmospheric $N_2O$ was increasing by about 0.9 ppb/yr. HIPPO data extend up to 12-14 km and provide a fuller perspective with respect to altitude than the NOAA data in Figures 4-5 of the stratospheric influence on tropospheric $N_2O$.**

## 3.2 Interannual variability in the seasonal $N_2O$ minimum

In the SH, polar lower stratospheric temperature (PLST) from the previous spring is significantly negatively correlated to NOAA surface station $N_2O$ monthly anomalies in austral summertime (January and February), when $N_2O$ is descending into its autumn seasonal minimum. This correlation is observed at several extratropical southern NOAA sites including Cape Grim, Tasmania (CGO), Palmer Station, Antarctica (PSA) and South Pole (SPO) (Figure 7). The sign of the correlation is such that more negative surface $N_2O$ anomalies occur during warm years, in which stronger than average descent of air depleted in $N_2O$ occurs into the polar lower stratosphere over the austral winter and spring. Similar correlations are observed between GEOSCCM PLST and austral summer $N_2O$ anomalies at these sites, although they are strongest in February and March, i.e., delayed by about 1 month relative to NOAA surface observations (Figure 7). In support of the observed surface correlations, altitude-latitude contour plots of QCLS aircraft data suggest more depleted $N_2O$ values in the extratropical SH in February 2017 during ATom compared to ORCAS in February 2016, with the full global span of data in ATom-2 indicating a stratospheric influence originating from the southern polar region (Figure 8). The ORCAS aircraft campaign took place after





a particularly cold lower stratosphere Antarctic spring (weak Brewer Dobson circulation) while the ATom-2 deployment took place after a relatively warm spring (strong Brewer Dobson circulation).

In the NH in contrast, PLST from the previous winter is not correlated significantly to $N_2O$ monthly anomalies at extratropical NOAA surface sites, either in August, the month of the seasonal minimum, nor in June and July, when $N_2O$ is descending into its seasonal minimum.  GEOSCCM also does not predict significant correlations between PLST and summer $N_2O$ anomalies at most northern NOAA sites, with the exception of Mace Head, Ireland (MHD), where a negative correlation is found in July at MHD for both $N_2O$ (R=-0.72) and $N_2O_{ST}$ (R=-0.78) (Supplementary Figure S2).


**Figure 7:  a) South Pole mean seasonal cycle in $N_2O$ for observed $N_2O$ and GEOSCCM total $N_2O$ and $N_2O_{ST}$. NOAA surface $N_2O$ seasonal anomalies in b) January and c) February at South Pole spanning 1997-2020, plotted vs. mean lower stratospheric MERRA-2 temperature at 100 hPa averaged over 60-90°S over the previous spring (September-November).  The labeled anomalies in 2016**
**and 2017 correspond to the year of the ORCAS and ATom-2 aircraft campaigns, respectively. Bottom row shows GEOSCCM seasonal anomalies at South Pole spanning 2000-2019 for d) total $N_2O$ in February and $N_2O_{ST}$ in e) February and f) March plotted**




vs. mean GEOSCCM lower stratospheric temperature at 100 hPa averaged over 60-90°S over the previous spring. The correlations between surface $N_2O$ and stratospheric temperature are strongest for NOAA $N_2O$ in January and February (austral summer), and for GEOSCCM $N_2O$ in February and March, when $N_2O$ is descending into its seasonal minimum.


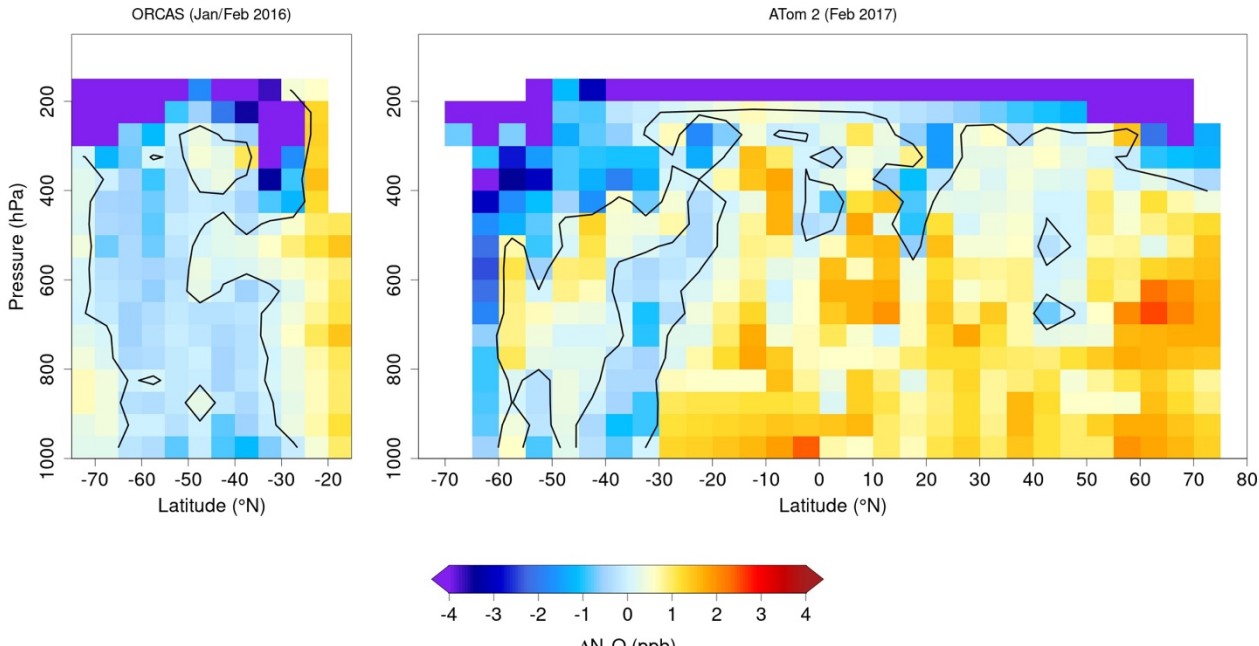

**Figure 8.** The 2 panels compare ORCAS (Jan.-Feb. 2016) and ATom-2 (Jan.-Feb. 2017) $N_2O$ in ppb as a function of altitude and latitude, with interpolation and deseasonalizing the same as Figure 6. The ATom plot uses only the southbound portion of ATom-2. The comparison supports a stronger stratospheric influence during ATom-2 (a year of strong Brewer-Dobson circulation) than during ORCAS (a year of weak Brewer-Dobson circulation), as indicated in Figure 7b,c. The right panel shows ATom-2 data over the full 65°S to 75°N latitude span, putting the stratospheric influence coming from the southern polar stratosphere into broader perspective.

### 3.3 Interannual Variability in the Atmospheric Growth Rate (AGR) of Surface $N_2O$

The QBO index is positively correlated to the NOAA surface $N_2O$ AGR in the SH, with an optimal correlation (R = 0.42) for

QBO in the upper stratosphere at 20 hPa with a time shift of about 19 (17-21) months relative to the $N_2O$ time series (Figure

9a). The correlation between GEOSCCM QBO and $N_2O$ AGR in the SH is weak (R = 0.24) but also positive in sign in the

upper stratosphere with a similar optimal shift in the GEOSCCM QBO of about 19 months (Figure 9c).







**Figure 9: (a,c)** Southern hemisphere N₂O atmospheric growth rate (AGR) for NOAA (a) and GEOSCCM (c) plotted with the QBO
index at 20 hPa with a 17-21 month forward shift in the index. **(b,d)** SH N₂O AGR plotted with mean lower stratospheric
temperature averaged over 60-90°S for September-November in the year prior to the annual label on the X axis. The AGR is
averaged from monthly N₂O data over the ensuing 12 month period May-April (solid red line), shifted plus or minus 1 month (dotted
red lines), for NOAA (b) and GEOSCCM (d). Note: to convert to %/yr (AGR units often used in the literature) ppb/yr can be
multiplied by 100/323 (~1/3), where 323 is the mean tropospheric mixing ratio of N₂O over 1999-2020.

In contrast to the SH, the NOAA surface N₂O AGR in the NH is negative in sign and not significantly correlated to the QBO

index at any altitude. The strongest of the weak correlations in the NH occurs for 50 hPa QBO (R=-0.23) with a 10-14 months

lag (Figure 10a). GEOSCCM predicts a significant negative correlation (R = -0.47) between the GEOSCCM QBO and the

NH N₂O AGR, which also is optimal around 50 hPa with 10-14 month QBO lag (Figure 10c).







**Figure 10: (a,c) Northern hemisphere N₂O atmospheric growth rate (AGR) for NOAA (a) and GEOS (c) plotted with the QBO index at 50 hPa with a 10-14 month forward shift in the index. (b,d) NH N₂O AGR plotted with mean lower stratospheric temperature averaged over 60-90°N over January-March of the year labeled on each data point. The AGR is averaged from monthly N₂O data over the encompassing 12 month period July-June (solid red line), shifted plus or minus 1 month (dotted red lines), for NOAA (b) and GEOSCCM (d).**

In the SH, PLST is not significantly correlated to the surface N₂O AGR observed by NOAA, but within GEOSCCM the two are negatively correlated (Figure 9b,d). The correlation with PLST in GEOSCCM occurs for the N₂O AGR averaged over a wide range of 12-month intervals extending from November-October, overlapping with and proceeding the September-November PLST average, through the following August-July, with the strongest correlation (R = -0.61) over May-April.

In the NH, winter PLST is negatively correlated to the NOAA surface N₂O AGR (R = -0.61) with an optimal correlation for the 12 month period of July-June encompassing the January-March PLST average (Figure 10b). A similar correlation is found



between the GEOSCCM PLST and the N$_2$O AGR in the NH (Figure 10d).

The Niño 3.4 index is negatively correlated (R = -0.5) to the NOAA surface N$_2$O AGR both globally and in the SH, with little to no monthly lag in the index. In the NH, the correlation is weaker (R=-0.35) with an optimal lag of 7 months in the Niño 3.4 index relative to the NOAA N$_2$O AGR (Figure 11).




Figure 11: NOAA N$_2$O AGR plotted with the Nino3.4 index for a) SH and global mean AGR with a 0-2 month shift in the index and b) NH AGR with a 6-8 month shift in the index.

## 4. Discussion

Both the model results and the NOAA surface station and QCLS aircraft observations presented here suggest that the stratosphere helps drive the seasonal minimum in tropospheric N$_2$O and also influences its atmospheric growth rate. Multiple lines of evidence point to this conclusion. First, vertical profiles of atmospheric N$_2$O from aircraft provide a new, big picture perspective, in which N$_2$O-depleted air originates in the winter polar lower stratosphere, crosses the tropopause in spring or



early summer around the time of vortex breakup, and descends downward and equatorward, reaching Earth's surface by
summer or early fall. Second, PLST and QBO indices correlate significantly to the surface $N_2O$ AGR in the NH and SH, respectively.  PLST also correlates with monthly anomalies at or near the time of the seasonal minimum in the SH.  These correlations are consistent with a stronger stratospheric influence in years with a stronger Brewer Dobson circulation and are similar to correlations found in previous studies (*Nevison et al.*, 2007; 2011). Finally, GEOSCCM simulations with an explicitly resolved stratospheric $N_2O$ tracer yield similar $N_2O$ AGR correlations with internally modeled QBO and PLST
indices, and show similar 3-dimensional patterns to those in the NOAA empirical background and in QCLS aircraft data, although with some differences in phasing and propagation time of the stratospheric signal to the surface.

**4.1 The Brewer Dobson circulation**

The mechanistic pathway by which the stratosphere imparts a distinct seasonal signature to surface $N_2O$ is linked to the Brewer Dobson circulation (BDC), which transports warm, $N_2O$-depleted air from the middle and upper stratosphere into the polar
lower stratosphere in the winter hemisphere (*Holton et al.*, 1995; *Liang et al.*, 2008; 2009; *Nevison et al.*, 2011; *Butchart*, 2014). This wintertime descent leads to a large seasonal amplitude in the polar lower stratosphere, in which the $N_2O$ mixing ratio reaches its minimum in spring just before the time of polar vortex break-up. $N_2O$-depleted air is brought into the troposphere by slow diabatic descent and mixing across the polar tropopause as well as entrainment due to the summertime increase in tropopause height. This air is then mixed between the mid and high latitudes via various synoptic-scale eddies in
extra-tropical cyclones (*Stohl et al.*, 2001). The cross-tropopause gradient between the spring polar lower stratosphere and the troposphere can be 50 ppb or more but stratospheric air is strongly diluted after it enters the troposphere. By the time the stratospheric depletion signal propagates down to the lower troposphere it is reduced by a factor of ~100, contributing to the < 1 ppb seasonal amplitude observed in $N_2O$ at surface sites (*Nevison et al.*, 2004; 2011; *Liang et al.*, 2009).

**4.2 The QBO and its relationship with the Brewer Dobson circulation**

The QBO is the primary mode of variability governing the amount of $N_2O$ that upwells from the tropical lower stratosphere into the middle and upper tropical stratosphere, the region of peak photochemical destruction (*Baldwin et al.*, 2001; *Prather et al.*, 2015; *Ruiz et al.,* 2021).  Photochemical destruction is highest when QBO winds at higher altitudes (~30 hPa and above) are in the westerly (positive) phase and lower altitude QBO winds are in the easterly (negative) phase.  This configuration is associated with increased vertical upwelling in the tropical lower stratosphere, which transports more $N_2O$ to its peak loss
region (~32 km) (*Strahan et al.*, 2021; *Ruiz et al.*, 2021).

In addition to the primary vertical circulation, the QBO has an associated secondary or meridional circulation, which in the SH involves transport of photochemically-depleted tropical air into the subtropical middle stratosphere followed by planetary wave-driven mixing, which homogenizes this air over a broad area known as the "surf zone" (*Strahan et al.*, 2015). The surf





zone extends from about 30 hPa to 10 hPa in altitude and 15-70º S in latitude.  Paradoxically, in the positive QBO phase in the upper stratosphere, when the $N_2O$ photochemical loss is at its peak, relatively less $N_2O$-depleted air enters the subtropical surf zone, due to the upward/clockwise flow of the secondary QBO circulation (*Strahan et al.*, 2015).

$N_2O$-depleted air in the surf zone (set by the QBO) subsequently is mixed into the polar region during the late spring breakup
of the Antarctic polar vortex and summer-to-fall SH vortex development. The $N_2O$ anomaly is then set into the Antarctic region as the polar vortex forms in the fall. The vortex-trapped $N_2O$ anomaly undergoes diabatic descent, driven by the BDC and in isolation from mixing with lower latitudes, through the fall and winter (*Rosenfield et al.,* 1994). After a few winter months of BDC-driven diabatic descent, the anomaly arrives in the Antarctic lower stratosphere in the July-September period, about 1 year after it formed in the middle tropical stratosphere (*Strahan et al.*, 2015). Continued descent and mixing across the polar
tropopause bring the $N_2O$ depleted air down to the surface ~4 months later, consistent with the long (17-19 month) delay between the QBO index at 20 hPa and the surface SH $N_2O$ AGR anomalies in NOAA surface station data found in our analysis (Figure 9).

The strong isolation of the Antarctic polar vortex prevents mixing with midlatitudes during the period of diabatic descent from
the altitude of the "surf zone" and is consistent with our finding that significant correlations with the NOAA surface $N_2O$ AGR in the SH occur mainly for higher altitude QBO indices between about 30 and 10 hPa. The positive sign of the $N_2O$ AGR correlation with the QBO index at those altitudes may be explained by the fact that the QBO meridional circulation brings relatively less $N_2O$ depleted air into the subtropical surf zone during the phase when the QBO is positive (westerly winds) in the upper tropical stratosphere above about 30 hPa (*Strahan et al.*, 2015).  This leads to a positive $N_2O$ anomaly (i.e.,
photochemically depleted, but less depleted than average) in the surf zone that ultimately is mixed into the polar region and transported into the troposphere via diabatic descent and mixing. When this signal of relatively low $N_2O$ depletion is felt at Earth's surface, it permits a more positive $N_2O$ AGR than normal, hence the positive correlation with the positive (westerly) QBO at 10-30 hPa that originally drove the stratospheric $N_2O$ anomaly well over a year prior.

The dynamics of the QBO, its interaction with the BDC and ultimate influence on surface $N_2O$ are complex, as described above. The QBO-associated photochemical destruction anomaly per se is not the main determinant of the stratospheric influence on surface $N_2O$, since one would otherwise expect a negative correlation between the upper altitude QBO and the surface $N_2O$ AGR instead of the positive correlation found in Figure 9.  *Ruiz et al.* (2021) similarly concluded that surface variability in $N_2O$ is not correlated directly to the QBO-driven variability in stratospheric loss, but rather by dynamical
variations in cross tropopause fluxes of air, which are governed at least in part by the BDC.

Like our study*, Ray et al.* (2020) found a positive correlation between the QBO index at 50 hPa and the NOAA surface $N_2O$ AGR in the SH (but not the NH).  Their QBO index was somewhat lower in altitude than our optimally selected altitude (20





hPa) and their optimal phase shift was somewhat less (8-12 months) than our optimal 19 month phase shift. This was likely a
result of the time lag in the downward propagation of the QBO winds. Our own correlation analysis across a range of altitudes
shows a positive correlation between QBO and the SH $N_2O$ AGR in which the correlation weakens and the optimal lag time
decreases with decreasing altitude (Supplementary Figure S3). At 50 hPa, we find an optimal lag time of 10-12 months (R =
0.33), consistent with *Ray et al.* (2020).

## 4.3 Northern vs. southern hemisphere differences

In the NH, some of the same mechanisms and interactions between the QBO and BDC occur, but they are more difficult to
isolate than in the SH due to the more complex atmospheric dynamics of the NH stratosphere. The deposition of momentum
from planetary scale Rossby waves propagating into the stratosphere is the fundamental driver of the BDC. *Holton and Tan*
(1980) originally showed that a deep and cold northern winter polar vortex was associated with the QBO westerly phase, and
a weaker and warmer vortex associated with the easterly phase. Hence, the year-to-year integrated strength of the BDC is tied
to the interaction of the NH mean flow with the QBO. Further, the BDC strength and structure is also tied to meridional mixing
of air. In addition to the QBO influence on the northern polar vortex, the QBO induces a meridional circulation that directly
impacts the northern mid-latitudes (e.g., *Randel and Wu*, 1996). The NH mid-winter QBO and wave mean-flow interaction
has two effects: 1) it modulates the strength and structure of the BDC, and 2) it also modifies mixing between the Arctic polar
vortex and the northern mid-latitudes by Rossby waves. These complex dynamics may explain why the $N_2O$ AGR at NH sites
correlates best with lower stratosphere QBO indices, why they have a negative rather than positive sign (due to the vertical
reversal of the sign of QBO with altitude https://acd-ext.gsfc.nasa.gov/Data_services/met/qbo/qbo.html) and why these
correlations are generally weaker than at southern sites.

In the NH, both NOAA surface stations and GEOSCCM show a significant negative correlation between PLST and the $N_2O$
AGR, which is consistent with a slower $N_2O$ growth rate during years with a stronger BDC. The planetary wave activity that
drives the BDC is strongest in the NH due to the more variable topography and stronger land-sea contrasts in the NH compared
to the SH. Thus the BDC-driven descent into the winter pole is more strongly seasonal in the NH than in the SH (*Holton et al.*, 1995). For both NOAA surface stations and GEOSCCM, the NH $N_2O$ AGR is more strongly correlated to PLST than it is
to the QBO, suggesting the stronger proximal influence of the BDC and stratosphere-troposphere exchange on the AGR,
consistent with *Ruiz et al.* (2021). The weak, statistically insignificant correlation between QBO and NOAA surface $N_2O$
AGR is consistent with the results of *Ray et al.* (2020) and the complex dynamical interactions between the BDC and the QBO
in the NH discussed above.

In contrast to the NH, the NOAA SH surface $N_2O$ AGR does not correlate to PLST but does correlate to QBO. This result is
somewhat puzzling, especially given the significant correlation between PLST and NOAA surface station $N_2O$ monthly





anomalies in January and February (Figure 7), which are corroborated by the ATom-2 and ORCAS data (Figure 8). It appears that the impact of the stratosphere in austral summer as tropospheric $N_2O$ descends into its seasonal minimum is not sufficient to influence the SH $N_2O$ AGR over a full 12-month period.

The SH $N_2O$ AGR results may reflect the strong preservation of the surf zone QBO signal that mixes into the polar region and is ultimately transported into the troposphere as per *Strahan et al.* (2015) combined with the relatively weaker BDC in the SH and/or the interference of ENSO-driven signals discussed below. The fact that PLST does correlate with the SH $N_2O$ AGR in GEOSCCM output suggests that GEOSCCM may overestimate the influence of the BDC in the SH or that the correlation may be cleaner due to the lack of a competing ENSO influence in GEOSCCM.

**4.4 Correlations with ENSO**

The NOAA surface station $N_2O$ AGR correlation with ENSO indices is similar in magnitude to the correlations with stratospheric indices in the SH (R = -0.49, 0-2 month phase shift) and relatively weaker in the NH (R=-0.35, 7 month optimal phase shift) (Figure 10). The correlation in the SH could in part reflect meteorological shifts in the tropical low level convergence pattern during positive ENSO (El Niño) conditions. For atmospheric gases with a positive north-south latitudinal

gradient, these shifts result in a lessened influence of winds from the NH on the tropical SH, e.g., at the NOAA Samoa site, and a heightened influence of southeasterly winds (*Prinn et al.*, 1992; *Nevison et al.*, 2007). The fact that the $N_2O$ AGR correlation with ENSO is considerably weaker in the NH than in the SH suggests a limited impact of ENSO on NH $N_2O$ and supports the hypothesis that reduced north-to-south transport during El Niño phases contributes to the correlation observed in the SH.


The negative correlation between $N_2O$ AGR and ENSO also may reflect a true reduction in the biogeochemical $N_2O$ source during the positive ENSO phase, for example, due to drought over tropical land or due to reduced upwelling in the tropical ocean (*Ishijima et al.*, 2009; *Thompson et al.*, 2013). The most well documented biogeochemical response of $N_2O$ to ENSO events occurs in the Eastern Tropical South Pacific (ETSP), a well known oxygen minimum zone (OMZ) and hot spot of

oceanic $N_2O$ emissions (*Arévalo-Martínez*, 2015; *Ji et al.*, 2019). El Niño conditions decrease upwelling in the ETSP, thereby reducing the surface productivity, deepening the oxycline, contracting the OMZ and decreasing the $N_2O$ sea-to-air flux (*Ji et al.*, 2019; *Babbin et al.*, 2015).

However, it is likely that less than one quarter of the total $N_2O$ budget comes from oceanic emissions, of which the ETSP is

only one component (*Yang et al.*, 2020; *Canadell et al.*, 2021). This raises questions about whether a reduced ETSP source has enough leverage to control the overall $N_2O$ AGR. *Ruiz et al.* (2021) removed the stratospheric influence from surface $N_2O$ data to tease out a source of ~ 1 Tg N (about 5% of the total annual $N_2O$ source) associated with the 2010 La Niña





event, which could have come from tropical land or ocean, or some combination of both. Similarly, *Kort et al.* (2011) found evidence of strong episodic pulses of ~ 1 Tg N from tropical regions, based on maxima in QCLS $N_2O$ data measured in the
middle and upper troposphere during aircraft campaigns in 2009. These pulses were not tied specifically to an ENSO event but rather more generally provided a testament to the strength of the tropical $N_2O$ source.

The 1 Tg N La Niña source inferred by Ruiz *et al.* (2021) raises the possibility that both ENSO and the stratosphere may jointly influence the $N_2O$ AGR, in a manner that may complicate single variable correlation analyses. Consistent with this hypothesis,
in our own study, a multivariate correlation of both QBO at 20 hPa with 19 month lag and the Niño 3.4 index with 0 months lag better captures the variability in the SH $N_2O$ AGR (R = 0.61) than either index alone (R = 0.42 and -0.49, respectively).

## 5 Conclusions

Global airborne surveys provide new insights into stratospheric influences on tropospheric $N_2O$ and advance our ability to understand surface variability in $N_2O$ sources. $N_2O$ observations from these surveys support GEOSCCM simulations in
showing that $N_2O$-depleted air accumulates throughout the winter in the polar lower stratosphere, crosses the polar tropopause in spring or early summer, and descends downward and equatorward, transmitting a coherent signal to Earth's surface in the summer to early-autumn period. In support of this view, significant correlations are found between the $N_2O$ AGR observed at long-term surface monitoring sites and either the QBO index in the SH or PLST in the NH, where PLST is a proxy for the strength of the BDC. Correlations between the $N_2O$ AGR and ENSO indices are also significant in the SH, suggesting a joint
influence of ENSO and the stratosphere on the AGR in that hemisphere. The QBO influences the rate at which $N_2O$ is delivered to and destroyed in the tropical middle to upper stratosphere, but complex atmospheric dynamics buffer how variations in the stratospheric $N_2O$ loss rate are transmitted across the tropopause to modulate the surface $N_2O$ AGR. Stratosphere-troposphere exchange in polar regions is linked closely to the BDC and appears to be a more direct influence than the QBO on the $N_2O$ AGR in the NH. In contrast, in the SH, the combination of a better-preserved QBO signal and weaker BDC may lead to a
direct (albeit with a ~1.5 year lag) correlation between the QBO and the SH $N_2O$ surface AGR, consistent with our understanding of stratospheric dynamics.

**Code Availability**

Codes are available from the corresponding author upon request.

**Data Availability**

NOAA $N_2O$ data can be obtained by contacting xin.lan@noaa.gov or through the NOAA Global Monitoring Laboratory at https://gml.noaa.gov/aftp/data/trace_gases/n2o/flask/. QCLS $N_2O$ data are openly available and archived in the Oak Ridge National Laboratory Distributed Active Archive Center (ORNL DAAC) https://doi.org/10.3334/ORNLDAAC/1925 (ATom),



and at the National Center for Atmospheric Research (NCAR) https://doi.org/10.5065/D6SB445X (ORCAS) and https://doi.org/10.3334/CDIAC/HIPPO_010 (HIPPO).

**Author contributions**

CDN designed and carried out the analysis and prepared the main manuscript and most of the figures. QL implemented separate stratospheric and tropospheric $N_2O$ tracers in GEOSCCM and provided model output. PN computed QBO indices and MERRA stratospheric temperatures and provided guidance on stratospheric dynamics. BBS, RC, YG and EK provided QCLS $N_2O$ data and BBS created contour plots of the QCLS data. XL and GD provided $N_2O$ surface data. All authors reviewed and approved the manuscript.

**Competing Interests**

The authors declare they have no conflicts of interest.

**Acknowledgments**

CDN and QL acknowledge support from the NASA MAPS program (award 16-MAP16-0049). The authors are grateful to Arlyn Andrews, Colm Sweeney, Bradley Hall, Ed Dlugokencky, Steve Wofsy, Bruce Daube, and many others who have made this study possible, through collection and analysis of surface station and NOAA aircraft flasks, in situ NOAA station measurements, and QCLS aircraft campaign observations. The HIPPO and ORCAS observations, and the contributions of BBS were supported by the National Center for Atmospheric Research, which is a major facility sponsored by the National Science Foundation under Cooperative Agreement No. 1852977.

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
