# Peer review of "Observational and model evidence for a prominent stratospheric influence on variability in tropospheric nitrous oxide"

_EGUsphere, 2023_

## Referee Comment (RC1)

**Review on egusphere-2023-2877**

*Northern vs southern hemisphere differences in the stratospheric influence on variability on tropospheric nitrous oxide* by Nevison et al.

Nevison et al. present a follow-up study of their publication in ACP from 2011. In their current study they present simulations with a GCM using a tagged stratospheric tracer of $N_2O$ to investigate the stratospheric influence on the seasonal cycle of tropospheric N2O. Additionally, they use the QBO and ENSO index and the polar lower stratospheric temperature as proxy for atmospheric transport and correlate these parameters with the $N_2O$ atmospheric growth rate. They find that the atmospheric growth rate anomaly in the southern hemisphere is better correlated to the QBO index, than to polar lower stratospheric temperature and that this hemispheric difference is consistent with the current knowledge of hemispheric difference in atmospheric dynamics.

This is a very interesting study and a valuable continuation of the previous work of Nevison et al. (2011). However, the current version of this manuscript is completely overloaded. There is too much analyses presented and a lot of necessary information missing, this is especially the case for the result section. You cannot expect the reader to read several papers additionally to your current manuscript to be able to follow the presentation of your current results. On the other hand, a very extensive discussion section is provided which rather confuses than helps to understand the dynamical processes behind.

Thus, I would suggest major revisions before publication in ACP based on the comments and corrections given below.

**General comment:**

- My major comment and suggestion for improvement would be to remove the entire part with the ENSO and QBO index and correlation of these to the atmospheric growth rate. I would suggest to instead focus on the experiment with the tagged stratospheric tracer which is a really great and convincing tool to study the stratospheric influence on the tropospheric seasonal cycle of $N_2O$. From the correlation analyses part you could make a second paper where you also use model experiments to better substantiate the dynamical processes behind. So far it has been done in a rather speculative way than providing real proof. With skipping that part you would still have 8 figures plus 2 from the supplement, thus 10 in total which would be a reasonable number.

- In many occasions your provide already statements on the results before you have shown the respective figure/analysis. This is not a good way of writing and causes only confusion. Thus, I would suggest that you first describe what is shown in the figures/what analyses has been done and then provide the results you gain with exactly stating what leads you to your specific result/conclusion.

- A lot of explanation/description of the figures/analyses is missing in the result section. It is really hard to follow what exactly has been done and why. This definitely needs to be improved. This of course will make your manuscript somewhat longer, but more valuable for the scientific community. Further, to have a not too long paper, we are back at my first comment to then skip the part with QBO and ENSO index, because the results of the tagged tracer experiment provide enough content for one paper.

- In the submission form you stated that your paper is a follow-up study of Nevison et al. (2011), but unfortunately, nowhere in the paper you state what is the new analyses/finding compared to your previous study.

- Figure captions should generally only describe what is shown in the figure not provide any statements on the results or method. This should be rather described in the respective sections. Thus, all figure captions need to be revised.

**Specific comments:**

P1, L20ff: The abstract needs to be revised. See ACP guidelines. You solely describe your results, but you do not provide any introduction to the topic (e.g. why is it important?). Further, the abstract should end with some sentences stating what the importance and implications of your results are.

P3, L82-83: This is a result and should not appear in the introduction.

P3, 87: In the introduction I would have expected more information on the tagging experiment you have done. What is tagging and for what is it used. Not every reader is familiar with this technique. So a short explanation is needed.

P4, L102: Add here some sentences stating how good the model is plus references. Are there evaluation studies that show the quality of the model simulations especially with respect to atmospheric dynamics? That many models have problems simulated a proper QBO is discussed e.g. in Khosrawi et al. (2013).

P5, L137: "local Kriging"? What to you mean with "Kriging"? I have never heard about this and I guess you should provide some more explanation. Further, the reference should appear without parenthesis (should be embedded in the text).

P5, L150: What is meant with "2-4"? Are these the campaign numbers? Please clarify.

P5, L152: What does this mean? Is thus data missing for this time/time period? Please clarify and rephrase sentence accordingly.

P8, L236: It would be better if you first would describe what is shown in the figure and what kind of analyses you have done before you make statements of what the result is. What exactly in this figures does point to your interpretation that there is a stratospheric influence?

P10, L251: Not stratospheric loss. I guess you mean here stratospheric air or air characterized by low N2O.

P10, L257: This deviation could have many reasons. Either the model has deficiencies in the sources and sinks of $N_2O$ or atmospheric transport. Not clear if stratospheric or tropospheric processes are the cause of the shift. It could be that $N_2O_{ST}$ is correct, but $N_2O$ not.

P10, Figure 1: You show 6 stations for the NH, but only 3 for SH. Why do you show less for the SH than for the NH. How have these stations been selected? What where the criteria for choosing these?

P10, Figure 1: This figure is overloaded. You do two analysis in one figure. First, you use observations to evaluate the $N_2O$ seasonal cycle from the observations. Then you use $N_2O_{ST}$ and the other curves to investigate the processes that affect the seasonal cycle. Thus, I would suggest to split the figure. One where you compare $N_2O_{obs}$ with $N_2O$ model and one where you show $N_2O_{ST}$, $N_2O_{soil}$, $N_2O_{ocean}$ and

$N_2O_{anthro}$. The complete figure showing all curves in one plot as done now, could be provided in the supplement.

P10, Figure 1: The amplitude of the seasonal cycle is max +-0.6. Assuming a tropospheric mixing ratio of 330 ppbv, this corresponds to less than 1 %. Thus, I would state that this is a minimal change and have difficulties to understand why is nevertheless still important to investigate the seasonal cycle of $N_2O$?

P10, Figure 1: Another comment on the figure or the discussion of this figure in the paragraph above. This figure indeed shows a stratospheric influence on the seasonal cycle of $N_2O$, but generally, I think there is now requirement that the minimum in $N_2O_{ST}$ overlaps with the minimum of N2O. Especially, with the known atmospheric transport time scales I would rather expect a time shift between the minima.

P11, L270: That descent is underestimated in atmospheric models has been shown in e.g. Brühl et al. (2007) and Khosrawi et al. (2009, 2018).

P11, Figure 4: Also here, first the figure should be described and then the results you get.

P11, Figure 4: Looking at the scale of the figure, the variation is +-1 ppb. As stated above, for a species having a mixing ratio of a few hundred ppb this change could be rated as being neglectable. Why is it nevertheless important to investigate the seasonal cycle?

P11, Figure 4 and respective text parts: Why is the anomaly strongest in summer. Shouldn't that rather be the case for winter/spring? Why do you detrend the data by using the de-seasonalized time series from Mauna Loa? Why do you do this and why is Mauna Loa used and not one of the other stations? Why you do what should be clearly described in the manuscript.

P12, L295: It is not clear why you use the HIPPO data here. You provide an explanation based on altitude, but although you show somewhat different figures than the previous one (latitude vs altitude instead of time vs altitude), the altitude region considered is the same. So the reasoning you provide is not clear at all.

P12, L292: Why is here now also the Atom data used? Repeat once again what your intention is of additionally using data from aircraft campaigns.

P12, Figure 5: Also here, it is not clear to me why you de-seasonalize the data with the $N_2O$ from Mauna Loa.

P13, Figure 6: Why does the HIPPO data show positive values $\Delta N_2O$ in the NH polar regions. I would have (at least in the winter months) expected negative values. I think I am now completely confused, because there is so much explanations/descriptions on your analyses missing.

P13, L300: The section title does not really reflect what you are actually showing in this section. You show here correlations between T and delta $N_2O$ and not the minimum in $N_2O$ or the seasonal variation of it.

P13, L301: There is a correlation but I would not call it significant. Have you checked if it is statistically significant? Is that what you mean here?

P13, L304: In figure 4 however only the South Pole is shown.

P13, L310: What are Atom and ORCAS? Campaigns? Even though you have described your data in the method section it should be repeated here once again.

P13, L300ff: This is all too quick and condensed described. Figure 7 and 8 definitely need more description and explanation so that it is possible to follow your analyses and results.

P14, L318-319: What does it mean that you derive only for Mace Head a significant correlation, but for all other stations not?

P14, Figure 7: You show here measurements and the results from the model, but why you show both and how they differentiate is not discussed in the text. Why then showing both?

P14, Figure : In this figure is too much information (see my previous comment). Also here it would be better to skip some of the subpanels or to split the figure into two figures.

P14, Figure 7 caption: "seasonal anomalies" since you look for minima I am confused, but the seasonal anomalies are also positive. Thus, as stated in my previous comment the section title is rather misleading.

P15, Figure 8: In order that these figure panels can be more easily compared with each other, they should be plotted in the same manner. Thus, the left panel should also be plotted from -70 S to 80 N, but masking the parts of the data that are not considered as white area.

P15, Figure 8 caption: The figure caption text is really confusing and needs more structure. Describe what is shown on the left and right (stick to the facts) and put everything else (results, reasoning etc.) into the main text.

P15, L339: Here, you should repeat again what your intention is with using the QBO and ENSO index.

P16, Figure 9 and P17, Figure 10: Add a legend? What is shown be the different colors and line styles?

P16, L345: Why is this correlation significant? Is it just statistically significant? From the correlation coefficient itself I would say that there is no or only a low correlation found.

P18, L375: Finishing reading this section I was wondering why these correlations are shown at all. It seems that there is no correlation or anti-correlation found. Why then discussing the correlation between $N_2O$ and QBO and ENSO?

P18, L380ff: Here you start to describe the atmospheric processes in detail, but that should be rather done somewhere in the paper between method and result section or embedded at the respective places in the result section. In the discussion you should just put your results rather in broader context of the scientific field and discuss uncertainties, agreements or disagreements with previous studies etc.

P19, L385: I do not see this.

P19, L398: Why slow? Couldn't also fast descent bring N2O depleted air down? Wouldn't it be that the amount of $N_2O$ depleted air would differ dependent on slow or fast descent?

P21, L456: What exactly is different in the NH compared to the SH? Wave activity is also high in the SH, so that alone cannot be responsible for the differences in dynamics between the hemispheres.

P21, L473: I am not sure if the differences in dynamics between the southern and northern hemisphere as described in Holton et al. (1995) are still valid. This should be checked. There have been many studies and new findings on atmospheric dynamics in the recent years.

P23, L521: The discussion section is much too long, too detailed and too speculative. This section should be significantly shortened and rewritten so that it is more concise. Focus on what is really important and directly related to your study and your results.

P23, L526: I would not say that the air accumulates there. The air descends during winter in the stratosphere and is then transported to the troposphere.

P23, L526: When exactly the air crosses the tropopause has not been shown and also no references are provided. How do you know when and to what amount (e.g. $N_2O$ fluxes) $N_2O$ is transported down to the troposphere? You get some information on this from the tagged tracer in your model simulation experiment, but that should be clearly discussed here.

P23, L526: Another question on this text line is, what is happening with $N_2O$ on its way? It is not chemically reacting in the troposphere, but the $N_2O$ depleted air will mix with the tropospheric $N_2O$ rich air on its way that will result in that the stratospheric signal will get less and less as longer $N_2O$ is on its way to the surface.

P23, L533: Stratosphere-troposphere exchange and how it is linked to the BDC is not shown and also no references have been provided.

References: The formatting of the references is not correct. See ACP manuscripts/guidelines. Further, there are several citations in the list that are not used, as e.g. Khosrawi et al. (2008) and Glatthor et al. (2007).

**Technical corrections:**

P5, L134: magl -> m AGL

P5, L149: four ~ month-long -> ~ four month-long

P7, L194: The abbreviation PLST has not been introduced.

P7, L198: Add which months you consider as winter months and which as spring months.

P15, L333: Figure 8 caption: First sentence not correct. Please rephrase.

P15, L334: I guess here it should read ORCAS and not Atom/Atom-2.

P15, L336-337: Sentence not clear. This needs more explanation. However, this should not be done in the figure caption, but in the main text.

P15, L338: Check title, some of the words should be rather starting with a small letter than a capital letter.

P17, L367: August-July -> July- August

P22, L505 and 506: The abbreviations OMZ and ETSP have not been introduced.

**References:**

Brühl, C., Steil, B., Stiller, G., Funke, B., and Jöckel, P.: Nitrogen compounds and ozone in the stratosphere: comparison of MIPAS satellite data with the chemistry climate model ECHAM5/MESSy1, Atmos. Chem. Phys., 7, 5585–5598, https://doi.org/10.5194/acp-7-5585-2007, 2007.

Khosrawi, F., Kirner, O., Stiller, G., Höpfner, M., Santee, M. L., Kellmann, S., and Braesicke, P.: Comparison of ECHAM5/MESSy Atmospheric Chemistry (EMAC) simulations of the Arctic winter 2009/2010 and 2010/2011 with Envisat/MIPAS and Aura/MLS observations, Atmos. Chem. Phys., 18, 8873–8892, https://doi.org/10.5194/acp-18-8873-2018, 2018.

Khosrawi, F., Müller, R., Urban, J., Proffitt, M. H., Stiller, G., Kiefer, M., Lossow, S., Kinnison, D., Olschewski, F., Riese, M., and Murtagh, D.: Assessment of the interannual variability and influence of the QBO and upwelling on tracer–tracer distributions of $N_2O$ and $O_3$ in the tropical lower stratosphere, Atmos. Chem. Phys., 13, 3619–3641, https://doi.org/10.5194/acp-13-3619-2013, 2013.

---

## Referee Comment (RC2)

**Review of "Northern vs. southern hemisphere differences in the stratospheric influence on variability in tropospheric nitrous oxide" by Nevison et al.**

This manuscript is a follow-up study of Nevison et al., (2011) published in ACP. In this study, the authors use simulations of nitrous oxide (N2O) from a Chemistry-Climate model (GEOSCCM) together with ground-based and air-borne observations of N2O to evaluate the impact of the stratospheric N2O on its concentrations in the troposphere. Both models and observations agree that N2O-poor air accumulates in the wintertime stratospheric polar regions and moves downward and equatorward, reaching the surface by early-autumn. The authors evaluate the impact of the ENSO, the QBO and the Brewer-Dobson Circulation (BDC, here quantified by the polar lower stratospheric temperatures) on the atmospheric growth rate of the surface N2O. They find that, in the Northern Hemisphere, the surface N2O atmospheric growth rate is negatively correlated with the polar lower stratospheric temperatures of the previous winter, and they attribute this correlation to the BDC. They also find that, in the Southern Hemisphere, the correlation between the surface N2O atmospheric growth rate and the polar lower stratospheric temperatures is not significant, but the correlations between the surface N2O atmospheric growth and the QBO and ENSO are significant. They argue that their findings are consistent with the current literature on the hemispheric differences in stratospheric transport and dynamics.

The manuscript presents a coherent analysis and is generally clearly written. The authors use a large number of datasets that makes this comparison really valuable and worth publication in ACP. However, I have the impression that the manuscript needs some adjustments before final publication in ACP. My main concern is that manuscript misses to appropriately highlight "what's new" and does not provide a possible outlook for future work. Therefore, I recommend publication in ACP once my comments below are addressed.

**Major Comments**

- Abstract. I suggest adding one/two sentences on "why this study is relevant". As it is, the authors mention right away what they did without providing information on why they did it. Similarly, I feel that the end of the Abstract lacks one/two conclusive sentences that highlight the main message of the paper and possible future paths to continue the work.

- Introduction. The Introduction lacks a paragraph briefly describing the datasets used in the study: the GEOSCCM model and ground-based and air-borne observational datasets. Such paragraph could be added around P3L70. Concerning the novelty of the study, the current paragraph (P3L71-76) could be improved by explicitly stating that his study is a follow-up of Nevison et al., 2011 (this info is available only for reviewers, is there a particular reason?) and that the novelty arises from 12 years of model development and acquisition of additional observations.
The Brewer Dobson Circulation (BDC) is mentioned a few times throughout the manuscript, but it is (briefly) described only in the Discussion section (Section 4.1). I think that such an important aspect of the manuscript deserves a few sentences in the Introduction (see also my minor comment about this below). The QBO also is described only in Section 2.4.2 and is hardly mentioned in the Introduction. Again, a few sentences about the QBO should appear in the Introduction (again, see also my minor comment about this below).

- Conclusions. The Conclusions lack to (re-)emphasize the novelty of the study and (more importantly) to highlight some strong forward-looking points of conclusions. As it is, the

Conclusions reads more like a summary of what has been done. To start, I suggest changing the title of the section from "Conclusions" to "Summary and conclusions". Then, I suggest reducing the repetition of information compared to the previous sections and provide a more detailed summary of what has been done (mention the model, observations, period, methods, a few numbers, …). Finally, I suggest adding some additional points of conclusions, here's a few ideas:

- o The authors find that the surface N2O growth rate presents hemispherical differences in the response to the impact from the QBO (strongest in the SH) and the BDC (strongest in the NH). Minganti et al., (2022) found hemispherical differences in the N2O trends in the stratosphere (positive in the SH and negative in the NH) in satellite observations and reanalyses. I wonder if these hemispherical differences in the stratospheric trends can be related to the differences in the surface N2O growth rate (or just mentioned).
- o It would be interesting to add one/two sentences on the possible impact of the solar activity on the N2O growth rate. The major chemical destruction of N2O occurs in the tropical upper stratosphere, so I do not expect large impact on the surface growth rate. However, some signal could still reach the troposphere and certainly an additional proxy for solar activity would help to better understand the N2O changes in the stratosphere.
- o I suggest mentioning the possible added/complementary value of satellite measurements of tropospheric (and stratospheric) N2O (e.g., the IASI, MLS, ACE-FTS instruments). The authors could also mention the use of Chemistry-Transport Models driven by different dynamical reanalyses. Since N2O has a very simple chemistry, such analysis would provide some insights about the dynamics of each reanalysis. In addition, maybe it could be worth mentioning the possible extension to additional CCMs?
- o The authors could mention the possibility to perform sensitivity tests with GEOSCCM. For example, an experiment with the QBO switched off (if possible) would isolate the patterns due only to the BDC.
- o The authors mention a few times the "stratosphere-troposphere exchange". I suggest dropping that terminology in the manuscript (as it opens a whole research subject) but highlighting the importance of further investigating the cross-tropopause transport/mixing of N2O and its seasonality.

- ENSO. I have the feeling that the discussion about the impact of the ENSO is out of scope in this study for two reasons:
- o First, the title of the manuscript states "stratospheric influence on….". The ENSO is primarily a tropospheric mode of variability that ultimately impacts the stratosphere. Because of that, I do not think that the ENSO belongs in a manuscript that addresses the impact of purely stratospheric processes such as the QBO and the BDC on the surface N2O AGR.
- o Second, the impact of the ENSO on the N2O AGR is not as direct to understand as for the QBO and the BDC. In Section 4.4, the authors argue that the correlation between N2O AGR and ENSO are mostly due to tropospheric meteorology and biogeochemical processes (first and second paragraphs). In addition, the authors declare that the impact of ENSO on the N2O AGR is hard to evaluate because of possible complicated mechanism difficult to capture (third and fourth).
  The difficulty to provide a convincing discussion for the impact of the ENSO on the

N2O AGR (contrarily to the QBO and BDC where the discussion is satisfying) is a sign that the ENSO should be removed from the manuscript (and maybe mentioned as possible future work in the Conclusions?).

Hence, I suggest removing the ENSO parts of the manuscript as it is not a purely stratospheric mode of variability, and its impact of the N2O AGR is difficult to interpret compared to the QBO and BDC. In addition, the removal of the ENSO discussion will imply the removal of Fig. 11 and the related discussion together with Sections 2.4.3 and 4.4. I think that this would be beneficial for the manuscript as it will allow more space for possibly enhancing the discussion about the PLST and the QBO.

- Results. The Results section suffer from a flaw that makes the manuscript a little hard to follow: the Figures are never properly introduced. Each Figure is always mentioned between parentheses after one/two sentences describing the results in that figure (e.g., P8L236-238). This is confusing as the reader does not know where to look (and what to look at) when going through the text. I suggest introducing the Figures at the beginning of every paragraph where each Figure is discussed – something like "Figure 1 shows latitude-altitude cross sections of N2O anomalies for GEOSCCM for January to December…".

**Minor Comments**

- P1L20 – as mentioned in my major comment above, I suggest starting the Abstract with one/two sentences on why this study is important.
- P1L25 – "… N2O atmospheric growth rate anomaly …" could the authors specify the months when this anomaly occurs?
- P1L25 – "… winter's polar lower stratospheric temperature." Could the authors specify these winter months? As the seasons could be austral or boreal, I think that specifying the month(s) avoids possible confusion (even when the Hemisphere is mentioned).
- P2L41 – I suggest adding the reference to Tian et al., (2020) together with Canadell et al., (2021).
- P2L62 – the QBO here should be briefly explained/introduced. See also my major comment above.
- P2L65 – I suggest removing the acronym STE as it is barely mentioned in the rest of the manuscript.
- P3L71 – I suggest adding a paragraph introducing the different datasets used in this study: the GEOSCCM model and the airborne and ground-based observations. See also my major comment above.
- P3L71-76 – This paragraph about the importance/novelty of the study should be improved. I suggest explicitly stating what is the added value of the study and that this is a follow-up of Nevison et al., (2011). See my major comment about this above.
- P3L77-87 – This paragraph should describe the structure of the manuscript, but it also introduces the GEOSCCM model (P3L80-82, this belongs to the paragraph describing the datasets mentioned above) and provides some anticipation of the results (P3L82-83 and P3L85-87). I suggest shortening this paragraph by moving the GEOSCCM part to the paragraph describing the datasets and removing the anticipation of the results. Furthermore, I suggest re-phrasing the paragraph like: "Section 2 describes the data and methods used in this work, Section 3 …."

- P3L84 – The ENSO is already defined, no need to repeat the meaning of the acronym.
- P3L90 – The GEOSCCM acronym is already defined.
- P3L93-95 – Could the authors provide more details on the implementation/meaning of these four N2O tracers?
- P3L96 – "… while a total N2O tracer …." Is this "total N2O" the result of the sum of the four tracers mentioned before (N2Ost, soil, ocean, anthropogenic source)? If yes, I suggest saying it explicitly.
- P3L97-98 – "… 2000-2019 …" could the authors mention the spin-up time for this run?
- P3L98 – "… 5 years of simulation …" why did the authors select the last 5 years of the simulation for the seasonal cycle and not the last 10? A longer period would smooth the internal variability of the model.
- P4L107 – I suggest changing "(NOAA/HATS) (Thompson et al., 2004)" to "(NOAA/HATS, Thompson et al., 2004)".
- P4108 – "(NOAA/CCGG)" is it possible to provide a reference here?
- P4L109 – "… 5 baseline sites." I suggest mentioning the location of the sites.
- P4L115 – I suggest replacing "near" with "approach".
- P4L118 – Is it possible to provide a reference instead of (or together with) the link?
- P5L137 – What is "Kriging"? I suggest explaining it or re-phrasing.
- P5L137 – I suggest changing "(Hammerling et al., 2012)" with "Hammerling et al., (2012)".
- P5L138 – I suggest replacing "QCLS" (not defined yet) with "airborne".
- P5L143 – Could the authors provide a reference for the HIPPO project?
- P5L151-152 – "(Note: ….)" I suggest removing this sentence as the Atom-1 deployment is not mentioned further in the manuscript.
- P6L166-167 – "which was plotted …. As described below." This sentence does not really belong here as it already mentions plotting and proxies. I suggest removing it.
- P6L180 – "BDC" the acronym was not introduced before. It should appear in the Introduction (where the BDC should be first described), see my major comment on that above.
- P6L183-185 – "(Note: …)" This is a good point and I suggest keeping it as a normal sentence (i.e., without brackets and "Note"). Something like "We highlight that strong local…."
- P6L188 – "… the latter…" Do the authors mean the QBO and ENSO indices here? If yes, I suggest specifying it by adding "… the latter two indices…" or something similar.
- P7L193-194 – "… in winter/spring … (September-November) in the SH …" I suggest re-phrasing this part to something like "… in January-March (winter/spring in the NH) and September-November (spring in the SH) …".
- P7L194 – Could the authors provide the meaning of the MERRA-2 acronym?
- P7L195 – "… stratospheric downwelling" I suggest adding "… stratospheric downwelling due to the BDC (e.g., Holton, 2004)."
- P7L198 – I suggest adding the months between parentheses after "Winter months" and "spring months". Something like "Winter months (January-March) …. spring months (September-November)". I think it is better to have the explicit months together with the season when discussing the Northern and Southern Hemispheres.
- P7L204-205 – This sentence is very pertinent for introducing the QBO. I suggest moving it to the appropriate new paragraph in the Introduction were the QBO and the BDC are first mentioned. See my major comment about this above.
- P7L208 – Is it possible to have a reference together with the link?
- P7L215 – The ENSO acronym was already defined before, there's no need to define it again.

- P7L216-217 – "El Nino is … Kelvin waves (…)" this sentence should be moved to the Introduction where the ENSO should first be mentioned. However, given my major comment about removing the ENSO discussion, this point can be disregarded.
- P8L231-233 – "(Note: ….)" This sentence about ENSO can be removed (if discussion about ENSO is removed) or re-phrased to avoid the "Note:" (if ENSO discussion is kept).
- P8L236 – Figure 1 needs to be introduced. See my major comment above.
- P8L236-238 – "GEOSCCM simulates … (SH) (Figure 1)" Could the authors specify the months of winter ("during winter in the polar lower stratosphere") and springtime/early summer ("in springtime (early summer)")?
- P8L240 – "the surface minimum … hemispheres" please rephrase with something like "the minimum in the lower troposphere does not reach the surface until …"
- P8L240 – please introduce Figure 2.
- P9, Figure 1 – Since the tropopause is mentioned in the text, it would be nice to show the tropopause level in Figure 1.
- P9, Figure 1 – I suggest providing the units of the N2O anomalies on the right of the colorbar, instead of repeating them at the top right corner of each panel.
- P9, Figure 1, caption – "GEOSCCM N2O anomalies" please specify the units of the N2O anomalies.
- P9, Figure 1, caption – please replace "in a monthly sequence" with "as a function of".
- P9, Figure 1, caption – "up to 30 hPa (24 km)" you state that you show from the surface to 30 hPa (24 km), but the top level of the panels of Figure 1 is 200 hPa (corresponding to 12 km). Could the authors explain that?
- P9, Figure 1, caption – please insert something like "From left to right: January, February, March, April (top row); May, … (middle row); September, … (bottom row)".
- P9, Figure 1, caption – "The GEOSCCM N2O …. the anomalies" this sentence belongs more to the text. I suggest moving it to the part describing Figure 1.
- P9, Figure 1, caption – "Mauna Loa" the authors never mentioned Mauna Loa before in the text. I suggest moving this part to the text and mention the Mauna Loa sampling in the Methods section. There are other occurrences of the Mauna Loa sampling in the caption of other figures, therefore I suggest moving the Mauna Loa sampling description into the appropriate Methods section and not repeating it again.
- P9, Figure 2 – Also here, please provide the units on the right of the colorbar.
- P9, Figure 2 – the bottom right panel (50S) has an additional label on the y axis (1000) that is not present in the other panels. I suggest removing it.
- P9, Figure 2 caption – please replace "N2O anomalies vs month" with something like "N2O anomalies as a function of month and altitude".
- P9, Figure 2 caption – please rephrase "in the northern (top row) and southern (bottom row) hemispheres" with "in the NH (top row) and in the SH (bottom row)".
- P9, Figure 2 caption – please specify the meaning of the columns. Something like: "From left to right: 70°, 60° and 50°".
- P9, Figure 2 caption – "The GEOSCCM N2O fields … relative to the NH" I suggest moving (and appropriately re-phrasing to fit the context) this sentence to the text where Figure 2 is introduced.
- P10L252 – please introduce Figure 3.
- P10L252 – I suggest replacing "stratospheric loss" with "N2O-depleted air".
- P10L254 – I suggest adding "for all sites" after "N2Ost minimum".
- P10L255 – I suggest replacing "Southern Hemisphere" with "SH".

- P10L256 – I suggest replacing "northern latitudes" with "latitudes in the NH".
- P10, Figure 3 – I suggest swapping the panels for Summit (70N) and Barrow (71N). Barrow is northern than Summit and it would be consistent with the current layout of the panels (north to south from top left to bottom right).
- P10, Figure 3 – I suggest removing the IDs of the stations on the bottom left corner of the panels (alt, sum, brw, …) because the names of the sites are already in the titles of the panels.
- P10, Figure 3 caption – please mention the names of the sites after "… 9 surface sites". Something like "Top row from left to right: Alert (Canada), …; middle row from left to right: Mace Head (Ireland), ….; bottom row from left to right: Cape Grim (Country?), ….".
- P10, Figure 3 caption – "The black heavy … observed N2O" should come before "the red line …".
- P10, Figure 3 caption – I suggest re-phrasing "The red line … tracer N2Ost" with something like "For GEOSCCM, the total N2O from all forcings is in red, and the stratospheric tracer N2Ost is in dashed red".
- P11L264 – please introduce Figure 4.
- P11L264-265 – "When viewed as a … up to 8 km" This sentence (properly re-phrased to correctly describe a figure) should be included in the description of Fig. 4.
- P11L266 – please replace "is felt in" with something like "reaches".
- P11L268 – again, please introduce Fig. 5 here.
- P11, Figure 4 caption – as for Figures 1 and 2, the authors mention Mauna Loa in the caption with no reference in the text. I suggest moving (and expanding) this part to the Methods section.
- P12L285 – please introduce Figure 6.
- P12L289 – I suggest removing "April".
- P12L289 – "By June it has descended…" I find difficult to see that the anomaly has descended into the troposphere by June. I suggest re-phrasing with something like "By June it starts descending into …".
- P12L291-292 – "Notably, … (Supplementary Figure 1)" this sentence states that the fuller dataset does not provide results as clear as the subset shown in Fig. 6. Do the authors know why? It would be interesting to provide a few words of explanation.
- P13, Figure 6 – The August panel of Figure 6 (for the NH) does not agree well with the corresponding August panel of Figure 4 (considering the same latitudes and altitudes). Do the authors know why? Maybe because the periods are different?
- P13, Figure 6 caption – could the authors explain the meaning of "transects"?
- P13, Figure 6 caption – "…. an annual sequence." I suggest saying explicitly what contains each panel. Something like "an annual sequence; from left to right: January, March, June (top row) and August, November (bottom row)."
- P13, Figure 6 caption – "HIPPO data extend … influence on tropospheric N2O" This sentence (properly re-phrased to blend into the text) belongs to the text in the introduction of Figure 6.
- P13L301 – please introduce Figure 7.
- P13L301 – "(PLST)" PLST is already defined – no need to define it again here.
- P13L303 – "seasonal minimum" I suggest adding the reference to the specific panel of the Figure (Fig. 7a).
- P13L304 – I suggest removing the IDs of the stations (CGO, PSA, SPO).
- P13L304 – I suggest changing "(Figure 7)" to something like "(Fig. 7b,c)".

- P13L308 – I suggest changing "(Figure 7)" to something like "(Fig. 7e,f)".
- P13L308 – "In support of …." before this sentence please introduce Figure 8.
- P13L308 – P14L313 – In Figure 8, the authors compare different observational datasets (Atom and ORCAS) for different periods (2016 and 2017). In my opinion, this makes the discussion difficult to follow. I suggest using only the ORCAS dataset for Figure 8. This would allow more room for discussion about the ORCAS dataset (maybe separating January and February?) and remove the asymmetry in Figure 8.
- P14L312 – I suggest changing "(weak Brewer Dobson circulation)" with "(weak BDC)".
- P14L313 – I suggest changing "(strong Brewer Dobson circulation)" with "(strong BDC)".
- P14L315-319 – this paragraph refers to generally unsignificant correlations for a Figure in the Supplement. I suggest reducing the importance of this paragraph to one/two sentences and move it to the end of the discussion of Figure 7.
- P14, Figure 7 – I suggest removing the bottom left panel (GEOS ΔN2O vs GEOS Strat T) for two reasons: 1) it is not explicitly mentioned in the text and 2) it shows N2Otot from GEOSCCM that is not the most relevant tracer for this comparison.
- P15, Figure 7 caption "The correlation between … its seasonal minimum" I suggest moving this sentence to the text.
- P15L339 – please introduce Figure 9.
- P15L339 – P18L371 – Discussion of Figures 9 and 10. As it is, the discussion forces the reader to swap back and forth between Fig. 9 and Fig. 10 to follow the structure of the text (currently describing first the QBO in the SH and NH and then the PLST in the SH and NH). I suggest keeping the structure of the text as it is and re-arranging the figures accordingly: Fig. 9 should contain the QBO for both SH and NH, and Fig. 10 the PLST for SH and NH.
- P16L353 – please remove "of the weak".
- P16L353 – please replace "correlations" with "correlation".
- P17L364 – please add "(Fig. 9b)" after "NOAA".
- P17L365 – please replace "(Figure 9b,d)" with "(Fig. 9d)".
- P18L373-375 – based on my previous comment of the ENSO part, this discussion (and Figure 11) should be removed. However, if the authors decide to keep it, please introduce Figure 11.
- P18L382 – "vertical profiles" there is no vertical profile in the manuscript (for me vertical profiles are line plots with the values on the x axis and the altitude/pressure on the y axis). I suggest replacing "vertical profiles" with something like "latitude-pressure cross sections".
- P18L382 – please re-phrase "big picture" with something like "broad".
- P19L387 – please replace "Brewer Dobson circulation" with "BDC".
- P19L394-395 – "which transports warm … in the winter hemisphere" The impact of the BDC is not limited to the transport of N2O-poor air: the BDC also transports ozone, GHGS and CFCs and maintains the thermal structure of the stratosphere (Butchart, 2014, Minganti et al., 2020). I suggest re-phrasing this sentence to include also these important effects. In addition, based on my major comment above, the BDC needs to be described in the Introduction, so I suggest moving this description to the appropriate part in the Introduction.
- P19L394 – please replace "N2O-depleted" with "N2O-poor".
- P19L396 – "a large seasonal amplitude in the polar lower stratosphere" could the authors specify the amplitude of what?
- P19L405 – I suggest replacing "upwells" with something like "is transported".
- P19L410 – I suggest replacing "(~32 km) (Strahan et al., 2021 ….)" with "(~32 km, Strahan et al., 2021 ….)".

- P20L423 – I suggest replacing "in the July-September period" with "in July-September".
- P20L430 – I suggest removing the quotes from surf zone.
- P20L433 – I suggest removing "relatively".
- P20L436 – also here, I suggest removing "relatively".
- P20L436 – I suggest replacing "is felt at" with "reaches the".
- P20L437 – I suggest replacing "permits" with "results in".
- P21L456 – I suggest being more specific and replace "more complex atmospheric dynamics in the NH stratosphere" with something like "larger wave activity in the NH compared to the SH (Scaife and James, 2000, Kidston et al., 2015)".
- P21L461 – "mixing of air" I suggest adding a reference here. For example, Shepherd, 2007.
- P21L466 – I suggest removing the link because the text between parentheses is not a citation.
- P21L474 – I suggest removing "stratosphere-troposphere exchange" as it was barely mentioned before in the manuscript. I suggest replacing it with something like "cross-tropopause transport".
- P22L496 – I suggest replacing "heightened" with "increased".
- P23L516 – I suggest re-phrasing "testament to the strength" with something like "demonstration of the strength".
- P23L522 – "Conclusions". As it is, the Conclusions section reads more like a summary than a Conclusion. Because of that, I suggest renaming this section as "Summary and Conclusions". In addition, the Conclusions section needs to re-emphasize what is new in this study and address more points of conclusion and possible future work. More on this in my major comment above.
- P23L527 – "summer to early-autumn period" could the authors specify the months here?
- P23L530 – I suggest replacing "delivered" with "transported".
- P23L532-533 – "Stratosphere-troposphere exchange". As one of my comments on this before, I suggest replacing "Stratosphere-troposphere exchange" with something like "Cross-tropopause transport".

**References**

Holton, J.: An Introduction to Dynamic Meteorology, no. v. 1 in An Introduction to Dynamic Meteorology, Elsevier Science, available at: https://books.google.be/books?id=fhW5oDv3EPsC (last access: 28 October 2020), 2004.

Kidston, J., Scaife, A., Hardiman, S. *et al.* Stratospheric influence on tropospheric jet streams, storm tracks and surface weather. *Nature Geosci* **8**, 433–440 (2015). https://doi.org/10.1038/ngeo2424

Minganti, D., Chabrillat, S., Christophe, Y., Errera, Q., Abalos, M., Prignon, M., Kinnison, D. E., and Mahieu, E.: Climatological impact of the Brewer–Dobson circulation on the $N_2O$ budget in WACCM, a chemical reanalysis and a CTM driven by four dynamical reanalyses, Atmos. Chem. Phys., 20, 12609–12631, https://doi.org/10.5194/acp-20-12609-2020, 2020.

Minganti, D., Chabrillat, S., Errera, Q., Prignon, M., Kinnison, D. E., Garcia, R. R., et al. (2022). Evaluation of the $N_2O$ rate of change to understand the stratospheric Brewer-Dobson circulation in a Chemistry-Climate Model. *Journal of Geophysical Research: Atmospheres*, 127, e2021JD036390. https://doi.org/10.1029/2021JD036390

Scaife, A.A. and James, I.N. (2000), Response of the stratosphere to interannual variability of tropospheric planetary waves. Q.J.R. Meteorol. Soc., 126: 275-297. https://doi.org/10.1002/qj.49712656214

Shepherd, T. G.: Transport in the middle atmosphere, J. Meteorol. Soc. Jpn. Ser. II, 85, 165–191, 2007.

Tian, H., Xu, R., Canadell, J.G. et al. A comprehensive quantification of global nitrous oxide sources and sinks. Nature 586, 248\u2013256 (2020). https://doi.org/10.1038/s41586-020-2780-0

---

## Author Comment (AC1)

**We thank the reviewers for their thorough and detailed reviews, which were very helpful in encouraging us to reorganize, clarify, and revise our manuscript. Please find the reviewers' comments below in plain text and our responses in bold font.**

Reviewer 1 Farahnaz Khosrawi
Nevison et al. present a follow-up study of their publication in ACP from 2011. This is quite interesting and valuable study and deserves to be published, however the current version of the manuscript needs significant improvements.
Please find my detailed review with comments and suggestions for improvement in the attachment. Since I submitted a community comment to their manuscript in 2011 and am now referee of this study I think I can skip being anonymous.
Attachment
Review on egusphere-2023-2877 Northern vs southern hemisphere differences in the stratospheric influence on variability on tropospheric nitrous oxide by Nevison et al. Nevison et al. present a follow-up study of their publication in ACP from 2011.
**This comment was visible to the reviewers as part of a cover letter statement to the editors, which asked why the paper was appropriate for ACP. However, an additional part of that statement, which apparently was not visible to the reviewers, was, "*This paper was inspired by competing results in the literature, which find alternatively that stratospheric influences or El Nino Southern Oscillation (ENSO) cycles are the dominant mechanism driving variability in tropospheric N2O. … Within the literature that identifies the stratosphere as the dominant influence, there are further disagreements and hemispheric differences regarding whether the quasi-biennial oscillation (QBO) directly influences surface N2O. These questions are relevant to our ability to interpret biogeochemical source signals in the tropospheric variability in $N_2O$ …*" Thus while we appreciate the suggestions of Reviewers 1 and 2 to remove the discussion of ENSO and QBO, this is difficult because this discussion provided some of the major motivation for the study. Since the Reviewers' suggestions were made in large part due to the need to reduce the length of the paper, we have addressed their comments by shortening the paper and rearranging the figures in a more logical order, while also providing an improved background and motivation for the paper.**

In their current study they present simulations with a GCM using a tagged stratospheric tracer of N2O to investigate the stratospheric influence on the seasonal cycle of tropospheric N2O. Additionally, they use the QBO and ENSO index and the polar lower stratospheric temperature as proxy for atmospheric transport and correlate these parameters with the N2O atmospheric growth rate. They find that the atmospheric growth rate anomaly in the southern hemisphere is better correlated to the QBO index, than to polar lower stratospheric temperature and that this hemispheric difference is consistent with the current knowledge of hemispheric difference in atmospheric dynamics. This is a very interesting study and a valuable continuation of the previous work of Nevison et al. (2011). However, the current version of this manuscript is completely overloaded. There is too much analyses presented and a lot of necessary information missing, this is especially the case for the result section. You cannot expect the reader to read several papers additionally to your current manuscript to be able to follow the presentation of your current results. On the other hand, a very extensive discussion section is provided which rather confuses than helps to understand the dynamical processes behind. Thus, I would suggest

major revisions before publication in ACP based on the comments and corrections given below. General comment:

• My major comment and suggestion for improvement would be to remove the entire part with the ENSO and QBO index and correlation of these to the atmospheric growth rate.

**Please see response above for why we have not removed these sections.**
I would suggest to instead focus on the experiment with the tagged stratospheric tracer which is a really great and convincing tool to study the stratospheric influence on the tropospheric seasonal cycle of N2O. From the correlation analyses part you could make a second paper where you also use model experiments to better substantiate the dynamical processes behind. So far it has been done in a rather speculative way than providing real proof. With skipping that part you would still have 8 figures plus 2 from the supplement, thus 10 in total which would be a reasonable number.
**We have clarified how the tagged tracer was calculated but have not**
• In many occasions your provide already statements on the results before you have shown the respective figure/analysis. This is not a good way of writing and causes only confusion. Thus, I would suggest that you first describe what is shown in the figures/what analyses has been done and then provide the results you gain with exactly stating what leads you to your specific result/conclusion.

**Although Reviewer 2 made a similar comment, some of us have been trained since graduate school NOT to present results this way! We were taught that it is better form to describe the point made by the figure rather than open the sentence by flatly stating what the figure shows, which in any case should be explained already by the figure caption. We have tried to compromise by citing the figure number early in the first sentence describing the figure, including a clause briefly describing what the figure shows, but also including the main point made by the figure in that opening sentence.**

• A lot of explanation/description of the figures/analyses is missing in the result section. It is really hard to follow what exactly has been done and why. This definitely needs to be improved. This of course will make your manuscript somewhat longer, but more valuable for the scientific community. Further, to have a not too long paper, we are back at my first comment to then skip the part with QBO and ENSO index, because the results of the tagged tracer experiment provide enough content for one paper.
**We have reorganized the figures in a more logical order and, as described above, rewritten the Results to present figure number early in the first sentence describing the figure. We have also rewritten the figure captions to explain more clearly what is shown in the figure and have removed all interpretative sentences from the figure captions and moved them to the text.**

• In the submission form you stated that your paper is a follow-up study of Nevison et al. (2011), but unfortunately, nowhere in the paper you state what is the new analyses/finding compared to your previous study.
**Added a paragraph to the Introduction: "This paper explores the causes of variability in both the seasonal cycle and the AGR of tropospheric $N_2O$. It follows up on previous work**

by *Nevison et al.* (2011), who inferred a stratospheric influence on surface atmospheric $N_2O$ observations based entirely on correlations between interannual variations in stratospheric indices and detrended $N_2O$ anomalies in months surrounding the seasonal $N_2O$ minimum. In the meantime, altitude-latitude cross sections have become available from aircraft surveys that span a full seasonal cycle. In addition, advances in model development allow for explicit simulation of stratospheric $N_2O$ tracers (*Ruiz et al.*, 2021; *Liang et al.*, 2022)."

• Figure captions should generally only describe what is shown in the figure not provide any statements on the results or method. This should be rather described in the respective sections. Thus, all figure captions need to be revised.
**All interpretative statements have been moved from the captions to the text.**

Specific comments:

P1, L20ff: The abstract needs to be revised. See ACP guidelines. You solely describe your results, but you do not provide any introduction to the topic (e.g. why is it important?). Further, the abstract should end with some sentences stating what the importance and implications of your results are.
**The abstract has been revised to begin with a motivating introduction and end with a forward-looking conclusion.**
P3, L82-83: This is a result and should not appear in the introduction.
**Removed**
P3, 87: In the introduction I would have expected more information on the tagging experiment you have done. What is tagging and for what is it used. Not every reader is familiar with this technique. So a short explanation is needed.
**We have revised the Introduction sentence to read, "(GEOSCCM) … includes a tagged stratospheric $N_2O$ tracer that is transported individually in the model and distinguished from tropospheric tracers of fresh surface emissions (*Liang et al.*, 2022)." We also added a fuller description of the tagging process in the Methods paragraph just below, "Following the approach of *Liang et al.* (2008), the tropospheric tagged tracers become the tagged stratospheric tracer, $N2O_{ST}$, when they are transported across the tropopause, and retain that identity even when $N2O_{ST}$ re-enters the troposphere, thereby providing a model estimate of the stratospheric influence on tropospheric $N_2O$."**
P4, L102: Add here some sentences stating how good the model is plus references. Are there evaluation studies that show the quality of the model simulations especially with respect to atmospheric dynamics? That many models have problems simulated a proper QBO is discussed e.g. in Khosrawi et al. (2013).
**Added, "GEOSCCM has been evaluated extensively in multi-model assessments and shown to represent well the mean atmospheric circulation, the interhemispheric exchange rate, the mean age of air in the tropical and polar stratosphere, and the mean atmospheric lifetime of $N_2O$ (*Liang et al.*, 2022 and references therein)."**
P5, L137: "local Kriging"? What to you mean with "Kriging"? I have never heard about this and I guess you should provide some more explanation. Further, the reference should appear without parenthesis (should be embedded in the text).
**The Kriging reference has been deleted – it is not really relevant to this paper.**
**Also, "(Hammerling et al., 2012)" was replaced with "Hammerling et al., (2012)".**

P5, L150: What is meant with "2-4"? Are these the campaign numbers? Please clarify.
**Rewrote the description of the ATom campaign with, "ATom consisted of 4 deployments over 3 years, with each deployment approximately 1 month long (*Thompson et al.*, 2022). QCLS N₂O was measured during the second through fourth ATom deployments in January/February 2017, September/October, 2017 and April/May 2018, respectively (*Gonzalez et al.*, 2021). (Note: N₂O measurements are not available from the first ATom deployment in July/August 2016 due to technical problems.)"**

P5, L152: What does this mean? Is thus data missing for this time/time period? Please clarify and rephrase sentence accordingly.
**Rewrote as (Note: N₂O measurements are not available from the first ATom deployment in July/August 2016 due to technical problems.)**

P8, L236: It would be better if you first would describe what is shown in the figure and what kind of analyses you have done before you make statements of what the result is. What exactly in this figures does point to your interpretation that there is a stratospheric influence?
**We have removed the old Figure 1**

P10, L251: Not stratospheric loss. I guess you mean here stratospheric air or air characterized by low N2O.
**Replaced "loss" with "air depleted in N₂O"**

P10, L257: This deviation could have many reasons. Either the model has deficiencies in the sources and sinks of N2O or atmospheric transport. Not clear if stratospheric or tropospheric processes are the cause of the shift. It could be that N2OST is correct, but N2O not.
**Agreed. We have added some sentences in the Discussion about the questions raised by the model. "**

P10, Figure 3: You show 6 stations for the NH, but only 3 for SH. Why do you show less for the SH than for the NH. How have these stations been selected? What where the criteria for choosing these?
**There are 4 sites in the SH and 5 in the NH, chosen to span a full range of latitudes. Section 2.2.1 also now includes, "All of the NOAA sites considered in this study, including Alert, Canada; Summit, Greenland; Mace Head, Ireland; Cape Matatula, Samoa; Palmer Station, Antarctica and the HATS baseline sites listed above, are long-standing remote sites situated away from strong local anthropogenic sources."**

P10, Figure 3: This figure is overloaded. You do two analysis in one figure. First, you use observations to evaluate the N2O seasonal cycle from the observations. Then you use N2OST and the other curves to investigate the processes that affect the seasonal cycle. Thus, I would suggest to split the figure. One where you compare N2Oobs with N2O model and one where you show N2OST, N2Osoil, N2Oocean and N2Oanthro. The complete figure showing all curves in one plot as done now, could be provided in the supplement.
**To reduce the overload in the figure we have combined N2Osoil, N2Oocean and N2Oanthro into a single line representing tropospheric emissions. This seems like a better solution than adding a separate figure, given the clear message from the reviewers that the paper is too long already.**

P10, Figure 3: The amplitude of the seasonal cycle is max +-0.6. Assuming a tropospheric mixing ratio of 330 ppbv, this corresponds to less than 1 %. Thus, I would state that this is a minimal change and have difficulties to understand why is nevertheless still important to investigate the seasonal cycle of N2O?

**Good point. We've addressed this issue by expanding a paragraph in the Introduction, saying, "High precision measurements of $N_2O$ have revealed interannual variability in its atmospheric growth rate (AGR) and small-amplitude seasonal cycles in the range of 0.4-1 ppb (i.e., 0.1 – 0.3% of the background mixing ratio) (*Nevison et al.*, 2004; 2007; 2011; *Jiang et al.*, 2007; *Thompson et al.*, 2013). Spatial gradients in atmospheric $N_2O$ are also small, e.g., the northern hemisphere (NH) minus southern hemisphere (SH) difference is approximately 1 ppb (*Thompson et al.*, 2014b; *Liang et al.*, 2022). While strong local sources can create larger spatial and seasonal signals in atmospheric $N_2O$ at some sites, such as those influenced by agriculture or coastal upwelling (*Lueker et al.*, 2003; *Nevison et al.*, 2018; *Ganesan et al.*, 2020), at sites remote from local sources, even variations of 0.2 ppb in estimated background $N_2O$ levels can significantly affect the magnitude of $N_2O$ emissions inferred from atmospheric inversions (*Nevison et al.*, 2018). "**

P10, Figure 3: Another comment on the figure or the discussion of this figure in the paragraph above. This figure indeed shows a stratospheric influence on the seasonal cycle of N2O, but generally, I think there is now requirement that the minimum in N2OST overlaps with the minimum of N2O. Especially, with the known atmospheric transport time scales I would rather expect a time shift between the minima.

P11, L270: That descent is underestimated in atmospheric models has been shown in e.g. Brühl et al. (2007) and Khosrawi et al. (2009, 2018).

**These references have been added to the discussion in section 4.0**

P11, Figure 4: Also here, first the figure should be described and then the results you get.

**Please see our general response to the Major Comment on introducing figures.**

P11, Figure 4: Looking at the scale of the figure, the variation is +-1 ppb. As stated above, for a species having a mixing ratio of a few hundred ppb this change could be rated as being neglectable. Why is it nevertheless important to investigate the seasonal cycle?

**This issue is addressed now in the Discussion, as described in the response to the similar comment above.**

P11, Figure 4 and respective text parts: Why is the anomaly strongest in summer. Shouldn't that rather be the case for winter/spring? Why do you detrend the data by using the de-seasonalized time series from Mauna Loa? Why do you do this and why is Mauna Loa used and not one of the other stations? Why you do what should be clearly described in the manuscript.

**The Methodology for the empirical background now describes why and how the Mauna Loa fit is removed.**

P12, L295: It is not clear why you use the HIPPO data here. You provide an explanation based on altitude, but although you show somewhat different figures than the previous one (latitude vs altitude instead of time vs altitude), the altitude region considered is the same. So the reasoning you provide is not clear at all.

**HIPPO is the only one of the 3 QCLS campaigns (HIPPO, ORCAS, ATom) that provides data over a full seasonal cycle. ORCAS was focused on summer in the SH, while ATom intended to cover a full seasonal cycle but the N2O measurements from the first deployment in July/August are not available due to technical problems)**

P12, L292: Why is here now also the Atom data used? Repeat once again what your intention is of additionally using data from aircraft campaigns.

**This is moved to the Discussion, in the context of describing the synoptic influences on the QCLS**

P12, Figure 5: Also here, it is not clear to me why you de-seasonalize the data with the N2O from Mauna Loa.

**This is now described in the Methods. It is a site that allows us to remove a mean value characteristic of the mid troposphere and present altitude-latitude contour plots of N₂O as anomalies.**

P13, Figure 6: Why does the HIPPO data show positive values ΔN2O in the NH polar regions. I would have (at least in the winter months) expected negative values. I think I am now completely confused, because there is so much explanations/descriptions on your analyses missing.

**The color scales of the N2O anomalies between Figure 4 (-1.0 to 1.0 ppb) and Figure 5 (-5 ppb to 5 ppb) are necessarily different, since HIPPO extends into the lower stratosphere where the depletion signals are stronger. HIPPO also represents a synoptic snapshot north-south across a single longitude where positive anomalies can occur due to plumes associated with strong surface emissions. The contours in the HIPPO plots are negative down to the surface in the August panel and are also negative through the mid troposphere in June.**

P13, L300: The section title does not really reflect what you are actually showing in this section. You show here correlations between T and delta N2O and not the minimum in N2O or the seasonal variation of it.

**We have renamed the section "Correlation analysis of N₂O seasonal anomalies." As noted in the methods, PLST is the only correlate considered because "QBO and ENSO are monthly indices for which it is not straightforward to choose a representative month to correlate to the N₂O anomaly, given that the anomaly might result from the cumulative effect over multiple months. PLST in contrast has one unique value each year that can be plotted against that year's N₂O anomaly for any given month."**

P13, L301: There is a correlation but I would not call it significant. Have you checked if it is statistically significant? Is that what you mean here?

**The p value is now shown on the plot to support the claim that the correlation is significant.**

P13, L304: In figure 4 however only the South Pole is shown.

**It is noted in the text that South Pole is shown. Reviewer 3 requested that the number of panels in this figure be reduced to avoid showing largely redundant results.**

P13, L310: What are Atom and ORCAS? Campaigns? Even though you have described your data in the method section it should be repeated here once again.

**Introduced with a reminder as follows, "Figure 8, which compares February altitude-by-latitude QCLS N₂O contour plots from the ORCAS and ATom airborne surveys, offers support for the observed surface correlations."**

P13, L300ff: This is all too quick and condensed described. Figure 7 and 8 definitely need more description and explanation so that it is possible to follow your analyses and results.

**Separate paragraphs are included for Figure 7 and 8 (now 6 and 7) and the description of them is expanded.**

P14, L318-319: What does it mean that you derive only for Mace Head a significant correlation, but for all other stations not?

**It may not be meaningful and is moved to the end of the results and de-emphasized.**

P14, Figure 7: You show here measurements and the results from the model, but why you show both and how they differentiate is not discussed in the text. Why then showing both?

**We have added a sentence to explain why both are shown, "The similarity of the observed and modelled correlations and the fact that they occur for $N_2O_{ST}$, which is not influenced by fresh surface emissions, as well as for total $N_2O$ in GEOCCM, suggests they are caused by stratospheric influences."**

P14, Figure 7: In this figure is too much information (see my previous comment). Also here it would be better to skip some of the subpanels or to split the figure into two figures.

**The figure has been reduced from 6 to 4 subpanels.**

P14, Figure 7 caption: "seasonal anomalies" since you look for minima I am confused, but the seasonal anomalies are also positive. Thus, as stated in my previous comment the section title is rather misleading.

**The section title is changed to "Correlation analysis of $N_2O$ seasonal anomalies."**

P15, Figure 8: In order that these figure panels can be more easily compared with each other, they should be plotted in the same manner. Thus, the left panel should also be plotted from -70 S to 80 N, but masking the parts of the data that are not considered as white area.

**That would leave a lot of blank space on the ORCAs panel. We've added a sentence to remind readers that ORCAS focused on the Southern Ocean and was restricted to the extratropical SH.**

P15, Figure 8 caption: The figure caption text is really confusing and needs more structure. Describe what is shown on the left and right (stick to the facts) and put everything else (results, reasoning etc.) into the main text.

**The detrending and binning information has been moved to the Methods and all reasoning has been moved into the text.**

P15, L339: Here, you should repeat again what your intention is with using the QBO and ENSO index.

**We have added sentences at the beginning of Section 3.2 and 3.3 that state the intention of the correlation analyses for N2O AGR and monthly anomalies, respectively.**

P16, Figure 9 and P17, Figure 10: Add a legend? What is shown by the different colors and line styles?

**The colors of the left and right Y axes are now matched to the respective lines on the graph.**

P16, L345: Why is this correlation significant? Is it just statistically significant? From the correlation coefficient itself I would say that there is no or only a low correlation found.

**The Introduction now includes some sentences acknowledging that the variability in N2O is very small, but that, "at sites remote from local sources, even variations of 0.2 ppb in estimated background $N_2O$ levels can significantly affect the magnitude of $N_2O$ emissions inferred from atmospheric inversions (*Nevison et al.*, 2018)." Due to the small variability, the correlations between N2O anomalies and causal indices are generally weak, but still statistically significant, as indicated by p values < 0.05.**

P18, L375: Finishing reading this section I was wondering why these correlations are shown at all. It seems that there is no correlation or anti-correlation found. Why then discussing the correlation between N2O and QBO and ENSO?

**We also now begin Section 3.2 by stating the intent of the correlation analyses, "In this section NOAA surface $N_2O$ AGR anomalies from 1998-2020 are plotted against polar lower stratospheric temperature (PLST) and QBO and ENSO indices, with varying lag times as described in the Methods. The analysis focuses on the NOAA global, NH, and SH mean products, with the premise that a significant correlation between the interannual**

**variability in the N₂O AGR and the stratospheric or ENSO indices can be interpreted to support a causal influence of the latter on the N₂O AGR.”**

P18, L380ff: Here you start to describe the atmospheric processes in detail, but that should be rather done somewhere in the paper between method and result section or embedded at the respective places in the result section. In the discussion you should just put your results rather in broader context of the scientific field and discuss uncertainties, agreements or disagreements with previous studies etc.

**The Discussion has been overhauled to remove this material. The BDC and QBO are now described more briefly in the Introduction.**

P19, L385: I do not see this.

**Replaced with, “These complex dynamics may explain why the N₂O AGR in the NH correlates best with lower stratosphere QBO indices, why the correlation has a negative rather than positive sign (due to the vertical reversal of the sign of QBO with altitude) and why the correlations is weaker than in the SH.”**

**(The wording in question was a relic from an earlier version of the paper that looked at specific NOAA surface sites.)**

P19, L398: Why slow? Couldn't also fast descent bring N2O depleted air down? Wouldn't it be that the amount of N2O depleted air would differ dependent on slow or fast descent?

**Deleted “slow”**

P21, L456: What exactly is different in the NH compared to the SH? Wave activity is also high in the SH, so that alone cannot be responsible for the differences in dynamics between the hemispheres.

**Have rewritten as, “In the NH, some of the same mechanisms and interactions between the QBO and BDC occur, but they are more difficult to isolate than in the SH due to the larger wave activity in the NH compared to the SH (Scaife and James, 2000, Kidston et al., 2015). ”**

P21, L473: I am not sure if the differences in dynamics between the southern and northern hemisphere as described in Holton et al. (1995) are still valid. This should be checked. There have been many studies and new findings on atmospheric dynamics in the recent years.

**We are not familiar with a better reference than Holton et al. 1995 to support this point, but would be happy to include additional references if the reviewer has some specific ones in mind.**

P23, L521: The discussion section is much too long, too detailed and too speculative. This section should be significantly shortened and rewritten so that it is more concise. Focus on what is really important and directly related to your study and your results.

**The Discussion has been overhauled and shortened from 2430 words to 1679 words.**

P23, L526: I would not say that the air accumulates there. The air descends during winter in the stratosphere and is then transported to the troposphere.

**Replaced “accumulates” with “descends … into”**

P23, L526: When exactly the air crosses the tropopause has not been shown and also no references are provided. How do you know when and to what amount (e.g. N2O fluxes) N2O is transported down to the troposphere? You get some information on this from the tagged tracer in your model simulation experiment, but that should be clearly discussed here.

**There are no references because this is a concluding statement but references are provided in the Discussion about how signals from the stratosphere are transmitted to the surface in GEOSCCM (Liang et al., 2008, 2009).**

P23, L526: Another question on this text line is, what is happening with N2O on its way? It is not chemically reacting in the troposphere, but the N2O depleted air will mix with the tropospheric N2O rich air on its way that will result in that the stratospheric signal will get less and less as longer N2O is on its way to the surface.

**Have added, "transmitting a diluted but still coherent signal."**

P23, L533: Stratosphere-troposphere exchange and how it is linked to the BDC is not shown and also no references have been provided.

**All references to STE have been removed from the paper and replaced with a more general term, cross-tropopause transport.**

References: The formatting of the references is not correct. See ACP manuscripts/guidelines. Further, there are several citations in the list that are not used, as e.g. Khosrawi et al. (2008) and Glatthor et al. (2007).

**These references have been deleted.**

Technical corrections:

P5, L134: magl -> m AGL

**Done**

P5, L149: four ~ month-long -> ~ four month-long

**Rewrote the description of the ATom campaign with, "ATom consisted of 4 deployments over 3 years, with each deployment approximately 1 month long (*Thompson et al.*, 2022).**

P7, L194: The abbreviation PLST has not been introduced.

**Defined as polar lower stratospheric temperature in the previous paragraph**

P7, L198: Add which months you consider as winter months and which as spring months.

**Added, "Winter months (January-March) were averaged in the NH and spring months (September-November) in the SH…"**

P15, L333: Figure 8 caption: First sentence not correct. Please rephrase.

**Rewrote $N_2O$ in ppb as a function of altitude and latitude from ORCAS (Jan.-Feb. 2016) and ATom-2 (Jan.-Feb. 2017). As in Figure 6, flight track data were interpolated onto a 5 degree latitude by 50 hPa grid using the akima package in R (Akima, 1978) and a deseasonalized fit to the NOAA time series at Mauna Loa was subtracted from all data.**

P15, L334: I guess here it should read ORCAS and not Atom/Atom-2.

**NA after rewording.**

P15, L336-337: Sentence not clear. This needs more explanation. However, this should not be done in the figure caption, but in the main text.

**Moved the interpretative part of the caption to the text.**

P15, L338: Check title, some of the words should be rather starting with a small letter than a capital letter.

**ATom is the correct acronym for Atmospheric Tomography Mission, so the T is intentionally capitalized.**

P17, L367: August-July -> July- August

**This section is describing the 12 month intervals over which the N2O AGR was averaged in order to compare to PLST (which has only 1 annual value). Thus August-July is correct (i.e., beginning in August of a given year and extending through July of the following year. We have shortened and reworded the caption to avoid confusion.**

P22, L505 and 506: The abbreviations OMZ and ETSP have not been introduced.
**They were introduced in the old line 504 immediately above.**

References:

Brühl, C., Steil, B., Stiller, G., Funke, B., and Jöckel, P.: Nitrogen compounds and ozone in the stratosphere: comparison of MIPAS satellite data with the chemistry climate model ECHAM5/MESSy1, Atmos. Chem. Phys., 7, 5585–5598, https://doi.org/10.5194/acp-7-5585-2007, 2007.

Khosrawi, F., Kirner, O., Stiller, G., Höpfner, M., Santee, M. L., Kellmann, S., and Braesicke, P.: Comparison of ECHAM5/MESSy Atmospheric Chemistry (EMAC) simulations of the Arctic winter 2009/2010 and 2010/2011 with Envisat/MIPAS and Aura/MLS observations, Atmos. Chem. Phys., 18, 8873–8892, https://doi.org/10.5194/acp-18-8873-2018, 2018.

Khosrawi, F., Müller, R., Urban, J., Proffitt, M. H., Stiller, G., Kiefer, M., Lossow, S., Kinnison, D., Olschewski, F., Riese, M., and Murtagh, D.: Assessment of the interannual variability and influence of the QBO and upwelling on tracer–tracer distributions of N2O and O3 in the tropical lower stratosphere, Atmos. Chem. Phys., 13, 3619–3641, https://doi.org/10.5194/acp-13-3619-2013, 2013.

**Reviewer 2 Anonymous**
Review of "Northern vs. southern hemisphere differences in the stratospheric influence on variability in tropospheric nitrous oxide" by Nevison et al. This manuscript is a follow-up study of Nevison et al., (2011) published in ACP. In this study, the authors use simulations of nitrous oxide (N2O) from a Chemistry-Climate model (GEOSCCM) together with ground-based and air-borne observations of N2O to evaluate the impact of the stratospheric N2O on its concentrations in the troposphere. Both models and observations agree that N2O-poor air accumulates in the wintertime stratospheric polar regions and moves downward and equatorward, reaching the surface by early-autumn. The authors evaluate the impact of the ENSO, the QBO and the Brewer-Dobson Circulation (BDC, here quantified by the polar lower stratospheric temperatures) on the atmospheric growth rate of the surface N2O. They find that, in the Northern Hemisphere, the surface N2O atmospheric growth rate is negatively correlated with the polar lower stratospheric temperatures of the previous winter, and they attribute this correlation to the BDC. They also find that, in the Southern Hemisphere, the correlation between the surface N2O atmospheric growth rate and the polar lower stratospheric temperatures is not significant, but the correlations between the surface N2O atmospheric growth and the QBO and ENSO are significant. They argue that their findings are consistent with the current literature on the hemispheric differences in stratospheric transport and dynamics. The manuscript presents a coherent analysis and is generally clearly written. The authors use a large number of datasets that makes this comparison really valuable and worth publication in ACP. However, I have the impression that the manuscript needs some adjustments before final publication in ACP. My main concern is that manuscript misses to appropriately highlight "what's new" and does not

provide a possible outlook for future work. Therefore, I recommend publication in ACP once my comments below are addressed.

Major Comments

• Abstract. I suggest adding one/two sentences on "why this study is relevant". As it is, the authors mention right away what they did without providing information on why they did it. Similarly, I feel that the end of the Abstract lacks one/two conclusive sentences that highlight the main message of the paper and possible future paths to continue the work.
**Added, "The literature presents differing views on how variability in surface nitrous oxide (N₂O) is influenced by the stratosphere and whether forcings associated with the El Niño Southern Oscillation (ENSO) outweigh those influences.  These issues, which are relevant to interpreting biogeochemical source signals in tropospheric N₂O, are investigated using…**

• Introduction. The Introduction lacks a paragraph briefly describing the datasets used in the study: the GEOSCCM model and ground-based and air-borne observational datasets. Such paragraph could be added around P3L70. Concerning the novelty of the study, the current paragraph (P3L71-76) could be improved by explicitly stating that his study is a follow-up of Nevison et al., 2011 (this info is available only for reviewers, is there a particular reason?) and that the novelty arises from 12 years of model development and acquisition of additional observations.

**Added a paragraph describing the datasets and another saying, "This paper explores the causes of variability in both the seasonal cycle and the AGR of tropospheric N₂O.  It follows up on previous work by *Nevison et al.* (2011), who inferred a stratospheric influence on surface atmospheric N₂O data based entirely on correlations between interannual variations in stratospheric indices and detrended N₂O anomalies in months surrounding the seasonal N₂O minimum.  In the meantime, altitude-latitude cross sections have become available from aircraft surveys that span a full seasonal cycle.  In addition, advances in model development allow for explicit simulation of stratospheric N₂O tracers (*Ruiz et al.*, 2021; *Liang et al.*, 2022)."**

The Brewer Dobson Circulation (BDC) is mentioned a few times throughout the manuscript, but it is (briefly) described only in the Discussion section (Section 4.1). I think that such an important aspect of the manuscript deserves a few sentences in the Introduction (see also my minor comment about this below).
**These sentences have been added to the Introduction: "The BDC is a planetary-wave-driven, large-scale meridional circulation that transports ozone, greenhouse gases, and other constituents poleward and maintains the thermal structure of the stratosphere (*Butchart*, 2014; *Minganti et al.*, 2020).  As part of this transport, the BDC brings warm, N₂O-poor air from the tropical middle and upper stratosphere into the polar lower stratosphere in the winter hemisphere (*Liang et al.*, 2008; 2009; *Nevison et al.*, 2011)."**
The QBO also is described only in Section 2.4.2 and is hardly mentioned in the Introduction. Again, a few sentences about the QBO should appear in the Introduction (again, see also my minor comment about this below).

**This sentence has been added to the Introduction, "The QBO is a tropical, stratospheric, downward-propagating zonal wind variation with an average period of ~28 months that dominates the variability of tropical lower stratospheric meteorology (*Baldwin et al.*, 2001; *Butchart*, 2014)."**

• Conclusions. The Conclusions lack to (re-)emphasize the novelty of the study and (more importantly) to highlight some strong forward-looking points of conclusions. As it is, the Conclusions reads more like a summary of what has been done.
**Changed the name of Section 5 to "Summary and Conclusions"**

To start, I suggest changing the title of the section from "Conclusions" to "Summary and conclusions". Then, I suggest reducing the repetition of information compared to the previous sections and provide a more detailed summary of what has been done (mention the model, observations, period, methods, a few numbers, …). Finally, I suggest adding some additional points of conclusions, here's a few ideas:
**The reviewer's suggestions are valuable but are beyond the scope of the authors' expertise, which is based more in aircraft data than satellite data. Satellite observations of N2O at least historically have mainly been used at ~ 100 hPa and above but do not resolve the smaller differences in the lower stratosphere and upper troposphere as well as aircraft data. We have therefore added some foreword-looking sentences about the importance of global airborne surveys to the Summary and Conclusion section.**
o The authors find that the surface N2O growth rate presents hemispherical differences in the response to the impact from the QBO (strongest in the SH) and the BDC (strongest in the NH). Minganti et al., (2022) found hemispherical differences in the N2O trends in the stratosphere (positive in the SH and negative in the NH) in satellite observations and reanalyses. I wonder if these hemispherical differences in the stratospheric trends can be related to the differences in the surface N2O growth rate (or just mentioned). o It would be interesting to add one/two sentences on the possible impact of the solar activity on the N2O growth rate. The major chemical destruction of N2O occurs in the tropical upper stratosphere, so I do not expect large impact on the surface growth rate. However, some signal could still reach the troposphere and certainly an additional proxy for solar activity would help to better understand the N2O changes in the stratosphere. o I suggest mentioning the possible added/complementary value of satellite measurements of tropospheric (and stratospheric) N2O (e.g., the IASI, MLS, ACE-FTS instruments). The authors could also mention the use of Chemistry-Transport Models driven by different dynamical reanalyses. Since N2O has a very simple chemistry, such analysis would provide some insights about the dynamics of each reanalysis. In addition, maybe it could be worth mentioning the possible extension to additional CCMs?
o The authors could mention the possibility to perform sensitivity tests with GEOSCCM. For example, an experiment with the QBO switched off (if possible) would isolate the patterns due only to the BDC.
o The authors mention a few times the "stratosphere-troposphere exchange". I suggest dropping that terminology in the manuscript (as it opens a whole research subject) but highlighting the importance of further investigating the cross-tropopause transport/mixing of N2O and its seasonality.
**We have removed the term STE and used the reviewer's suggested term instead.**

• ENSO. I have the feeling that the discussion about the impact of the ENSO is out of scope in this study for two reasons: o First, the title of the manuscript states "stratospheric influence on….". The ENSO is primarily a tropospheric mode of variability that ultimately impacts the stratosphere. Because of that, I do not think that the ENSO belongs in a manuscript that addresses the impact of purely stratospheric processes such as the QBO and the BDC on the surface N2O AGR. o Second, the impact of the ENSO on the N2O AGR is not as direct to understand as for the QBO and the BDC. In Section 4.4, the authors argue that the correlation between N2O AGR and ENSO are mostly due to tropospheric meteorology and biogeochemical processes (first and second paragraphs). In addition, the authors declare that the impact of ENSO on the N2O AGR is hard to evaluate because of possible complicated mechanism difficult to capture (third and fourth). The difficulty to provide a convincing discussion for the impact of the ENSO on the N2O AGR (contrarily to the QBO and BDC where the discussion is satisfying) is a sign that the ENSO should be removed from the manuscript (and maybe mentioned as possible future work in the Conclusions?). Hence, I suggest removing the ENSO parts of the manuscript as it is not a purely stratospheric mode of variability, and its impact of the N2O AGR is difficult to interpret compared to the QBO and BDC. In addition, the removal of the ENSO discussion will imply the removal of Fig. 11 and the related discussion together with Sections 2.4.3 and 4.4. I think that this would be beneficial for the manuscript as it will allow more space for possibly enhancing the discussion about the PLST and the QBO.

**We appreciate this suggestion but are reluctant to eliminate the discussion of ENSO, since one of the main motivations in writing this paper was to address the relative importance of the stratosphere vs. ENSO in determining variability in surface N2O.  The recent IPCC chapter on N2O, for example, emphasizes ENSO as the main driver and we want to present a different opinion.  Instead of eliminating the ENSO section we have shortened other parts of the Discussion and have rewritten the Introduction as well as the Abstract to provide more background and motivation for why we are addressing this issue.**

• Results. The Results section suffer from a flaw that makes the manuscript a little hard to follow: the Figures are never properly introduced. Each Figure is always mentioned between parentheses after one/two sentences describing the results in that figure (e.g., P8L236-238). This is confusing as the reader does not know where to look (and what to look at) when going through the text. I suggest introducing the Figures at the beginning of every paragraph where each Figure is discussed – something like "Figure 1 shows latitude-altitude cross sections of N2O anomalies for GEOSCCM for January to December…".

**Although Reviewer 1 made a similar comment, some of us have been trained since graduate school NOT to present results this way! We were taught that it is better to describe the point made by the figure rather than open the sentence by flatly stating what the figure shows, which in any case should be explained already by the figure caption.  We have tried to compromise by citing the figure number at the beginning of the sentence that first references the figure, including a clause briefly describing what the figure shows, but also including the main point made by the figure in that opening sentence.  On a related note, we have revised our captions to remove all interpretative material in response to another request made by the reviewers.**

Minor Comments

• P1L20 – as mentioned in my major comment above, I suggest starting the Abstract with one/two sentences on why this study is important.

**Keeping in mind ACP's 250 word limit on the abstract we have started with a brief motivatin statement, "The literature presents diverging views on how the stratospheric quasi-biennial oscillation (QBO) influences surface N₂O and has found distinct differences between the northern and southern hemispheres. These issues, which are relevant to the interpretation of biogeochemical source signals in tropospheric N₂O, are investigated…"**

• P1L25 – "… N2O atmospheric growth rate anomaly …" could the authors specify the months when this anomaly occurs?

**Rewrote as, "In the northern hemisphere, the annually averaged surface N₂O atmospheric growth rate anomaly derived from long-term monitoring data is negatively correlated with winter (January-March) polar lower stratospheric temperature."**

• P1L25 – "… winter's polar lower stratospheric temperature." Could the authors specify these winter months? As the seasons could be austral or boreal, I think that specifying the month(s) avoids possible confusion (even when the Hemisphere is mentioned).

**Please see rewording in response to previous comment.**

• P2L41 – I suggest adding the reference to Tian et al., (2020) together with Canadell et al., (2021).

**Added Tian et al., 2020.**

• P2L62 – the QBO here should be briefly explained/introduced. See also my major comment above.

**Added here (moved from Methods), "The QBO is a tropical, stratospheric, downward-propagating zonal wind variation with an average period of ~28 months that dominates the variability of tropical lower stratospheric meteorology (*Baldwin et al.*, 2001; *Butchart*, 2014)."**

• P2L65 – I suggest removing the acronym STE as it is barely mentioned in the rest of the manuscript.

**Removed**

• P3L71 – I suggest adding a paragraph introducing the different datasets used in this study: the GEOSCCM model and the airborne and ground-based observations. See also my major comment above.

**Second to last paragraph in Intro: "This study uses the NASA Goddard GEOS-5 Chemistry-Climate Model (GEOSCCM), which includes a tagged stratospheric N₂O tracer that is transported individually in the model and distinguished from tropospheric tracers of fresh surface emissions (*Liang et al.*, 2022). The study also examines atmospheric N₂O data measured by recent global airborne surveys spanning both hemispheres and collected by the National Oceanic Atmospheric Administration (NOAA) during routine aircraft monitoring in the northern hemisphere. Finally, atmospheric N₂O data from global ground-based NOAA monitoring sites are used in a correlation analysis of the N₂O AGR against ENSO and QBO indices as well as polar lower stratospheric temperature (PLST), which reflects the influence of the BDC."**

• P3L71-76 – This paragraph about the importance/novelty of the study should be improved. I suggest explicitly stating what is the added value of the study and that this is a follow-up of Nevison et al., (2011). See my major comment about this above.

**Added a paragraph to the Introduction: "This paper explores the causes of variability in both the seasonal cycle and the AGR of tropospheric N₂O. It follows up on previous work by *Nevison et al.* (2011), who inferred a stratospheric influence on surface atmospheric N₂O**

**observations based entirely on correlations between interannual variations in stratospheric indices and detrended N$_2$O anomalies in months surrounding the seasonal N$_2$O minimum. In the meantime, altitude-latitude cross sections have become available from aircraft surveys that span a full seasonal cycle. In addition, advances in model development allow for explicit simulation of stratospheric N$_2$O tracers (*Ruiz et al.*, 2021; *Liang et al.*, 2022)."**
• P3L77-87 – This paragraph should describe the structure of the manuscript, but it also introduces the GEOSCCM model (P3L80-82, this belongs to the paragraph describing the datasets mentioned above) and provides some anticipation of the results (P3L82-83 and P3L85-87). I suggest shortening this paragraph by moving the GEOSCCM part to the paragraph describing the datasets and removing the anticipation of the results. Furthermore, I suggest re-phrasing the paragraph like: "Section 2 describes the data and methods used in this work, Section 3 …."

**All results have been removed from the final introductory paragraph, which now read as, "The paper is organized as follows: Section 2 describes the data and methods used. Section 3 presents the results, beginning with a subsection examining climatological mean seasonal cycles and latitude-altitude cross sections of N$_2$O from GEOSCCM and aircraft data. Section 3.2 examines correlations between variability in the N$_2$O AGR from NOAA long-term surface monitoring data, PLST, and indices of QBO and ENSO, with the premise that significant correlations offer evidence of causation. Section 3.3 examines correlations between PLST and variability in monthly N$_2$O anomalies near the month of seasonal minimum. Sections 3.2 and 3.3 include parallel correlation analyses of variabililty in GEOSCCM N$_2$O sampled at NOAA surface sites and model-derived QBO and PLST indices. Section 4 interprets and discusses the results. Section 5 concludes with a summary and conclusion."**

• P3L84 – The ENSO is already defined, no need to repeat the meaning of the acronym.
**Removed**
• P3L90 – The GEOSCCM acronym is already defined.
**Removed**
• P3L93-95 – Could the authors provide more details on the implementation/meaning of these four N2O tracers?
**Rewritten as, "In addition to the standard total N$_2$O tracer, four additional N$_2$O tracers were included to track: 1) aged air from the stratosphere (N2O$_{ST}$), and 2) soil, 3) ocean, and 4) anthropogenic sources freshly emitted in the troposphere. Following the approach of *Liang et al.* (2008), the tropospheric tagged tracers become the tagged stratospheric tracer, N2O$_{ST}$, when they are transported across the tropopause, and retain that identity even when N2O$_{ST}$ re-enters the troposphere, thereby providing a model estimate of the stratospheric influence on tropospheric N$_2$O."**
• P3L96 – "… while a total N2O tracer …." Is this "total N2O" the result of the sum of the four tracers mentioned before (N2Ost, soil, ocean, anthropogenic source)? If yes, I suggest saying it explicitly.
**See above. Yes, effectively total N2O is the sum of the four tracers, but total N2O is also its own explicit tracer.**
• P3L97-98 – "… 2000-2019 …" could the authors mention the spin-up time for this run?
**The full answer to this question is somewhat complicated and described in detail in Liang et al., 2022. We have added a few sentences outlining the details, "The full GEOSCCM**

**simulation spanned 1980-2019, but this study focuses on the final 20 years from 2000-2019… As described in detail in *Liang et al.* (2022), the GEOSCCM N$_2$O lifetime decreased slightly after 2000 (to 116 ± 2 yr in 2010s down from 119 ± 2 yr in the 1990s) and model emissions were gradually increased to keep model atmospheric N$_2$O in balance with observations.**

• P3L98 – "… 5 years of simulation …" why did the authors select the last 5 years of the simulation for the seasonal cycle and not the last 10? A longer period would smooth the internal variability of the model.

**This was an incorrect holdover from Dr. Liang's isotopic work (the isotopic cycles tended to blow up if run too long) and has been removed. In fact, the 20-year simulation from 2000-2019 was used to calculate the mean seasonal cycle of N2O and N2Ost.**

• P4L107 – I suggest changing "(NOAA/HATS) (Thompson et al., 2004)" to "(NOAA/HATS, Thompson et al., 2004)".

**HATS and CCGG have different references. NOAA/HATS and NOAA/CCGG have been changed to HATS (Thompson et al) and CCGG (Lan et al).**

• P4108 – "(NOAA/CCGG)" is it possible to provide a reference here?

**Lan et al. 2022 added.**

• P4L109 – "… 5 baseline sites." I suggest mentioning the location of the sites.

**Added: at Barrow, Alaska; Niwot Ridge, Colorado; Mauna Loa, Hawaii; Cape Grim, Tasmania; and South Pole, Antarctica.**

• P4L115 – I suggest replacing "near" with "approach".

**Done**

• P4L118 – Is it possible to provide a reference instead of (or together with) the link?

**Added Hall et al. 2007 (which is the reference given on the website).**

• P5L137 – What is "Kriging"? I suggest explaining it or re-phrasing.

**The Kriging reference has been deleted – it is not really relevant to this paper.**

• P5L137 – I suggest changing "(Hammerling et al., 2012)" with "Hammerling et al., (2012)".

**Done**

• P5L138 – I suggest replacing "QCLS" (not defined yet) with "airborne".

**Renamed the section as N$_2$O data from global airborne surveys**

• P5L143 – Could the authors provide a reference for the HIPPO project?

**Yes, Wofsy et al., 2011**

• P5L151-152 – "(Note: ….)" I suggest removing this sentence as the Atom-1 deployment is not mentioned further in the manuscript.

**Rewrote the description of the ATom campaign with, "ATom consisted of 4 deployments over 3 years, with each deployment approximately 1 month long (*Thompson et al.*, 2022). QCLS N2O was measured during the second through fourth ATom deployments in January/February 2017, September/October, 2017 and April/May 2018, respectively (*Gonzalez et al.*, 2021). (Note: N2O measurements are not available from the first ATom deployment in July/August 2016 due to technical problems.)" It is relevant to note why ATom 1 N2O data are not available, so we prefer not to delete. Since ATom did not complete a full seasonal cycle for N2O, HIPPO is the only one of the 3 QCLS campaigns to do so, which is why it is feature in Figure 5. This is noted in the text.**

• P6L166-167 – "which was plotted …. As described below." This sentence does not really belong here as it already mentions plotting and proxies. I suggest removing it.

**The sentence was moved to the beginning of Section 2.4.**

• P6L180 – "BDC" the acronym was not introduced before. It should appear in the Introduction (where the BDC should be first described), see my major comment on that above.

**The BDC acronym is now defined in the Introduction**

• P6L183-185 – "(Note: …)" This is a good point and I suggest keeping it as a normal sentence (i.e., without brackets and "Note"). Something like "We highlight that strong local…."

**Moved out of parentheses and reworked into an introductory paragraph describing the variability in N2O.**

• P6L188 – "… the latter…" Do the authors mean the QBO and ENSO indices here? If yes, I suggest specifying it by adding "… the latter two indices…" or something similar.

**Changed to, "The monthly $N_2O$ anomaly correlation analysis was applied only to the PLST BDC proxy, because QBO and ENSO are monthly indices for which it is not straightforward to choose a representative month to correlate to the $N_2O$ anomaly"**

• P7L193-194 – "… in winter/spring … (September-November) in the SH …" I suggest rephrasing this part to something like "… in January-March (winter/spring in the NH) and September-November (spring in the SH) …".

**Done**

• P7L194 – Could the authors provide the meaning of the MERRA-2 acronym?

**Spelled out as the Modern-Era Retrospective Analysis for Research Applications, Version 2 (MERRA-2)**

• P7L195 – "… stratospheric downwelling" I suggest adding "… stratospheric downwelling due to the BDC (e.g., Holton, 2004)."

**Added**

• P7L198 – I suggest adding the months between parentheses after "Winter months" and "spring months". Something like "Winter months (January-March) …. spring months (September-November)". I think it is better to have the explicit months together with the season when discussing the Northern and Southern Hemispheres.

**Done**

• P7L204-205 – This sentence is very pertinent for introducing the QBO. I suggest moving it to the appropriate new paragraph in the Introduction were the QBO and the BDC are first mentioned. See my major comment about this above.

**This sentence was moved to the Introduction.**

• P7L208 – Is it possible to have a reference together with the link?

**This is a product created by co-author P. Newman but does not have a dedicated published reference.**

• P7L215 – The ENSO acronym was already defined before, there's no need to define it again.

**Removed definition**

• P7L216-217 – "El Nino is … Kelvin waves (…)" this sentence should be moved to the Introduction where the ENSO should first be mentioned. However, given my major comment about removing the ENSO discussion, this point can be disregarded.

**Moved to the Introduction with some rewording**

• P8L231-233 – "(Note: ….)" This sentence about ENSO can be removed (if discussion about ENSO is removed) or re-phrased to avoid the "Note:" (if ENSO discussion is kept).

**Rephrased to avoid "Note:"**

• P8L236 – Figure 1 needs to be introduced. See my major comment above.

**We have decided to move this Figure to the Supplement. The old Figure 3 is now Figure 1 and leads off the Results section.**

• P8L236-238 – "GEOSCCM simulates … (SH) (Figure 1)" Could the authors specify the months of winter ("during winter in the polar lower stratosphere") and springtime/early summer ("in springtime (early summer)")?

**Done**

• P8L240 – "the surface minimum … hemispheres" please rephrase with something like "the minimum in the lower troposphere does not reach the surface until …"

**Done**

• P8L240 – please introduce Figure 2.

**Please see our general response to the Major Comment on introducing figures.**

P9, Figure 1 – Since the tropopause is mentioned in the text, it would be nice to show the tropopause level in Figure 1.

**This would be quite difficult to do so we request to be excused from this request. Also, Figure 1 has been moved to the Supplementary Materials.**

• P9, Figure 1 – I suggest providing the units of the N2O anomalies on the right of the colorbar, instead of repeating them at the top right corner of each panel.

**Done.**

• P9, Figure 1, caption – "GEOSCCM N2O anomalies" please specify the units of the N2O anomalies.

**The caption has been changed to, "Figure 1: GEOSCCM monthly $N_2O$ anomalies in ppb as a function of latitude and altitude extending from the surface up to 200 hPa (about 12 km). From left to right: January, February, March, April (top row); May, June, July, August (middle row); September, October, November, December (bottom row)."**

• P9, Figure 1, caption – please replace "in a monthly sequence" with "as a function of". • P9, Figure 1, caption – "up to 30 hPa (24 km)" you state that you show from the surface to 30 hPa (24 km), but the top level of the panels of Figure 1 is 200 hPa (corresponding to 12 km). Could the authors explain that?

**See rewording of caption above.**

• P9, Figure 1, caption – please insert something like "From left to right: January, February, March, April (top row); May, … (middle row); September, … (bottom row)".

**Done**

• P9, Figure 1, caption – "The GEOSCCM N2O …. the anomalies" this sentence belongs more to the text. I suggest moving it to the part describing Figure 1.

**See rewording of caption above and note below about moving Mauna Loa detrending to the methods Section 2.1.**

• P9, Figure 1, caption – "Mauna Loa" the authors never mentioned Mauna Loa before in the text. I suggest moving this part to the text and mention the Mauna Loa sampling in the Methods section. There are other occurrences of the Mauna Loa sampling in the caption of other figures, therefore I suggest moving the Mauna Loa sampling description into the appropriate Methods section and not repeating it again.

**We have included this statement in Section 2.1 and removed it from the Figure 1 caption: "The GEOSCCM $N_2O$ fields were saved as monthly means and were detrended and converted to anomalies by subtracting a deseasonalized fit to the model time series sampled at Mauna Loa."**

• P9, Figure 2 – Also here, please provide the units on the right of the colorbar.
**Done**
• P9, Figure 2 – the bottom right panel (50S) has an additional label on the y axis (1000) that is not present in the other panels. I suggest removing it.
**Removed**
• P9, Figure 2 caption – please replace "N2O anomalies vs month" with something like "N2O anomalies as a function of month and altitude".
**Done**
• P9, Figure 2 caption – please rephrase "in the northern (top row) and southern (bottom row) hemispheres" with "in the NH (top row) and in the SH (bottom row)".
**Done**
• P9, Figure 2 caption – please specify the meaning of the columns. Something like: "From left to right: 70°, 60° and 50°".
**Done**
• P9, Figure 2 caption – "The GEOSCCM N2O fields … relative to the NH" I suggest moving (and appropriately re-phrasing to fit the context) this sentence to the text where Figure 2 is introduced.
**Moved into the text, "In Figure 2, the SH panels are plotted with a 6 month shift to help visualize the earlier seasonal phasing of the stratospheric influence in the NH relative to the SH."**
• P10L252 – please introduce Figure 3.
**We have moved the old Figure 3 to Figure 1 because it most clearly illustrates that N2Ost dominates the total N2O seasonal cycle at most surface sites in GEOSCCM.**
• P10L252 – I suggest replacing "stratospheric loss" with "N2O-depleted air".
**Done**
• P10L254 – I suggest adding "for all sites" after "N2Ost minimum".
**Added, "… minimum at all extratropical sites" (MLO and SMO are exceptions).**
• P10L255 – I suggest replacing "Southern Hemisphere" with "SH".
**Done**
• P10L256 – I suggest replacing "northern latitudes" with "latitudes in the NH".
**Done**
• P10, Figure 3 – I suggest swapping the panels for Summit (70N) and Barrow (71N). Barrow is northern than Summit and it would be consistent with the current layout of the panels (north to south from top left to bottom right).
**Done**
• P10, Figure 3 – I suggest removing the IDs of the stations on the bottom left corner of the panels (alt, sum, brw, …) because the names of the sites are already in the titles of the panels.
**Done**
• P10, Figure 3 caption – please mention the names of the sites after "… 9 surface sites". Something like "Top row from left to right: Alert (Canada), …; middle row from left to right: Mace Head (Ireland), ….; bottom row from left to right: Cape Grim (Country?), ….".
**Done**
• P10, Figure 3 caption – "The black heavy … observed N2O" should come before "the red line …".
**Done**
• P10, Figure 3 caption – I suggest re-phrasing "The red line … tracer N2Ost" with something like "For GEOSCCM, the total N2O from all forcings is in red, and the stratospheric tracer N2Ost is in dashed red".

**Done**
• P11L264 – please introduce Figure 4.
**Please see our general response to the Major Comment on introducing figures.**
• P11L264-265 – "When viewed as a … up to 8 km" This sentence (properly re-phrased to correctly describe a figure) should be included in the description of Fig. 4.
**Done**
• P11L266 – please replace "is felt in" with something like "reaches".
**Done**
• P11L268 – again, please introduce Fig. 5 here.
**Please see our general response to the Major Comment on introducing figures.**
• P11, Figure 4 caption – as for Figures 1 and 2, the authors mention Mauna Loa in the caption with no reference in the text. I suggest moving (and expanding) this part to the Methods section.
**Moved to Methods**
• P12L285 – please introduce Figure 6.
**Please see our general response to the Major Comment on introducing figures.**
• P12L289 – I suggest removing "April".
**Done**
• P12L289 – "By June it has descended…" I find difficult to see that the anomaly has descended into the troposphere by June. I suggest re-phrasing with something like "By June it starts descending into …".
**Done**
• P12L291-292 – "Notably, … (Supplementary Figure 1)" this sentence states that the fuller dataset does not provide results as clear as the subset shown in Fig. 6. Do the authors know why? It would be interesting to provide a few words of explanation.
**The QCLS data represent snapshots that contain synoptic features (e.g., plumes of strong surface sources) that may obscure the stratospheric depletion signal, which as Reviewer 1 points out, is small < 1ppb against a 330 ppb background.**
• P13, Figure 6 – The August panel of Figure 6 (for the NH) does not agree well with the corresponding August panel of Figure 4 (considering the same latitudes and altitudes). Do the authors know why? Maybe because the periods are different?
**This is addressed now in the Discussion. The color scales are different, since HIPPO extends into the lower stratosphere where the depletion signals are stronger. HIPPO also represents a synoptic snapshot north-south across a single longitude. In contrast, the NOAA empirical background data are averaged as monthly means across 170W-50W.**
• P13, Figure 6 caption – could the authors explain the meaning of "transects"?
**A transect according to the dictionary is a "line that cuts through a natural landscape (in the case of the figure by latitude) so that standardized observations and measurements can be made. In the case of the HIPPO and other QCLS data, it is not a line but rather a 2-dimensional plan spanning both latitude and altitude. We have added, "Each panel represents a north-to-south transect across with vertical profiling from the surface to 14 km."**
• P13, Figure 6 caption – "…. an annual sequence." I suggest saying explicitly what contains each panel. Something like "an annual sequence; from left to right: January, March, June (top row) and August, November (bottom row)."
**Done**

• P13, Figure 6 caption – "HIPPO data extend … influence on tropospheric N2O" This sentence (properly re-phrased to blend into the text) belongs to the text in the introduction of Figure 6.

• P13L301 – please introduce Figure 7.
**Please see our general response to the Major Comment on introducing figures.**
• P13L301 – "(PLST)" PLST is already defined – no need to define it again here.
**Since most readers are probably not familiar with PLST as an acronym, and since it hasn't been mentioned since Section 2, it seems better to reiterate what it stands for rather than risk confusing readers.**
• P13L303 – "seasonal minimum" I suggest adding the reference to the specific panel of the Figure (Fig. 7a).
**Done**
• P13L304 – I suggest removing the IDs of the stations (CGO, PSA, SPO).
**Done**
• P13L304 – I suggest changing "(Figure 7)" to something like "(Fig. 7b,c)".
**Done**
• P13L308 – I suggest changing "(Figure 7)" to something like "(Fig. 7e,f)".
**Done**
• P13L308 – "In support of …." before this sentence please introduce Figure 8.
**Please see our general response to the Major Comment on introducing figures.**
• P13L308 – P14L313 – In Figure 8, the authors compare different observational datasets (Atom and ORCAS) for different periods (2016 and 2017). In my opinion, this makes the discussion difficult to follow. I suggest using only the ORCAS dataset for Figure 8. This would allow more room for discussion about the ORCAS dataset (maybe separating January and February?) and remove the asymmetry in Figure 8.
• P14L312 – I suggest changing "(weak Brewer Dobson circulation)" with "(weak BDC)".
**Done**
• P14L313 – I suggest changing "(strong Brewer Dobson circulation)" with "(strong BDC)".
**Done**
• P14L315-319 – this paragraph refers to generally unsignificant correlations for a Figure in the Supplement. I suggest reducing the importance of this paragraph to one/two sentences and move it to the end of the discussion of Figure 7.
**Moved to the end of the section and trimmed.**
• P14, Figure 7 – I suggest removing the bottom left panel (GEOS ΔN2O vs GEOS Strat T) for two reasons: 1) it is not explicitly mentioned in the text and 2) it shows N2Otot from GEOSCCM that is not the most relevant tracer for this comparison.
**Figure 7 has been reduced to 4 panels as per the request of a different reviewer and the text now explicitly mentions N2Ost, which is relevant because it suggests that the correlation shown is due to the influence of PLST on N2Ost and not fortuitously caused by surface surface sources.**
• P15, Figure 7 caption "The correlation between … its seasonal minimum" I suggest moving this sentence to the text.
**Done**
• P15L339 – please introduce Figure 9.
**Please see our general response to Major Comment on introducing figures.**

• P15L339 – P18L371 – Discussion of Figures 9 and 10. As it is, the discussion forces the reader to swap back and forth between Fig. 9 and Fig. 10 to follow the structure of the text (currently describing first the QBO in the SH and NH and then the PLST in the SH and NH). I suggest keeping the structure of the text as it is and re-arranging the figures accordingly: Fig. 9 should contain the QBO for both SH and NH, and Fig. 10 the PLST for SH and NH.

**We have restructured the text to first discuss the SH, both QBO and PLST correlations, referencing Figure 9 (now 8) and then the NH, referencing Figure 10 (now 9).**

• P16L353 – please remove "of the weak".

**Done**

• P16L353 – please replace "correlations" with "correlation".

**Done**

• P17L364 – please add "(Fig. 9b)" after "NOAA".

**Done**

• P17L365 – please replace "(Figure 9b,d)" with "(Fig. 9d)".

**Done**

• P18L373-375 – based on my previous comment of the ENSO part, this discussion (and Figure 11) should be removed. However, if the authors decide to keep it, please introduce Figure 11.

**Please see our general response to Major Comment on introducing figures.**

• P18L382 – "vertical profiles" there is no vertical profile in the manuscript (for me vertical profiles are line plots with the values on the x axis and the altitude/pressure on the y axis). I suggest replacing "vertical profiles" with something like "latitude-pressure cross sections".

**Done**

• P18L382 – please re-phrase "big picture" with something like "broad".

**Broad-scale**

• P19L387 – please replace "Brewer Dobson circulation" with "BDC".

**Done**

• P19L394-395 – "which transports warm … in the winter hemisphere" The impact of the BDC is not limited to the transport of N2O-poor air: the BDC also transports ozone, GHGS and CFCs and maintains the thermal structure of the stratosphere (Butchart, 2014, Minganti et al., 2020). I suggest re-phrasing this sentence to include also these important effects. In addition, based on my major comment above, the BDC needs to be described in the Introduction, so I suggest moving this description to the appropriate part in the Introduction.

**Rewritten as, "The mechanistic pathway by which the stratosphere imparts a distinct seasonal signature to surface $N_2O$ is linked to the Brewer Dobson circulation (BDC), which transports ozone, greenhouse gases, and other constituents downward and poleward and maintains the thermal structure of the stratosphere (*Butchart*, 2014; *Minganti et al.*, 2020). As part of this transport, the BDC brings warm, $N_2O$-poor air from the tropical middle and upper stratosphere into the polar lower stratosphere in the winter hemisphere (*Liang et al.*, 2008; 2009; *Nevison et al.*, 2011)."**

• P19L394 – please replace "N2O-depleted" with "N2O-poor".

**Done throughout**

• P19L396 – "a large seasonal amplitude in the polar lower stratosphere" could the authors specify the amplitude of what?

**Rewritten as, "…leads to a large seasonal amplitude in the $N_2O$ mixing ratio in the polar lower stratosphere"**

• P19L405 – I suggest replacing "upwells" with something like "is transported".

**Done**

• P19L410 – I suggest replacing "(~32 km) (Strahan et al., 2021 ….)" with "(~32 km, Strahan et al., 2021 ….)".

**Rewritten as, "…transports more N$_2$O to its peak loss region around 32 km (*Strahan et al.*, 2021; *Ruiz et al.*, 2021)."**

• P20L423 – I suggest replacing "in the July-September period" with "in July-September".

**Done**

• P20L430 – I suggest removing the quotes from surf zone. • P20L433 – I suggest removing "relatively".

**Done**

• P20L436 – also here, I suggest removing "relatively".

**Done**

• P20L436 – I suggest replacing "is felt at" with "reaches the".

**Done**

• P20L437 – I suggest replacing "permits" with "results in".

**Done**

• P21L456 – I suggest being more specific and replace "more complex atmospheric dynamics in the NH stratosphere" with something like "larger wave activity in the NH compared to the SH (Scaife and James, 2000, Kidston et al., 2015)".

**Done**

• P21L461 – "mixing of air" I suggest adding a reference here. For example, Shepherd, 2007.

**Done**

• P21L466 – I suggest removing the link because the text between parentheses is not a citation.

**Done**

• P21L474 – I suggest removing "stratosphere-troposphere exchange" as it was barely mentioned before in the manuscript. I suggest replacing it with something like "crosstropopause transport".

**Done**

• P22L496 – I suggest replacing "heightened" with "increased".

**Done**

• P23L516 – I suggest re-phrasing "testament to the strength" with something like "demonstration of the strength".

**Rewritten as "demonstrated the strength"**

• P23L522 – "Conclusions". As it is, the Conclusions section reads more like a summary than a Conclusion. Because of that, I suggest renaming this section as "Summary and Conclusions". In addition, the Conclusions section needs to re-emphasize what is new in this study and address more points of conclusion and possible future work. More on this in my major comment above.

**We have added some foreward looking lines to the conclusions, renamed as the Summary and Conclusions, "To further refine our understanding of variability in tropospheric N$_2$O, long-term monitoring at surface and aircraft-based sites is essential and would be complemented by more global airborne surveys extending into the lower stratosphere. The latter provide new insights into stratospheric influences on tropospheric N$_2$O and advance our ability to interpret and quantify surface N$_2$O sources."**

• P23L527 – "summer to early-autumn period" could the authors specify the months here?

**Replaced with, "late summer to early autumn (August-September in the NH, April-May in the SH)."**

• P23L530 – I suggest replacing "delivered" with "transported".

**Done**
• P23L532-533 – "Stratosphere-troposphere exchange". As one of my comments on this before, I suggest replacing "Stratosphere-troposphere exchange" with something like "Crosstropopause transport".
**Done**

References

Holton, J.: An Introduction to Dynamic Meteorology, no. v. 1 in An Introduction to Dynamic Meteorology, Elsevier Science, available at: https://books.google.be/books?id=fhW5oDv3EPsC (last access: 28 October 2020), 2004.

Kidston, J., Scaife, A., Hardiman, S. et al. Stratospheric influence on tropospheric jet streams, storm tracks and surface weather. Nature Geosci 8, 433–440 (2015). https://doi.org/10.1038/ngeo2424

Minganti, D., Chabrillat, S., Christophe, Y., Errera, Q., Abalos, M., Prignon, M., Kinnison, D. E., and Mahieu, E.: Climatological impact of the Brewer–Dobson circulation on the N2O budget in WACCM, a chemical reanalysis and a CTM driven by four dynamical reanalyses, Atmos. Chem. Phys., 20, 12609– 12631, https://doi.org/10.5194/acp-20-12609-2020, 2020.

Minganti, D., Chabrillat, S., Errera, Q., Prignon, M., Kinnison, D. E., Garcia, R. R., et al. (2022). Evaluation of the N2O rate of change to understand the stratospheric Brewer-Dobson circulation in a Chemistry-Climate Model. Journal of Geophysical Research: Atmospheres, 127, e2021JD036390. https://doi.org/10.1029/2021JD036390

Scaife, A.A. and James, I.N. (2000), Response of the stratosphere to interannual variability of tropospheric planetary waves. Q.J.R. Meteorol. Soc., 126: 275- 297. https://doi.org/10.1002/qj.49712656214

Shepherd, T. G.: Transport in the middle atmosphere, J. Meteorol. Soc. Jpn. Ser. II, 85, 165–191, 2007. Tian, H., Xu, R., Canadell, J.G. et al. A comprehensive quantification of global nitrous oxide sources and sinks. Nature 586, 248\u2013256 (2020). https://doi.org/10.1038/s41586-020-2780-0

Reviewer 3 Anonymous

This study investigates the seasonal and interannual variability of N2O at the surface driven by transport from the stratosphere. This is a follow up of previous work by the first author on this topic with the addition of model output and aircraft data. The strength of the study is the use of aircraft data to fill in the seasonal transport picture from the tropopause region to the surface. Analysis of this type is important to more fully understand the causes of the variability of surface N2O and subsequently the variability of emission sources. But the manuscript needs to be revised to be more readable, primarily by focusing the discussion on the main points. Some specific comments are included below.

Specific comments

Section 2.1: You mention that the 'temperature and QBO are both internally generated by the GEOS GCM'. I assume that means this is a free-running simulation? That is, not forced by a reanalysis meteorology? It would be helpful to clarify this point. The model run is referenced to the Liang et al., 2022 study which likely clarifies the details of the run but it would be nice to have just a bit more information here.

**Added, "GEOSCCM temperature and QBO do not necessarily correspond to observations since both are internally generated by the GEOS GCM, which is free-running rather than forced by a reanalysis meteorololgy."**

Section 3: Some of the figures could be consolidated. For instance, Figure 7b and c as well as 7e and f are so similar that it doesn't seem necessary to show them both.

**We have reduced Figure 7 (now Figure 9) to 4 panels.**

In Figure 8 it's hard to see a clear overall difference between these aircraft data sets. In some regions the ORCAS data seems lower. Maybe a difference plot between the two years would be easier to interpret and more concise.

**The discussion has been expanded in Section 4.0 to acknowledge that, "QCLS data are measured across a narrow longitude band of the flight track for any given latitude on a limited number of days, while the NOAA empirical background is shown as a monthly mean, zonally averaged across most of the western hemisphere (170°-50°W). Consequently, QCLS data are more likely to display synoptic-scale variability, such as the apparent surface source plumes over the Southern Ocean seen in Figure 10." (We have retained the original figure rather than a difference plot because it is easier to visualize than a difference plot, especially given the different spatial domains of ORCAS and ATom.)**

There are also some figures that could use more discussion such as Figure 5. The contrast between the positive summer anomalies in the lower troposphere and the observed negative anomalies seems significant. Where is the positive anomaly in the model coming from? In general, only cursory mention of the differences between modeled and observed features are made rather than any insight into why the model differs.

**The second paragraph in the discussion now addresses this, "the seasonality of**

**surface N$_2$O emissions may not be well represented in the GEOSCCM simulation, e.g., summer soil emissions may be overestimated, leading to unrealistic surface maxima in July (*Liang et al.*, 2022)."**

Section 4: This section is too lengthy and repetitive. Since there aren't really any figures that focus on the stratosphere, aside from averaged metrics or the lowermost stratosphere, the discussion here asks a lot of the reader to follow the descriptions of stratospheric processes and is essentially a summary of previous work anyway. Some of the discussion in the section could be incorporated into Section 3 where the figures are discussed. But at a minimum this section should be considerably shortened to focus on the main findings, which as the title suggests is the NH vs. SH differences.

**The discussion of the QBO and Brewer-Dobson circulation has been greatly shortened and summarized. The bulk of the original discussion was moved to a section in the Supplementary Materials for those interested in reading the details.**

Line 520: The mention of a multivariate correlation that can explain more of the N2O variance almost seems like a throwaway here since it's just before the conclusions and there is no previous mention of it. Yet this seems like a promising result and worth more exploration or more discussion if the analysis has already been performed for the observations and model output.

**These sentences were deleted, since we agree that there is not enough room to do the topic justice, especially as other reviewers are recommending that the discussion of ENSO be removed entirely.**

---

## Referee Report (RR1)

**Review on ACP Manuscript No acp-2023-2877 Revised**

Observational and model evidence for a prominent stratospheric influence on variability in tropospheric nitrous oxide by Nevison et al.

The manuscript has significantly improved and I appreciate the effort the authors have put in shortening and better structuring their paper. I have only some minor comments that are left and should be considered before publication.

Abstract: The abstract is much better now, but I still have some issues with it. For example the introductory two sentence you provide are somewhat independent from what you write in the next few sentences. To understand this connection the reader needs to be either an expert on the topic or have to read the paper first.

Some thoughts on the abstract:

- I would suggest to make the transition at line 25 a bit smoother. I think my major problem here is that you as motivation mention ENSO, but then discuss the results you get from the model without mentioning ENSO again until you come to the results concerning the correlations.
- Isn't the point here that you investigate the stratospheric influence on the seasonal cycle? Or are you investigating all processes that influence N2O cycle? This did not come really across in the abstract.
- ENSO is a tropical circulation, but then you discuss the influence of BDC and polar descent on the seasonal cycle how do then these processes fit together?
- My suggestion for the abstract would be: 1. Introductory sentence, 2. Data/model that are used, 3. There are hemispheric differences and then provide at the end your results and then the closing sentences.

**Specific comments:**

P1, L22: What forcing?

P1, L23: Which issues?

P1, L28: What is meant with similar cycles?

P2, L41: delete "ozone-depleting substance" since to my knowledge O3 is not directly reacting with N2O, but due to the conversion of N2O to the photolysis, ozone is destroyed by the resulting products and thus it is indirectly depleting ozone. If you would like to keep the sentence as is I would suggest to add the rereference of the paper by Ravinshankara et al. in Science ( https://www.science.org/doi/10.1126/science.1176985).

P4, L90: Here it is not clear from your sentence if Ruiz et al. (2021) did not find such an influence in the SH or if they did not investigate the SH. Thus, I would suggest to rephrase the sentence so that this becomes more clear.

P5, L118: add "can be" so that it reads "can be distinguished from tropospheric tracers…….."

P5, L122: Is the aircraft data only used/available for the NH? Thus, SH solely based on model data? Please state more clearly what data has been used for which hemisphere.

P5, L124: reflects? I would rather write "is used as tracer for".

P6, L160: Again I have to ask why Mauna Loa is used. You provide an answer in your reply to my comments, but you have not added a reasoning here. I would suggest to add here a short explanation why you picked Mauna Loa and not one of the other stations.

P7, L178: you mean "approach the end of their lifetime" or are these instruments just getting old? In the latter case I would rather write "as the instruments are aging" than "as the instruments approach their lifetime".

P7, L187: Listed above? Which ones? Are these the same ones as for NOAA? Please rephrase the sentence to be more clear with which stations you mean.

P7, L208: See my comment a tP6, L160. Again here you mention that Mauna Loa has been used for detrending the data, but without given a reason. You should at some point in the manuscript provide one sentence why Mauna Loa is used and not another station.

P14, L336: What one can clearly see from the cross sections is the downward transport of the air. But I have difficulties to see horizontal transport and mixing. Since this are known transport processes I would suggest to add here some adequate references.

P17, Figure 5: Why do you use for the aircraft data a different color scheme than for the other datasets?

P19, Figure 6: Not for all panels the p value has been added.

P29, 618ff: You are not really providing here a summary at least not in the sense what was the aim of the study and what has been done. Further, the order as you discuss the results in your summary and conclusion is somewhat weird. Why do you start with the aircraft data instead with the model and NOAA data which are the main data sets of your study? Further, the discussed results cannot only be derived with aircraft data. This can be achieved with other measurement data sets as well.

Additional comment: Concerning my comment on P21, L473 of the previous version of your manuscript concerning if the reference of the text book by Holton (1995) is still valid for the hemispheric differences in the BDC. First of all, here you should rather cite the 2$^{nd}$ edition of the text book published in 2006 or check the following papers by Garny et al. (2013), Butchart et al. (2014) or Fu et al. (2019).

**Technical corrections:**

P7, L191: Empirical background -> empirical background

P8, L211: Abbreviation HIPPO not introduced. It's done on L219, but this should appear at the first instance where the abbreviation is used.

P8, L223: Abbreviation ORCAS has not been introduced.

P17, L390: "an annual sequence" appears twice. One is thus obsolete.

P17, L395: Add here the section number.

P19, L425: Since all the correlations you consider are rather week I would suggest to omit the term "strongest". I would rather use the term "highest". Further, when the correlation is negative you should either clearly state that this correlation is negative or call it an anticorrelation.

P20, Figure 7: Also here in some of the panels the p-value is not given.

P21, Figure 8: The grey lines are hardly visible on a printout version of your manuscript. Please use a somewhat darker grey for these lines.

P25, L515: The reference Khosrawi et al. (2009) is missing in the reference list. Instead you still have there the Khosrawi et al, (2013) reference which is actually not cited.

P26, L541: Didn't you state before that the strongest correlation for the QBO is found at 50 hPa? Please check the numbers and levels if everything is correct and consistent discussed.

P29, L617: "Summary and" should also be in bold face.

P31, L665: Check the formatting of the references. Indents for the consecutive lines of each reference are missing and different style for the references is used. This should be done in a uniform style and according to the ACP guidelines.

P33, L754: Reference Khosrawi et al. (2013) appears twice, but has not been cited in the manuscript. Further, Khosrawi et al. (2009) which has been cited in the manuscript is not listed in the reference list.

Note: Figure should appear as Fig. in the text, except at the begin of a sentence (see ACP manuscript preparation guidelines).

Supplement, 2nd page, 2nd paragraph: What do you mean with "a year prior"? Do you mean "a prior year"?

Supplement: Figure captions -> remove Supplement before S1, S2, and S3.

**References:**

Butchart N., The Brewer-Dobson circulation (2014), Reviews of Geophysics, 52 (2), pp. 157 - 184, DOI: 10.1002/2013RG000448.

Garny H., Bodeker G.E., Smale D., Dameris M., Grewe V. (2013), Drivers of hemispheric differences in return dates of mid-latitude stratospheric ozone to historical levels, Atmospheric Chemistry and Physics, 13 (15), pp. 7279 - 7300, DOI: 10.5194/acp-13-7279-2013.

Fu Q., Solomon S., Pahlavan H.A., Lin P. (2019), Observed changes in Brewer-Dobson circulation for 1980-2018, Environmental Research Letters, 14 (11), art. no. 114026,  DOI: 10.1088/1748-9326/ab4de7

---

## Referee Report (RR2)

**Review of Nevison et al. – round 2**

I appreciate the thorough work of the authors: the quality of the manuscript is highly improved, and the results are clearer and the reading smoother. I have the feeling now that my problem with the ENSO discussion was the lack of a strong introduction and motivation.

However, a few of my comments were only partially answered or not discussed at all, and the revised version of the manuscript brought to the surface an issue concerning the discussion about the impact of the BDC. In addition, I have some minor/technical corrections that should be addressed.

I recommend publications after the points below are addressed.

**Partially answered/unanswered comments.**

In the following, I copy my comments of the previous review in italic.

- Conclusions. The authors highlight the relevance of airborne measurements for the current study in comparison with satellite data and that is a perfectly fair point. However, the authors do not provide any comments regarding the following possible points of discussion that I raised in the first review (pasted below). If the authors decide to disregard these suggestions, I would be interested to know why.
    - *The authors find that the surface N2O growth rate presents hemispherical differences in the response to the impact from the QBO (strongest in the SH) and the BDC (strongest in the NH). Minganti et al., (2022) found hemispherical differences in the N2O trends in the stratosphere (positive in the SH and negative in the NH) in satellite observations and reanalyses. I wonder if these hemispherical differences in the stratospheric trends can be related to the differences in the surface N2O growth rate (or just mentioned).*
    - *It would be interesting to add one/two sentences on the possible impact of the solar activity on the N2O growth rate. The major chemical destruction of N2O occurs in the tropical upper stratosphere, so I do not expect large impact on the surface growth rate. However, some signal could still reach the troposphere and certainly an additional proxy for solar activity would help to better understand the N2O changes in the stratosphere.*
    - *The authors could mention the possibility to perform sensitivity tests with GEOSCCM. For example, an experiment with the QBO switched off (if possible) would isolate the patterns due only to the BDC.*
- Results. I appreciate the compromise of the authors, but Figures 7 and 8 do not seem to meet this compromise (respectively, P19L421 and P20L439).
- *In Figure 8, the authors compare different observational datasets (Atom and ORCAS) for different periods (2016 and 2017). In my opinion, this makes the discussion difficult to follow. I suggest using only the ORCAS dataset for Figure 8. This would allow more room for discussion about the ORCAS dataset (maybe separating January and February?) and remove the asymmetry in Figure 8.*
Can the authors clarify on why this comment was not answered?

**Discussion related to the BDC impact.**

P1L32. "warm". I think the "warm" here (and throughout the manuscript) arises from some confusion. The N2O-poor stratospheric air that is transported by the BDC over that region is not necessarily warm. The downwelling (i.e., the downward transport) due to the BDC at high

latitudes during winter warms up the lower stratosphere because it's a diabatic process. Because of that, the PLST is used as a proxy of the strength of the BDC, i.e., warmer PLST indicate stronger downwelling due to stronger BDC.

In a nutshell, the BDC does not bring "warm air" to the lower stratospheric high latitudes, the air over that region is warm because of the BDC (e.g., Holton et al., 1995).

This comment does not change the conclusions of the authors: warmer PLST indeed indicate stronger BDC, but I suggest removing the "warm" before "N2O-poor air" throughout the manuscript as it might generate confusion.

**Minor/technical comments.**

- P1L15. Izana -> Izaña.
- P1L24. I suggest removing "atmospheric".
- P1L32. circulation -> Circulation (throughout the manuscript).
- P2L42. I suggest removing "(GWP)" as the abbreviation is not used further in the manuscript.
- P2L50. I suggest changing "ppb" with "ppbv" throughout the manuscript.
- P2L52. N is not defined her, but it is defined at the line below (L53). I suggest moving "nitrogen (N)" to L52.
- P3L61-65. "While larger … (Nevison et al., 2018)." I suggest re-phrasing this long sentence into two sentences separated by a period.
- P3L75-76. "… phases in the eastern tropical Pacific (ETP.)". I suggest being more specific here and mention sea surface temperature – something like: "… phases in sea temperatures over the eastern tropical Pacific (ETP)".
- P3L82. "northern hemisphere" and "southern hemisphere" are already defined.
- P4L90. I suggest changing "dynamics" with "dynamical processes".
- P4L113. "altitude-latitude cross sections" I suggest specifying that you are talking about measurements here.
- P5L130-131. "with the premise …. of causation". This sentence belongs more to the Methods section. Also, could you provide your reasoning (or some reference) regarding why correlation is evidence of causation for this case?
- P6L154. I suggest swapping "116+-2" with "119+-2" since the authors highlight that the lifetime is decreasing after 2000.
- P6L158. "they". Do the authors refer to the QBO and temperature here? If yes, could they specify it?
- P6L166. (HATS) (Thompson et al., 2004) -> (HATS, Thompson et al., 2004).
- P6L167. (CCGG) (Lan et al., 2022) -> (CCGG, Lan et al., 2022)
- P7L174. Is the "X2006A" necessary here? It sounds strange to someone not familiar with this terminology like me.
- P7L182. Is 13 a subset of the ~55 sites mentioned before? If yes, could the authors specify it?
- P8L211. "HIPPO" not yet defined here.
- P8L219. "HIAPER" not yet defined here.
- P8L223. "ORCAS" not yet defined here.
- Section 2.3.2. This section contains only one paragraph and I suggest merging it with Section 2.3.1 to retain only Section 2.3.
- P10L265. PLST is already defined here.
- P10L269-270 "PLST reflects …. (Holton, 2004)". This sentence belongs more to the Introduction where the authors first talk about the PLST.
- P11L295. "0.4 degrees C" -> "0.4 degrees °C".

- Figure 1 caption. I suggest changing "…atmospheric N2O modeled…." with "… N2O mixing ratios [ppbv] modeled …". For the captions of Figures 2, 3 and 4, I suggest something like "Anomalies of N2O mixing ratios [ppbv] ….".
- P15L364. I suggest adding "in the NH" after "altitude-latitude contours".
- Figure 5 caption. "… pressure-latitude contour plots arranged to form ….". I suggest re-phrasing with something like "… pressure-latitude contours of anomalies of N2O mixing ratios [ppbv] to form ….".
- P17L393-394. Given that the manuscript has shortened and become clearer, I would remove the additional definition of PLST here.
- P17L395. I suggest adding "(Section 2)" after "Methods".
- P17L397. Again, why do the authors assume that significant correlation between N2O AGR and one index implies causal influence of that index on the N2O AGR?
- Figure 6. I suggest swapping panels 6b and 6c. This way, the discussion can focus on the QBO first and then on the PLST (i.e., first discuss 6a and 6b, then 6c and 6d).
- Figure 6 caption. Can the authors specify the units of the AGR in the caption?
- P19L421. I suggest introducing Figure 7 as was done for the previous figures.
- Figure 7. I suggest re-arranging the discussion of Figure 7 so it would smoothly describe 7a, 7b, 7c and 7d. In alternative, the authors could keep the discussion as it is and re-arrange the panels in Figure 7 accordingly (panels *a* and *b* for temperature and *c* and *d* for QBO). I find the current discussion of Figure 7 (7b, 7d, 7a, 7c) rather counter-intuitive.
- Figures 7 and 8 captions. As for Figure 6, I suggest specifying the units of the AGR in the caption.
- P20L439. Also here, I suggest introducing Figure 8 as for the previous figures.
- P20L440. I suggest re-phrasing "…with little to no …".
- P21L450. I suggest re-phrasing "… of the seasonal cycle …. NOAA sites." with "… of the seasonal cycle for NOAA sites at remote mid and high latitude."
- P22L458. "January". Why do the authors mention January if panel 9b and its caption say February?
- P22L464. Similarly to the comment above, why do the authors mention March if panels 9c,d and their captions say February?
- Figure 9 caption. I suggest re-phrasing " a) … mean seasonal cycle in N2O …" with "a) … mean seasonal cycles in N2O mixing ratios [ppbv] for the NOAA surface station observations (Obs), and the GEOSCCM total N2O (N2Otot) and stratospheric N2O (N2Ost) …".
  Also, I suggest re-phrasing "b) NOAA surface N2O seasonal anomalies … " with "b) NOAA surface seasonal anomalies of N2O mixing ratios [ppbv] ….".
  I also suggest a similar re-phrasing for "Bottom row shows seasonal anomalies for …." with "Bottom row shows seasonal anomalies of N2O mixing ratios [ppbv] for …".
- P23L476. The authors mention "February" but the ORCAS dataset also covers January.
- Figure 10 caption. I suggest changing "N2O anomalies in ppb …." with "Anomalies of N2O mixing ratios [ppbv] ….".
- P24L490-494. I feel that this paragraph gives too much importance to a figure that is not shown in the main manuscript. I suggest reducing this paragraph to a sentence that captures its essence.
- P25L500. I suggest replacing "shows up" with "enters".
- P25L512. I suggest re-phrasing "…simulates too long a delay …" with "… simulates a too long delay ….".
- P25L513. "The rate of descent". Can the authors specify what is descending?

- P25L517. "… may be overestimated" why do the authors think that the summer soil emission might be overestimated? Was that a conclusion of Liang et al., 2022?
- P26L534. Given my comment above about the warming due to the BDC, I suggest removing "warm".
- P26L539-540. I suggest moving the reference to Ray et al. 2020 at the end of the sentence and put it between parentheses (Ray et al., 2020).
- P27L562. "Paradoxically". Why do they authors say that? Did they expect something different?
- Section 4.2. Very interesting section. However, I think it can be summarized and merged with Section 4.1 to highlight their main points. When doing that, I suggest clearly separating the discussion between the SH and NH (you could even have a subsection for the SH and one for the NH).
- P28L589. I suggest removing "ENSO".
- P29L611. I suggest re-phrasing "tease out".
- All figures. Please replace the occurrences of "ppb" with "ppbv".

---

## Author Response (AR2)

We thank the reviewers and Dr. **Šácha** for their thorough and detailed reviews, which were very helpful in encouraging us to reorganize, clarify, and revise our manuscript. Please find the reviewers' comments below in plain text and our responses in bold font.

RNDr. Petr Šácha, Ph.D.
Editor

Reviewer 1 - Dr. Farahnaz Khosrawi, after commenting on the significant improvements in the revised version, goes on to recommend acceptance after minor revisions. Still, Dr. Khosrawi highlights few outstanding minor issues, which are partly editorial in nature, but also lists some scientific critiques under the Specific comments. I urge you to take advantage of the editorial suggestions and to carefully reply to the scientific comments. Based on my own reading, I would like to add to the specific comment P19 Fig. 6 by Dr. Khosrawi on statistical significance a point, that the reader needs to see additional statistical information also around Figs. 1, 2, 3, 4 and 9a. Please add information here on the spread behind the mean seasonal cycles and on the representativeness of the mean monthly anomalies, considering also the uncertainties of the measurements where applicable.

**We have added uncertainty on the mean seasonal cycles of GEOSCCM and NOAA data shown in Figure 1 and 9a. The method is explained in Section 2.3. "Mean seasonal cycles for NOAA surface $N_2O$ observations and GEOSCCM $N_2O$ tracers were estimated using a bootstrapping method in which 20% of the timeseries was randomly removed and the remaining 80% was fit to a $3^{rd}$ order polynomial plus first 4 harmonics. These steps were repeated over 500 iterations to estimate the range of uncertainty in the harmonic components of the fit." This a a common approach for estimating uncertainty in NOAA data. The std deviation of multiple flask measurements for each NOAA surface N2O data point is reported but is generally similar throughout the time series for a given site. We therefore did not apply weights (e.g., the inverse square of the stddev) to each data point when computing the harmonic fits.**

**For the GEOSCCM contour plots in Figures 2-3 we apologize that we do not have an obvious way to estimate the uncertainty, nor was this done in the related publication Liang et al. 2022, which focused wholly on GEOSCCM and presented similar contour plots. However, we have emphasized that we are showing these plots to qualitatively illustrate the influence of the stratosphere, but it is beyond the scope of this paper to quantitatively calculate that influence. Similarly, we have used the NOAA empirical background qualitatively to illustrate the stratospheric influence on the troposphere. In addition, in Figure 4 we have expanded the number of years used to compute the climatology to 2005-2013 (previously we used 2009-2013). We also show for the editor's benefit below that the panels show a similar pattern regardless of which 5 year segment is used.**

NOAA Empirical Background anomalies of N2O mixing ratio

[Figure]

Finally, Dr. Farahnaz Khosrawi has graciously offered a list of technical corrections which are necessary to improve the understanding, and therefore the impact, of what you are trying to say. We should both thank the reviewer for this as it improves the final product.

**We have added to the Acknowledgements, "The authors thank Dr. Farahnaz Khosrawi and 2 other anonymous reviewers whose detailed and helpful comments much improved the manuscript."**

Recommendation for minor revisions is echoed also by Reviewer 2, who appears to remain less convinced about the quality and significance of your manuscript. In spite of this the reviewer first makes it clear that the manuscript has been improved. Importantly, the reviewer then brings to our attention that some of their original remarks have not been addressed. This includes possible important points as the solar cycle effect and relation to stratospheric N2O studies. I second the reviewer's request that all of their comments from the previous round need to be properly addressed. The reviewer also graciously offers a long list of minor and technical comments and suggestions, which we should again thank the reviewer very much for, because they are driven with the intention to improve the manuscript on all fronts.
**We appreciate the time that Dr. Khosrawi and Reviewer 2 have taken to provide very detailed and helpful comments, both here and in their previous reviews, and have done our best to respond to them.**

Finally, I would like to add one additional comment based on my own reading and shear interest in your study. It seems to me that the positive anomalies shown across the figures are never discussed in the manuscript (except in the Supplement).
Do you assume the positive anomalies to be a poor consequence of your methodology and the existence of negative anomalies?

**We have focused on the negative anomalies because they are more likely to come from the stratosphere, which is the main interest of this paper. However, we have added a paragraph in the Results discussing Figure 3. "The positive anomalies in Fig. 3 also differ between model and observations, with the summer-dominant soil source assumed in GEOSCCM appearing as surface contours at 40° and 50°N, while the NOAA empirical background shows positive anomalies at the surface in late winter and spring, likely reflecting North American agricultural sources (*Nevison et al.*, 2018). At 60° and 70°N, the stronger contrast between positive and negative anomalies in GEOSCCM compared to NOAA throughout the atmospheric column reflects the model's larger seasonal cycle, as seen also in Fig. 1**."

Especially based on Figs. 5 and 10 I would argue that the positive anomalies can stand for some physical process omitted in the discussion. For example, in Fig. 10 for 2017 (ATom2) except the existence of the positive anomaly centered around 55°S seemingly stemming from the tropics, the negative anomalies can be locally seen stronger than in 2016, which contradicts your discussion around this figure and the link with BDC you are trying to draw. **We acknowledge in the presentation of Fig 10 in the Results that, "However, the positive anomaly in the mid-troposphere observed at 40-60˚S during ATom-2, which may be a source plume from the Southern Ocean, tends to contradict the hypothesis that SH tropospheric N₂O was lower overall in 2017 than in 2016."**

**We also discuss the limits of the QCLS data in the 3ʳᵈ paragraph of the Discussion, including that, "QCLS data are measured across a narrow longitude band of the flight track for any given latitude on a limited number of days" and that "QCLS data are more likely to display synoptic-scale variability**, **such as the apparent surface source plume over the Southern Ocean seen in Fig. 10.."**

Observational and model evidence for a prominent stratospheric influence on variability in tropospheric nitrous oxide by Nevison et al.

The manuscript has significantly improved and I appreciate the effort the authors have put in shortening and better structuring their paper. I have only some minor comments that are left and should be considered before publication.

Abstract: The abstract is much better now, but I still have some issues with it. For example the introductory two sentence you provide are somewhat independent from what you write in the next few sentences. To understand this connection the reader needs to be either an expert on the topic or have to read the paper first.

Some thoughts on the abstract:

- I would suggest to make the transition at line 25 a bit smoother. I think my major problem here is that you as motivation mention ENSO, but then discuss the results you get from the model without mentioning ENSO again until you come to the results concerning the correlations.
- Isn't the point here that you investigate the stratospheric influence on the seasonal cycle? Or are you investigating all processes that influence N2O cycle? This did not come really across in the abstract.
- ENSO is a tropical circulation, but then you discuss the influence of BDC and polar descent on the seasonal cycle how do then these processes fit together?
- My suggestion for the abstract would be: 1. Introductory sentence, 2. Data/model that are used, 3. There are hemispheric differences and then provide at the end your results and then the closing sentences.

   **Abstract rewritten to accommodate the suggestions above, "The literature presents different views on how the stratosphere influences variability in surface nitrous oxide (N₂O) and on whether that influence is outweighed by surface emission changes driven by the El Niño Southern Oscillation (ENSO). These questions are investigated using a chemistry-climate model with a stratospheric N₂O tracer, surface and aircraft-based N₂O measurements, and indices for ENSO, polar lower stratospheric temperature (PLST), and the stratospheric quasi-biennial oscillation (QBO). The model simulates well-defined seasonal cycles in tropospheric N₂O that are caused mainly by the seasonal descent of N₂O-poor stratospheric air in polar regions with subsequent cross-tropopause transport and**

mixing.  Similar seasonal cycles are identified in recently available N₂O data from aircraft. A correlation analysis between the N₂O atmospheric growth rate (AGR) anomaly in long-term surface monitoring data and the ENSO, PLST, and QBO indices reveals hemispheric differences.  In the northern hemisphere, the surface N₂O AGR is negatively correlated with winter (January-March) PLST. This correlation is consistent with an influence from the Brewer Dobson Circulation, which brings N₂O-poor air from the middle and upper stratosphere into the lower stratosphere, with associated warming due to diabatic descent. In the southern hemisphere, the N₂O AGR is better correlated to ENSO and QBO indices. These different hemispheric influences on the N₂O AGR are consistent with known atmospheric dynamics and the complex interaction of the QBO with the Brewer Dobson Circulation.  More airborne surveys extending to the tropopause, would help elucidate the stratospheric influence** on tropospheric N₂O, allowing for better understanding of surface sources.”

**Specific comments:**

P1, L22: What forcing?  **This term is no longer used in the rewritten abstract -see above.**
P1, L23: Which issues?  **We now use the word “questions” which refer directly to the previous sentence, i.e., the “how” and “whether” clauses.**
P1, L28: What is meant with similar cycles? **Have specified “similar *seasonal* cycles.”**

P2, L41: delete “ozone-depleting substance” since to my knowledge O3 is not directly reacting with N2O, but due to the conversion of N2O to the photolysis, ozone is destroyed by the resulting products and thus it is indirectly depleting ozone. If you would like to keep the sentence as is I would suggest to add the rereference of the paper by Ravinshankara et al. in Science ( https://www.science.org/doi/10.1126/science.1176985). **We have removed the ODS reference from the opening sentence, since it is discussed in a later sentence with mention of NOx as the actual catalyst of O3 destruction.  We have also added the Ravishankara et al. 2009 reference to that latter sentence.**

P4, L90: Here it is not clear from your sentence if Ruiz et al. (2021) did not find such an influence in the SH or if they did not investigate the SH. Thus, I would suggest to rephrase the sentence so that this becomes more clear. **Reworded as, “*Ruiz et al.* (2021) found a direct correlation between the QBO and N₂O photochemical loss rates in the tropical middle stratosphere but concluded that interannual variability in surface N₂O globally was governed more indirectly by QBO-related changes in the dynamical processes of the lowermost stratosphere.  They showed evidence for a coherent influence of cross-tropopause transport on the surface N₂O seasonal cycle in the NH but not the SH.”   Note: Ruiz et al. upon rereading are a bit ambiguous but seem to be referring to global patterns (see their supplementary Fig. S3). However, their methodology is complex, and involves creating a 28 month “composite QBO signal “in chemistry transport models and comparing to surface N2O from NOAA.  They found that QBO-related loss rates of F11 and N2O are only weakly correlated in the tropical stratosphere but that the QBO composite surface signal of F11 and N2O are nearly 100% correlated, hence their conclusion.  Their finding of a strong influence due to cross-tropopause transport in the NH but not the SH refers to the mean seasonal cycle rather than IAV.**

P5, L118: add “can be” so that it reads “can be distinguished from tropospheric tracers........”
**Done**

P5, L122: Is the aircraft data only used/available for the NH? Thus, SH solely based on model data? Please state more clearly what data has been used for which hemisphere. **We have rewritten the sentence as, "The study also examines atmospheric N$_2$O data collected in the NH by the National Oceanic Atmospheric Administration (NOAA) during routine aircraft monitoring, as well as N$_2$O data measured by recent global airborne surveys spanning both hemispheres." This reordering clarifies that the NOAA data are in the NH only but that the airborne surveys (referring to HIPPO, ORCAS, ATom) span both hemispheres. (It is a somewhat lengthy process to spell out all the acronyms and describe the QCLS instrument, so that text is deferred to Section 2.2.3 in the methods.)**

P5, L124: reflects? I would rather write "is used as tracer for". **Substituted, "is used as a tracer for"**

P6, L160: Again I have to ask why Mauna Loa is used. You provide an answer in your reply to my comments, but you have not added a reasoning here. I would suggest to add here a short explanation why you picked Mauna Loa and not one of the other stations. **Added, "The N$_2$O time series at MLO is a good proxy for the global N$_2$O trend and thus its subtraction provides a convenient, single-station approach for calculating anomalies of the N$_2$O mixing ratio for contour plots." (Note: by "convenient, single-station" we mean that the reason to use one station and not a combination of many is that it's a lot easier to extract matching values and make the calculations for model output as well as observations.)**

P7, L178: you mean "approach the end of their lifetime" or are these instruments just getting old? In the latter case I would rather write "as the instruments are aging" than "as the instruments approach their lifetime". **Replaced with "as the instruments age."**

P7, L187: Listed above? Which ones? Are these the same ones as for NOAA? Please rephrase the sentence to be more clear with which stations you mean. **The 5 HATS sites are named in the preceding paragraph. We modified the next paragraph to read, "This study used the NOAA combined HATS/CCGG N$_2$O product from 1998-2021, which is based on monthly medians from the CATS *in situ* program (at the 5 HATS baseline sites) and monthly means from the CCGG flask program at a selected subset of 12 of the ~55 total sites (https://doi.org/10.15138/GMZ7-2Q16; *Hall et al.*, 2007). All of the NOAA sites considered in this study are long-standing remote sites situated away from strong local anthropogenic sources. They include Alert, Canada; Summit, Greenland; Mace Head, Ireland; Trinidad Head, California, Cape Kumakahi, Hawaii, Cape Matatula, Samoa; Palmer Station, Antarctica, and the 5 HATS baseline sites (at which CCGG also makes overlapping flask measurements). In addition to these 12 individual sites, global, NH and SH means are estimated from the latitude-binned and mass-weighted means of the combined monthly means for the 12 sites (*Hall et al.*, 2011). The combined monthly data are first aggregated at the measurement program level for each sampling location. At sites where both HATS and CCGG measure, a weighted mean is calculated based on the programs' monthly uncertainties."**

P7, L208: See my comment a tP6, L160. Again here you mention that Mauna Loa has been used for detrending the data, but without given a reason. You should at some point in the manuscript provide one sentence why Mauna Loa is used and not another station. **See response above re: MLO.**

P14, L336: What one can clearly see from the cross sections is the downward transport of the air. But I have difficulties to see horizontal transport and mixing. Since this are known transport processes I would suggest to add here some adequate references. **We have added the references latitudes (*Liang et al.*, 2009; 2022), which describe horizontal transport and mixing for GEOSCCM.**

P17, Figure 5: Why do you use for the aircraft data a different color scheme than for the other datasets? **The aircraft panel figures were created by Dr. Stephens while the other panels were created by Dr. Nevison using different software.**

P19, Figure 6: Not for all panels the p value has been added.

**We have added p values for all the AGR plots which involved correlating monthly N2O AGR to monthly QBO and ENSO indices. These correlations required special treatment because they have a variable N due to the autocorrelation that is introduced in part by the 12-month running mean used to deseasonalize $N_2O$ to compute the AGR. To account for autocorrelation in each time series, we used an effective N ($N_{eff} = N/$ ), which is described in the new section S2 in the Supplementary Materials.**

P29, 618ff: You are not really providing here a summary at least not in the sense what was the aim of the study and what has been done. Further, the order as you discuss the results in your summary and conclusion is somewhat weird. Why do you start with the aircraft data instead with the model and NOAA data which are the main data sets of your study? Further, the discussed results cannot only be derived with aircraft data. This can be achieved with other measurement data sets as well.

**We have reversed the order of the first 2 sentence, presenting the GEOSCCM results first, then the aircraft results. In the penultimate sentence, we have changed "To further refine" to "To help refine" which doesn't exclude other measurement data sets (e.g., from satellites) but focuses on summarizing the data sets presented in the current study and how our analysis can be improved going forward.**

Additional comment: Concerning my comment on P21, L473 of the previous version of your manuscript concerning if the reference of the text book by Holton (1995) is still valid for the hemispheric differences in the BDC. First of all, here you should rather cite the 2[nd] edition of the text book published in 2006 or check the following papers by Garny et al. (2013), Butchart et al. (2014) or Fu et al. (2019).

**For the discussion of the NH vs. SH difference in Section 4.2 we believe that Holton 1995 (which is a paper in Reviews of Geophysics), while old, is still a good reference for the assertion that the BDC is stronger in the NH than the SH (see, e.g., their Table 1). We have cited it along with Scaife and James, 2000 and Kidston et al., 2015 and Butchart 2014. We have additionally cited the Holton et al. 2004 textbook earlier in Section 2.4.1 as a one of the references describing why we use PLST as a proxy for the strength of the BDC. We appreciate the Garny, and Fu references but our reading suggests that those papers are focused mainly on recent changes in the strength of the BDC rather than the fundamental NH vs. SH difference. However, Butchart 2014 mentions the stronger BDC in the NH so we have cited it.**

**Technical corrections:**

P7, L191: Empirical background -> empirical background **Changed to lower case**

P8, L211: Abbreviation HIPPO not introduced. It's done on L219, but this should appear at the first instance where the abbreviation is used. **Changed HIPPO to "first of the airborne surveys described below" and spelled out High-performance Instrumented Airborne Platform for Environmental Research (HIAPER) Pole to Pole Observations (HIPPO) in the preceding section 2.2.3 describing the airborne surveys.**

P8, L223: Abbreviation ORCAS has not been introduced. **O₂/N₂ Ratio and CO₂ Airborne Southern Ocean (ORCAS)**

P17, L390: "an annual sequence" appears twice. One is thus obsolete. **Deleted extra "annual sequence"**

P17, L395: Add here the section number. **Added Methods (Section 2).**

P19, L425: Since all the correlations you consider are rather week I would suggest to omit the term "strongest". I would rather use the term "highest". Further, when the correlation is negative you should either clearly state that this correlation is negative or call it an anticorrelation. **Rewritten, "In contrast to the SH, the NOAA surface N₂O AGR in the NH is anticorrelated significantly to winter PLST (R = -0.67), with an optimal correlation for the 12-month period from July-June encompassing the January-March PLST average (Fig. 7b). A similar anticorrelation is found between the GEOSCCM PLST and NH N₂O AGR (Fig. 7d). Also in contrast to the SH, the NOAA NH N₂O AGR is correlated only weakly to the QBO index at all altitudes, with a negative sign. The highest correlation in the NH occurs for 50 hPa QBO (R= -0.23, p>0.1) with a 10-14 months lag (Fig. 7a). GEOSCCM also predicts an anticorrelation (R = -0.47, p < 0.05) between the GEOSCCM QBO and the NH N₂O AGR, which also is optimal around 50 hPa with 10-14 month QBO lag (Fig. 7c)."**

P20, Figure 7: Also here in some of the panels the p-value is not given. **We have added p values to Fig. 7 – see response above for Fig.6.**

P21, Figure 8: The grey lines are hardly visible on a printout version of your manuscript. Please use a somewhat darker grey for these lines. **We have used a darker grey for the ENSO lines.**

P25, L515: The reference Khosrawi et al. (2009) is missing in the reference list. Instead you still have there the Khosrawi et al, (2013) reference which is actually not cited. **We have removed Khosrawi et al. 2013 and added Khosrawi et al. 2009 to the References.**

P26, L541: Didn't you state before that the strongest correlation for the QBO is found at 50 hPa? Please check the numbers and levels if everything is correct and consistent discussed. **We have clarified by adding "… consistent with Ray et al. (2020) (who only presented results for QBO = 50 hPa)." In other words, we are pointing out that our lag time is consistent with Ray et al. 2020 at 50 hPa, which was the only pressure at which they considered the QBO. (We considered the QBO index at a range of pressures from 100 hPa to 10 hPa.)**

P29, L617: "Summary and" should also be in bold face. **Done**

P31, L665: Check the formatting of the references. Indents for the consecutive lines of each reference are missing and different style for the references is used. This should be done in a uniform style and according to the ACP guidelines. **We have indented the references and put them in ACP format.**

P33, L754: Reference Khosrawi et al. (2013) appears twice, but has not been cited in the manuscript. Further, Khosrawi et al. (2009) which has been cited in the manuscript is not listed in the reference list. **We have removed Khosrawi et al. 2013 and added Khosrawi et al. 2009 to the References.**

Note: Figure should appear as Fig. in the text, except at the begin of a sentence (see ACP manuscript preparation guidelines). **Changed Figure to Fig. throughout the text.**

Supplement, 2nd page, 2nd paragraph: What do you mean with "a year prior"? Do you mean "a prior year"? **Changed to "one year earlier."**

Supplement: Figure captions -> remove Supplement before S1, S2, and S3. **Done**

**References:**

Butchart N., The Brewer-Dobson circulation (2014), Reviews of Geophysics, 52 (2), pp. 157 - 184, DOI: 10.1002/2013RG000448.

Fu Q., Solomon S., Pahlavan H.A., Lin P.: Observed changes in Brewer-Dobson circulation for 1980-2018, Environmental Research Letters, 14 (11), 114026, DOI: 10.1088/1748- 9326/ab4de7, 2019.

Garny H., Bodeker G.E., Smale D., Dameris M., Grewe V.: Drivers of hemispheric differences in return dates of mid-latitude stratospheric ozone to historical levels, Atmospheric Chemistry and Physics, 13 (15),7279 - 7300, DOI: 10.5194/acp-13-7279-2013, 2013.

**Reviewer 2**

I appreciate the thorough work of the authors: the quality of the manuscript is highly improved, and the results are clearer and the reading smoother. I have the feeling now that my problem with the ENSO discussion was the lack of a strong introduction and motivation.

However, a few of my comments were only partially answered or not discussed at all, and the revised version of the manuscript brought to the surface an issue concerning the discussion about the impact of the BDC. In addition, I have some minor/technical corrections that should be addressed.

I recommend publications after the points below are addressed.

**Partially answered/unanswered comments.**

In the following, I copy my comments of the previous review in italic.

• Conclusions. The authors highlight the relevance of airborne measurements for the current study in comparison with satellite data and that is a perfectly fair point. However, the authors do not provide any comments regarding the following possible points of discussion that I raised in the first review (pasted below). If the authors decide to disregard these suggestions, I would be interested to know why.

o *TheauthorsfindthatthesurfaceN2Ogrowthratepresentshemisphericaldifferences in the response to the impact from the QBO (strongest in the SH) and the BDC (strongest in the NH). Minganti et al., (2022) found hemispherical differences in the N2O trends in the stratosphere (positive in the SH and negative in the NH) in satellite observations and reanalyses. I wonder if these hemispherical differences in the stratospheric trends can be related to the differences in the surface N2O growth rate (or just mentioned).*

**We have changed the Summary and Conclusions to Summary and Outlook. We include these sentences in the Outlook portion, "Another important issue for future research is the impact on N₂O of climate change driven increases in the strength of the BDC (*Garny et al.*, 2013; *Butchart et al.*, 2014; *Fu et al.*, 2019). Of particular relevance to the results presented here are studies based on**

ground or satellite-based N$_2$O observations that find a decrease in the N$_2$O lifetime (*Prather et al.*, 2023) and interhemispheric differences in stratospheric N$_2$O trends (*Minganti et al.*, 2022)."

o *Itwouldbeinterestingtoaddone/twosentencesonthepossibleimpactofthesolar activity on the N2O growth rate. The major chemical destruction of N2O occurs in the tropical upper stratosphere, so I do not expect large impact on the surface growth rate. However, some signal could still reach the troposphere and certainly an additional proxy for solar activity would help to better understand the N2O changes in the stratosphere.*

We have also added to the Summary and Outlook, **"The solar cycle is an additional influence on variability in N$_2$O that may be worth investigating in future studies. While previous studies have estimated a relatively small effect over the 2000s and 2010s due to low solar activity (Ruiz et al., 2021; Prather et al., 2023), solar cycle-driven changes in the UV flux affect N2O photolysis both directly and indirectly through the impact stratospheric O3."**

o *TheauthorscouldmentionthepossibilitytoperformsensitivitytestswithGEOSCCM. For example, an experiment with the QBO switched off (if possible) would isolate the patterns due only to the BDC.*

**We appreciate the suggestion but feel that it is beyond the scope of our study. While it is possible to switch off QBO in the GEOSCCM model simulation as a sensitivity study, it is computationally expensive to do so. And since GEOS is free running GCM, switching off QBO will likely lead to other changes that complicate the interpretation.**

- Results. I appreciate the compromise of the authors, but Figures 7 and 8 do not seem to meet this compromise (respectively, P19L421 and P20L439). **We have rewritten these paragraphs to introduce Fig. 7 and 8 by first describing what they show and then presenting the salient results. (We also did this for Fig. 6.)**
- *In Figure 8, the authors compare different observational datasets (Atom and ORCAS) for different periods (2016 and 2017). In my opinion, this makes the discussion difficult to follow. I suggest using only the ORCAS dataset for Figure 8. This would allow more room for discussion about the ORCAS dataset (maybe separating January and February?) and remove the asymmetry in Figure 8.*
  Can the authors clarify on why this comment was not answered?

**We apologize for overlooking this comment. We did however respond to a related comment in the first review that requested that the ORCAS and ATom panels in Figure 8 be plotted in the same manner. "Thus, the left panel should also be plotted from -70 S to 80 N, but masking the parts of the data that are not considered as white area."**
**Our response then and now is that this would leave a lot of blank space on the ORCAs panel. We've added a sentence to remind readers that ORCAS focused on the Southern Ocean and was restricted to the extratropical SH, which is why there is an asymmetry in the two panels. The purpose of Fig. 8 is to provide a comparison of aircraft data in the extratropical SH in two successive years with opposite extremes in PLST and notable differences in their Feb. seasonal anomalies (as shown in Fig 9). While the comparison is qualitative and is complicated/undermined by synoptic scale features in the ATom panel, which we now acknowledge, we believe that this plot is important and relevant to our paper.**

**Discussion related to the BDC impact.**

P1L32. "warm". I think the "warm" here (and throughout the manuscript) arises from some confusion. The N2O-poor stratospheric air that is transported by the BDC over that region is not necessarily warm. The downwelling (i.e., the downward transport) due to the BDC at high latitudes during winter warms up the lower stratosphere because it's a diabatic process. Because of that, the PLST is used as a proxy of the strength of the BDC, i.e., warmer PLST indicate stronger downwelling due to stronger BDC.

In a nutshell, the BDC does not bring "warm air" to the lower stratospheric high latitudes, the air over that region is warm because of the BDC (e.g., Holton et al., 1995).

**Thank you for this correction. We rewrote as, "The mean PLST in each hemisphere was treated as a proxy for the integrated strength of the BDC, which brings N₂O-poor air from the middle to upper tropical stratosphere into the polar winter lower stratosphere through diabatic descent. This warms up the lower stratosphere, with warmer PLST corresponding to stronger descent."**

This comment does not change the conclusions of the authors: warmer PLST indeed indicate stronger BDC, but I suggest removing the "warm" before "N2O-poor air" throughout the manuscript as it might generate confusion.

**Minor/technical comments.**

- P1L15. Izana -> Izaña. **Done**
- P1L24. I suggest removing "atmospheric". **Done**
- P1L32. circulation -> Circulation (throughout the manuscript). **Done**
- P2L42. I suggest removing "(GWP)" as the abbreviation is not used further in the manuscript. **Done**
- P2L50. I suggest changing "ppb" with "ppbv" throughout the manuscript.

**Our NOAA and QCLS co-authors Dr. Lan and Dr. Kort encounter this issue frequently but are firm that ppbv should NOT be used in place of ppb. Briefly, NOAA and QCLS are not measuring or reporting the N2O mole fraction in ppbv. The mole fraction they report is defined as the number of molecules of N2O in any given air parcel divided by the total number of all molecules (except water) in that parcel. For N2O it is usually expressed as parts per billion, abbreviated as ppb. To make values in mole fraction (in ppb) the same as those in ppbv, one would have to assume ideal gas law works, which is not the case in reality, and Dr. Lan and Dr. Kort don't make such approximations during their measurements or reporting. For more details please see section 7 of https://gml.noaa.gov/aftp/data/trace_gases/n2o/flask/surface/README_n2o_surface-flask_ccgg.html And also the WMO guidelines https://library.wmo.int/records/item/57135-20th-wmo-iaea-meeting-on-carbon-dioxide-other-greenhouse-gases-and-related-measurement-techniques-ggmt-2019**

- P2L52. N is not defined her, but it is defined at the line below (L53). I suggest moving

  "nitrogen (N)" to L52. **Done**

- P3L61-65. "While larger ... (Nevison et al., 2018)." I suggest re-phrasing this long sentence

  into two sentences separated by a period. **Rewritten, "Larger spatial and seasonal signals in atmospheric N₂O have been observed at sites influenced by strong local agricultural or**

coastal upwelling sources (*Lueker et al.*, 2003; *Nevison et al.*, 2018; *Ganesan et al.*, 2020). However, at sites remote from local sources even variations of 0.2 ppb in estimated background $N_2O$ levels can significantly affect the magnitude of $N_2O$ emissions inferred from atmospheric inversions (*Nevison et al.*, 2018)."

- P3L75-76. "... phases in the eastern tropical Pacific (ETP.)". I suggest being more specific here

  and mention sea surface temperature – something like: "... phases in sea temperatures over

  the eastern tropical Pacific (ETP)". **Done**

- P3L82. "northern hemisphere" and "southern hemisphere" are already defined. **Thank you for catching this, replaced here with "NH and SH."**
- P4L90. I suggest changing "dynamics" with "dynamical processes". **Done**
- P4L113. "altitude-latitude cross sections" I suggest specifying that you are talking about

  measurements here. **Added "observed altitude…"**

- P5L130-131. "with the premise .... of causation". This sentence belongs more to the Methods

  section. Also, could you provide your reasoning (or some reference) regarding why

  correlation is evidence of causation for this case? **Rewritten as, "…with the assumption that significant correlations offer support, although not proof, of causation" We elaborate on this statement in Section 3.2 (see below) but include it briefly here in response to earlier reviewer comments that requested a clearer blueprint in the Introduction of why the correlation analysis was included.**

- P6L154. I suggest swapping "116+-2" with "119+-2" since the authors highlight that the

  lifetime is decreasing after 2000. **Replaced with "from 119 ± 2 yr in the 1990s down to 116 ± 2 yr in the 2010s"**

- P6L158. "they". Do the authors refer to the QBO and temperature here? If yes, could they

  specify it? **Yes, replaced with, "However, GEOSCCM QBO and PLST were computed in the same way as the observed indices…"**

- P6L166. (HATS) (Thompson et al., 2004) -> (HATS, Thompson et al., 2004). **Done**
- P6L167. (CCGG) (Lan et al., 2022) -> (CCGG, Lan et al., 2022) **Done**
- P7L174. Is the "X2006A" necessary here? It sounds strange to someone not familiar with this

  terminology like me. **Our NOAA co-author and lab expert Dr. Lan prefers to keep the name of the WMO scale, since it is meaningful to experimentalists.**

- P7L182. Is 13 a subset of the ~55 sites mentioned before? If yes, could the authors specify it? **Added, "at a selected subset of 13 of the ~55 total sites"**
- P8L211. "HIPPO" not yet defined here. **"the (HIAPER) Pole to Pole Observations (HIPPO) project" (HIPPO is an acronym within an acronym, taking its HI from HIAPER)**

- P8L219. "HIAPER" not yet defined here. **"the High-performance Instrumented Airborne Platform for Environmental Research (HIAPER)"**
- P8L223. "ORCAS" not yet defined here. **$O_2/N_2$ Ratio and $CO_2$ Airborne Southern Ocean (ORCAS)**
- Section 2.3.2. This section contains only one paragraph and I suggest merging it with Section

  2.3.1 to retain only Section 2.3. **We have eliminated the subheadings 2.3.1 and 2.3.2 and relabeled 2.3 as Interannual variability in surface $N_2O$ for the correlation analysis.**

- P10L265. PLST is already defined here. **Replaced with "plotted against PLST as described below."**
- P10L269-270 "PLST reflects .... (Holton, 2004)". This sentence belongs more to the

  Introduction where the authors first talk about the PLST**. Consolidated and rewrote as "The mean PLST in each hemisphere was treated as a proxy for the integrated strength of the BDC, which brings $N_2O$-poor air from the middle to upper tropical stratosphere into the polar winter lower stratosphere through diabatic descent.  PLST represents the cumulative effect of descent throughout fall and winter, with warmer PLST corresponding to stronger descent (*Holton*, 2004; *Nevison et al.*, 2007; 2011)."  We introduce the BDC in the Introduction but feel that these sentences are more appropriate here because we are describing why PLST is used as a proxy  for the BDC.  To go into this level of detail in the Introduction would potentially confuse and distract readers.  Note, with this rewording we are also addressing the reviewer's request to remove references to the BDC bringing warm air into the lower stratosphere.**

- P11L295. "0.4 degrees C" -> "0.4 degrees °C".  **Replaced**

- Figure 1 caption. I suggest changing "...atmospheric N2O modeled...." with "... N2O mixing ratios [ppbv] modeled ...". For the captions of Figures 2, 3 and 4, I suggest something like "Anomalies of N2O mixing ratios [ppbv] ....". **Done for Figs 1-4 captions.**
- P15L364. I suggest adding "in the NH" after "altitude-latitude contours". **Done**
- Figure 5 caption. "... pressure-latitude contour plots arranged to form ....". I suggest re-

  phrasing with something like "... pressure-latitude contours of anomalies of N2O mixing

  ratios [ppbv] to form ....".  **Rewrote, "Sequence of five HIPPO pressure-latitude contours of anomalies of $N_2O$ mixing ratio (ppb)…"**

- P17L393-394. Given that the manuscript has shortened and become clearer, I would remove

  the additional definition of PLST here. **We can remove it if necessary, but given that it will be an unfamiliar acronym for many and this is its first mention in the Results, we would prefer to err on the side of repeating it here.**

- P17L395. I suggest adding "(Section 2)" after "Methods".  **Done**
- P17L397. Again, why do the authors assume that significant correlation between N2O AGR

  and one index implies causal influence of that index on the N2O AGR? **We have reiterated from the Introduction that correlations have limited usefulness and don't prove causation, but do provide reasonable support for causation, especially in the context of other evidence**

**from GEOSCCM and aircraft observations. "a significant correlation between the interannual variability in the N₂O AGR and one or more of the indices can be interpreted to support a causal influence of the latter on the N₂O AGR. However, correlation does not prove causation and cannot distinguish possible confounding effects, such as the influence of ENSO on both interhemispheric transport and surface sources."**

- Figure 6. I suggest swapping panels 6b and 6c. This way, the discussion can focus on the QBO

  first and then on the PLST (i.e., first discuss 6a and 6b, then 6c and 6d). **We have rearranged the discussion to discuss 6a,b,c, and d in order, as for Fig. 7 below. We have also introduced Fig. 6 by describing what it shows and then presenting the salient results.**

- Figure 6 caption. Can the authors specify the units of the AGR in the caption? **Added "atmospheric growth rate (AGR in ppb/yr)"**
- P19L421. I suggest introducing Figure 7 as was done for the previous figures. Rewritten as, "Figure 7, which presents the corresponding correlations for the NH surface N₂O AGR, shows that,…"
- Figure 7. I suggest re-arranging the discussion of Figure 7 so it would smoothly describe 7a,

  7b, 7c and 7d. In alternative, the authors could keep the discussion as it is and re-arrange the panels in Figure 7 accordingly (panels *a* and *b* for temperature and *c* and *d* for QBO). I find the current discussion of Figure 7 (7b, 7d, 7a, 7c) rather counter-intuitive. **We have followed the first suggestion to rearrange the discussion.**

- Figures 7 and 8 captions. As for Figure 6, I suggest specifying the units of the AGR in the caption. ? **Added "atmospheric growth rate (AGR in ppb/yr)"**
- P20L439. Also here, I suggest introducing Figure 8 as for the previous figures. **Rewritten as "Figure 8, which shows correlations between the Niño 3.4 index and the surface N₂O AGR, shows that that the two are significantly anticorrelated (R = -0.5, p < 0.05) for both the global and SH N₂O AGR, …"**
- P20L440. I suggest re-phrasing "...with little to no ...". **Changed to "little or no"**
- P21L450. I suggest re-phrasing "... of the seasonal cycle .... NOAA sites." with "... of the

  seasonal cycle for NOAA sites at remote mid and high latitude." **Replaced with, "for NOAA sites at remote mid and high latitudes."**

- P22L458. "January". Why do the authors mention January if panel 9b and its caption say

  February? **Rearranged the sentence as, "This correlation is shown for February at South Pole in Figure 9b and is observed in both January and February at several extratropical southern NOAA sites including Cape Grim, Tasmania, Palmer Station, Antarctica, and South Pole."**

- P22L464. Similarly to the comment above, why do the authors mention March if panels 9c,d

  and their captions say February? **Reworded as, "GEOSCCM simulates similar correlations between PLST and austral summer N₂O anomalies at these sites, both for N₂O$_{ST}$ and total N₂O in February (Figure 9c,d), and also March, i.e., the correlations are delayed by about 1 month relative to NOAA surface observations."**

- Figure 9 caption. I suggest re-phrasing " a) ... mean seasonal cycle in N2O ..." with "a) ... mean

  seasonal cycles in N2O mixing ratios [ppbv] for the NOAA surface station observations (Obs), and the GEOSCCM total N2O (N2Otot) and stratospheric N2O (N2Ost) ...".
  Also, I suggest re-phrasing "b) NOAA surface N2O seasonal anomalies ... " with "b) NOAA surface seasonal anomalies of N2O mixing ratios [ppbv] ....".

  I also suggest a similar re-phrasing for "Bottom row shows seasonal anomalies for ...." with

"Bottom row shows seasonal anomalies of N2O mixing ratios [ppbv] for ...". **Fig 9 Caption reworded as, "Figure 9:  Top row shows a) mean seasonal cycles in $N_2O$ for NOAA surface station observations (Obs) and GEOSCCM surface total $N_2O$ ($N_2O_{tot}$) and stratospheric $N_2O$ ($N_2O_{ST}$) and b) NOAA surface seasonal anomalies of $N_2O$ mixing ratio (ppb) in February at South Pole spanning 1998-2020, plotted vs. mean lower stratospheric MERRA-2 temperature at 100 hPa averaged over 60-90°S over the previous spring (September-November).  The labeled anomalies in 2016 and 2017 correspond to the year of the ORCAS and ATom-2 aircraft surveys, respectively. Bottom row shows GEOSCCM surface seasonal anomalies of $N_2O$ mixing ratio (ppb) for c) $N_2O_{ST}$ and d) $N_2O_{tot}$ in February at South Pole spanning 2000-2019, plotted vs. mean GEOSCCM lower stratospheric temperature, which is sampled the same way as the MERRA-2 temperature."**

- P23L476. The authors mention "February" but the ORCAS dataset also covers January. **Figure 9 originally showed panels for both January and February (NOAA) and February and March (GEOSCCM), (which has a delayed stratospheric signal at the surface relative to observations).  In the first round of reviews, Reviewer 3 said that the extra panels (January and March) were redundant and should be deleted.  Thus, February was the logical choice as the optimal single month to be featured in Figure 9.**
- Figure 10 caption. I suggest changing "N2O anomalies in ppb ...." with "Anomalies of N2O

  mixing ratios [ppbv] ....".  **Replaced with "Anomalies of N2O in ppb" -see comment 5 response for explanation of why we prefer ppb to ppbv.**

- P24L490-494. I feel that this paragraph gives too much importance to a figure that is not

  shown in the main manuscript. I suggest reducing this paragraph to a sentence that captures

its essence. **Reduced to one sentence, "In contrast to the SH, PLST in the NH from the previous winter is not correlated significantly to $N_2O$ monthly anomalies at extratropical surface sites for either NOAA or GEOSCCM in any of the months surrounding the NH $N_2O$ seasonal minimum, with the exception of Mace Head, Ireland, where a negative correlation is found in July in GEOSCCM (Supplementary Figure S3)."**

- P25L500. I suggest replacing "shows up" with "enters". **Done**
- P25L512. I suggest re-phrasing "...simulates too long a delay ..." with "... simulates a too long

  delay ....". **Done**

- P25L513. "The rate of descent". Can the authors specify what is descending? **Replaced with "diabatic descent"**

P25L517. "... may be overestimated" why do the authors think that the summer soil emission might be overestimated? Was that a conclusion of Liang et al., 2022? **Added some text and 2 references to better explain this point "summer soil emissions are from a soil biogeochemistry model and may be overestimated, leading to unrealistic surface maxima in July (*Saikawa et al.*, 2013; *Nevison et al.*, 2018; *Liang et al.*, 2022)." The model is from Saikawa et al. and is mentioned in Liang et al. The latter doesn't directly state the overestimate although Nevison et al., which used the Saikawa soil source, found that it overestimated summer emissions in North America.**

- P26L534. Given my comment above about the warming due to the BDC, I suggest removing "warm". **Done**
- P26L539-540. I suggest moving the reference to Ray et al. 2020 at the end of the sentence and put it between parentheses (Ray et al., 2020). **Done**
- P27L562. "Paradoxically". Why do they authors say that? Did they expect something different? **Added (i.e., when photochemical destruction is highest). The paradox being that less N2O-poor air is transported toward the poles during this configuration.**
- Section 4.2. Very interesting section. However, I think it can be summarized and merged with Section 4.1 to highlight their main points. When doing that, I suggest clearly separating the discussion between the SH and NH (you could even have a subsection for the SH and one for the NH). **We have consolidated 4.1 and 4.2 into one section. We did not find a good way to separate the SH and NH into subsections since the discussion moves back and forth between the two, comparing and contrasting.**
- P28L589. I suggest removing "ENSO". **Removed**
- P29L611. I suggest re-phrasing "tease out". **Replaced with "infer."**
- All figures. Please replace the occurrences of "ppb" with "ppbv".
  **See response above to comment 5 for explanation of why we stick with ppb.**

---

## Author Response (AR3)

**We thank the reviewers and Dr. Šácha again for their attention and commitment throughout the review process, which we agree was demanding but constructive. Please see our responses to Dr. Šácha's requested corrections below.**

The corrections I ask for concern the captions of Figs. 1 and 9 and the related discussion of results, where you should indicate that you newly incorporate also the uncertainty.

Figure 1:

Caption: We have amended the Fig. 1 caption, "**Figure 1: Detrended seasonal cycles in $N_2O$ mixing ratio (ppb) modeled by GEOSCCM and compared to NOAA surface station data at 9 surface sites, with uncertainty estimated using a bootstrap method.**

Text: We have added a second sentence to the presentation of Fig. 1 in Section 3.1: "Figure 1 shows that the GEOSCCM $N_2O$ mean seasonal cycle at surface sites is dominated by stratospheric air depleted in $N_2O$ that is transported to the surface, rather than by the influence of surface sources. **This dominance holds within the uncertainty of the seasonal cycles, as estimated using a bootstrap method.**"

As before, the bootstrap method is explained in Section 2.3. "Mean seasonal cycles for NOAA surface $N_2O$ observations and GEOSCCM $N_2O$ tracers were estimated using a bootstrap method in which 20% of the timeseries was randomly removed and the remaining 80% was fit to a $3^{rd}$ order polynomial plus first 4 harmonics. These steps were repeated over 500 iterations to estimate the range of uncertainty in the harmonic components of the fit."

Figure 9:

Caption: We have revised the Fig. 9a caption **Top row shows a) mean seasonal cycles in $N_2O$ for NOAA surface station observations (Obs) and GEOSCCM surface total $N_2O$ ($N_2O_{tot}$) and stratospheric $N_2O$ ($N_2O_{ST}$), with uncertainty estimated using a bootstrap method"**

Text: Revised to read, "**"Figure 9a shows that the mean seasonal cycles in SH surface $N_2O$ for NOAA and GEOSCCM total $N_2O$ ($N_2O_{tot}$) and stratospheric $N_2O$ ($N_2O_{ST}$), as illustrated at South Pole (SPO), all have similar autumn seasonal minimum, **within the range of uncertainty as estimated using a bootstrap method**. Figure 9b shows that PLST from the previous spring is significantly anticorrelated to NOAA SPO $N_2O$ monthly anomalies in February, when $N_2O$ is descending into its minimum. This correlation is observed in both January and February at SPO as well as several extratropical southern NOAA sites including Cape Grim, Tasmania and Palmer Station, Antarctica."